# Transductive Active Learning: Theory and Applications

**Jonas Hübotter**[*]
Department of Computer Science
ETH Zürich, Switzerland

**Bhavya Sukhija**
Department of Computer Science
ETH Zürich, Switzerland

**Lenart Treven**
Department of Computer Science
ETH Zürich, Switzerland

**Yarden As**
Department of Computer Science
ETH Zürich, Switzerland

**Andreas Krause**
Department of Computer Science
ETH Zürich, Switzerland

## Abstract

We study a generalization of classical active learning to real-world settings with concrete prediction targets where sampling is restricted to an accessible region of the domain, while prediction targets may lie outside this region. We analyze a family of decision rules that sample adaptively to minimize uncertainty about prediction targets. We are the first to show, under general regularity assumptions, that such decision rules converge uniformly to the smallest possible uncertainty obtainable from the accessible data. We demonstrate their strong sample efficiency in two key applications: active fine-tuning of large neural networks and safe Bayesian optimization, where they achieve state-of-the-art performance.

## 1 Introduction

Machine learning, at its core, is about designing systems that can extract knowledge or patterns from data. One part of this challenge is determining not just how to learn given observed data but deciding what data to obtain next, given the information already available. More formally, given an unknown and sufficiently regular function $f$ over a domain $\mathcal{X}$: *How can we learn $f$ sample-efficiently from (noisy) observations?* This problem is widely studied in *active learning* and *experimental design* (Chaloner & Verdinelli, 1995; Settles, 2009).

Active learning methods commonly aim to learn $f$ globally, i.e., across the entire domain $\mathcal{X}$. However, in many real-world problems, **(i)** the domain is so large that learning $f$ globally is hopeless or **(ii)** agents have limited information and cannot access the entire domain (e.g., due to restricted access or to act safely). Thus, global learning is often not desirable or even possible. Instead, intelligent systems are typically required to act in a more *directed* manner and *extrapolate* beyond their limited information. This work formalizes the above two aspects of active learning, which have remained largely unaddressed by prior work. We provide a comprehensive overview of related work in Section 6.

**"Directed" transductive active learning** We consider the generalized problem of *transductive active learning*, where given two arbitrary subsets of the domain $\mathcal{X}$; a *target space* $\mathcal{A} \subseteq \mathcal{X}$, and a *sample space* $\mathcal{S} \subseteq \mathcal{X}$, we study the question:

*How can we learn $f$ within $\mathcal{A}$ by actively sampling observations within $\mathcal{S}$?*

---

[*]Correspondence to `jonas.huebotter@inf.ethz.ch`

38th Conference on Neural Information Processing Systems (NeurIPS 2024).

This problem is ubiquitous in real-world applications such as safe Bayesian optimization, where $\mathcal{S}$ is a set of safe parameters and $\mathcal{A}$ might represent parameters outside $\mathcal{S}$ whose safety we want to infer. Active fine-tuning of neural networks is another example, where the target space $\mathcal{A}$ represents the test set over which we want to minimize risk, and the sample space $\mathcal{S}$ represents the dataset from which we can retrieve data points to fine-tune our model to $\mathcal{A}$. Figure 1 visualizes some instances of transductive active learning.

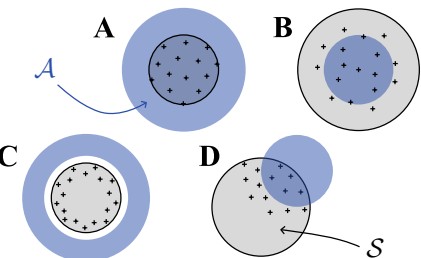

Figure 1: Instances of transductive active learning with target space $\mathcal{A}$ shown in blue and sample space $\mathcal{S}$ shown in gray. The points denote plausible observations within $\mathcal{S}$ to "learn" $\mathcal{A}$. In **(A)**, the target space contains "everything" within $\mathcal{S}$ as well as points *outside* $\mathcal{S}$. In **(B, C, D)**, one makes observations *directed* towards learning about a particular target. Prior work on inductive active learning has focused on the instance $\mathcal{A} = \mathcal{S}$.

Whereas most prior work has focused on the "global" inductive instance $\mathcal{X} = \mathcal{A} = \mathcal{S}$, MacKay (1992) was the first to consider specific target spaces $\mathcal{A}$ and proposed the principle of selecting points in $\mathcal{S}$ to minimize the "posterior uncertainty" about points in $\mathcal{A}$. Since then, several works have studied this principle empirically (e.g., Seo et al., 2000; Yu et al., 2006; Bogunovic et al., 2016; Wang et al., 2021; Kothawade et al., 2021; Bickford Smith et al., 2023). In this work, we model $f$ as a Gaussian process or (equivalently) as a function in a reproducing kernel Hilbert space, for which the above principle is analytically and computationally tractable. Our contributions are:

- **Theory (Section 3):** We are the first to give rates for the uniform convergence of uncertainty over the target space $\mathcal{A}$ to the smallest attainable value, given samples from the sample space $\mathcal{S}$ (Theorems 3.2 and 3.3), Our results provide a theoretical justification for the principle of minimizing posterior uncertainty in transductive active learning, and indicate that transductive active learning can be more sample efficient than inductive active learning.

- **Applications:** We show that transductive active learning improves upon the state-of-the-art in the batch-wise *active fine-tuning* of neural networks for image classification (Section 4) and in *safe Bayesian optimization* (Section 5).

## 2 Problem Setting

We assume for now that the target space $\mathcal{A}$ and sample space $\mathcal{S}$ are finite, and relax these assumptions in the appendices. We model $f$ as a stochastic process and denote the marginal random variables $f(\boldsymbol{x})$ by $f_{\boldsymbol{x}}$, and joint random vectors $\{f_{\boldsymbol{x}}\}_{\boldsymbol{x} \in X}$ for some $X \subseteq \mathcal{X}, |X| < \infty$ by $\boldsymbol{f}_X$. Let $\boldsymbol{y}_X$ denote the noisy observations of $\boldsymbol{f}_X$, $\{y_{\boldsymbol{x}} = f_{\boldsymbol{x}} + \varepsilon_{\boldsymbol{x}}\}_{\boldsymbol{x} \in X}$, where $\varepsilon_{\boldsymbol{x}}$ is independent noise.[2] We study the "adaptive" setting, where in round $n$ the agent selects a point $\boldsymbol{x}_n \in \mathcal{S}$ and observes $y_n = y_{\boldsymbol{x}_n}$. The agent's choice of $\boldsymbol{x}_n$ may depend on the outcome of prior observations $\mathcal{D}_{n-1} \stackrel{\text{def}}{=} \{(\boldsymbol{x}_i, y_i)\}_{i<n}$.

**Background on information theory** We briefly recap several important concepts from information theory of which we provide formal definitions in Appendix B. The (differential) entropy $\mathrm{H}[\boldsymbol{f}]$ is one possible measure of uncertainty about $\boldsymbol{f}$ and the conditional entropy $\mathrm{H}[\boldsymbol{f} \mid \boldsymbol{y}]$ is the (expected) posterior uncertainty about $\boldsymbol{f}$ after observing $\boldsymbol{y}$. The information gain $\mathrm{I}(\boldsymbol{f}; \boldsymbol{y}) = \mathrm{H}[\boldsymbol{f}] - \mathrm{H}[\boldsymbol{f} \mid \boldsymbol{y}]$ measures the (expected) reduction in uncertainty about $\boldsymbol{f}$ due to $\boldsymbol{y}$. We denote the information gain about $\mathcal{A}$ from observing $X$ by $\mathrm{I}(\boldsymbol{f}_{\mathcal{A}}; \boldsymbol{y}_X)$. The maximum information gain about $\mathcal{A}$ from $n$ observations within $\mathcal{S}$ is

$$\gamma_{\mathcal{A},\mathcal{S}}(n) \stackrel{\text{def}}{=} \max_{\substack{X \subseteq \mathcal{S} \\ |X| \leq n}} \mathrm{I}(\boldsymbol{f}_{\mathcal{A}}; \boldsymbol{y}_X).$$

This "information capacity" measures the information about $\boldsymbol{f}_{\mathcal{A}}$ that is accessible from within $\mathcal{S}$, and has been used previously (e.g., by Srinivas et al., 2009; Chowdhury & Gopalan, 2017; Vakili et al., 2021) in the setting where $\mathcal{X} = \mathcal{A} = \mathcal{S}$, taking the form of $\gamma_n \stackrel{\text{def}}{=} \gamma_{\mathcal{X}}(n) \stackrel{\text{def}}{=} \gamma_{\mathcal{X},\mathcal{X}}(n)$. We remark that $\gamma_{\mathcal{A},\mathcal{S}}(n) \leq \gamma_{\mathcal{S}}(n)$ holds uniformly for all $\mathcal{A}, \mathcal{S}$, and $n$ due to the data processing inequality. Generally, $\gamma_{\mathcal{A},\mathcal{S}}(n)$ can be substantially smaller if the target space is a sparse subset of the sample space.

---

[2] $X$ may be a multiset in which case repeated occurrence of $\boldsymbol{x}$ corresponds to independent observations of $y_{\boldsymbol{x}}$.

# 3 Main Results

We analyze the following principle for transductive active learning:

> *Select samples to minimize the posterior "uncertainty" about $f$ within $\mathcal{A}$.*     (†)

This principle yields a family of simple and natural decision rules which depend on the chosen measure of "uncertainty". Two natural measures of uncertainty are (1) the entropy of prediction targets, $\mathrm{H}[\boldsymbol{f}_\mathcal{A}]$, and (2) their total variance, $\sum_{\boldsymbol{x}' \in \mathcal{A}} \mathrm{Var}[f_{\boldsymbol{x}'}]$. The corresponding decision rules are

$$(1) \qquad \boldsymbol{x}_n = \arg\min_{\boldsymbol{x} \in \mathcal{S}} \mathrm{H}[\boldsymbol{f}_\mathcal{A} \mid \mathcal{D}_{n-1}, y_{\boldsymbol{x}}] = \arg\max_{\boldsymbol{x} \in \mathcal{S}} \mathrm{I}(\boldsymbol{f}_\mathcal{A}; y_{\boldsymbol{x}} \mid \mathcal{D}_{n-1}), \qquad (\text{ITL})$$

$$(2) \qquad \boldsymbol{x}_n = \arg\min_{\boldsymbol{x} \in \mathcal{S}} \mathrm{tr}\, \mathrm{Var}[\boldsymbol{f}_\mathcal{A} \mid \mathcal{D}_{n-1}, y_{\boldsymbol{x}}] \qquad\qquad\qquad (\text{VTL})$$

with an implicit expectation over the feedback $y_{\boldsymbol{x}}$. That is, ITL (short for ***Information-based Transductive Learning***) and VTL (***Variance-based TL***) select $\boldsymbol{x}_n$ so as to minimize the uncertainty about the prediction targets $\boldsymbol{f}_\mathcal{A}$ (in expectation) after having received the feedback $y_n$. Unlike VTL, ITL takes into account the mutual dependence between points in $\mathcal{A}$. These decision rules were suggested previously (MacKay, 1992; Seo et al., 2000; Yu et al., 2006) without deriving theoretical guarantees; and they generalize several widely used algorithms which we discuss in more detail in Section 6. Most prominently, in the inductive setting where $\mathcal{S} \subseteq \mathcal{A}$, ITL reduces to $\boldsymbol{x}_n = \arg\max_{\boldsymbol{x} \in \mathcal{S}} \mathrm{I}(f_{\boldsymbol{x}}; y_{\boldsymbol{x}} \mid \mathcal{D}_{n-1})$, i.e., is "undirected" and reduces to standard uncertainty-based active learning strategies (cf. Appendix C.1). The convergence properties for the special instance of ITL with $\mathcal{S} = \mathcal{A}$ have been studied extensively. To the best of our knowledge, we are the first to extend these guarantees to the more general setting of transductive active learning.

In our presented results, we make the following assumption.

**Assumption 3.1.** In the case of ITL, the information gain $\psi_\mathcal{A}(X) = \mathrm{I}(\boldsymbol{f}_\mathcal{A}; \boldsymbol{y}_X)$ is submodular. In the case of VTL, the variance reduction $\psi_\mathcal{A}(X) = \mathrm{tr}\,\mathrm{Var}[\boldsymbol{f}_\mathcal{A}] - \mathrm{tr}\,\mathrm{Var}[\boldsymbol{f}_\mathcal{A} \mid \boldsymbol{y}_X]$ is submodular.

Under this assumption, $\psi_\mathcal{A}(\boldsymbol{x}_{1:n})$ is a constant factor approximation of $\max_{X \subseteq \mathcal{S}, |X| \le n} \psi_\mathcal{A}(X)$ due to the seminal result on submodular function maximization by Nemhauser et al. (1978). Similar assumptions have been made, e.g., by Bogunovic et al. (2016) and Kothawade et al. (2021). Assumption 3.1 is satisfied exactly for ITL when $\mathcal{S} \subseteq \mathcal{A}$ and $f$ is a Gaussian process (cf. Lemma C.9), and we provide an extensive discussion of our results in Appendix C.4 for instances where Assumption 3.1 is satisfied approximately, relying on the notion of weak submodularity (Das & Kempe, 2018).

## 3.1 Gaussian Process Setting

When $f \sim \mathcal{GP}(\mu, k)$ is a Gaussian process (GP, Williams & Rasmussen, 2006) with known mean function $\mu$ and kernel $k$, and the noise $\varepsilon_{\boldsymbol{x}}$ is mutually independent and zero-mean Gaussian with known variance, the ITL and VTL objectives have a closed form expression (cf. Appendix F) and can be optimized efficiently. Further, the information capacity $\gamma_n$ is sublinear in $n$ for a rich class of GPs (Srinivas et al., 2009; Vakili et al., 2021), with rates summarized in Table 3 of the appendix.

**Convergence to irreducible uncertainty** So far, our discussion was centered around the role of the target space $\mathcal{A}$ in facilitating *directed* learning. An orthogonal contribution of this work is to study *extrapolation* from the sample space $\mathcal{S}$ to points $\boldsymbol{x} \in \mathcal{A} \setminus \mathcal{S}$. To this end, we derive bounds on the marginal posterior variance $\sigma_n^2(\boldsymbol{x}) \stackrel{\text{def}}{=} \mathrm{Var}[f(\boldsymbol{x}) \mid \mathcal{D}_n]$ for points in $\mathcal{A}$. These bounds depend on the instance of transductive active learning (i.e., $\mathcal{A}$ and $\mathcal{S}$) and might be of independent interest for active learning. For ITL and VTL, they imply uniform convergence of the variance for a rich class of GPs. To the best of our knowledge, this work is the first to present such bounds.

We define the *irreducible uncertainty* as the variance of $f(\boldsymbol{x})$ provided complete knowledge of $f$ in $\mathcal{S}$:

$$\eta_\mathcal{S}^2(\boldsymbol{x}) \stackrel{\text{def}}{=} \mathrm{Var}[f_{\boldsymbol{x}} \mid \boldsymbol{f}_\mathcal{S}].$$

As the name suggests, $\eta_\mathcal{S}^2(\boldsymbol{x})$ represents the smallest uncertainty one can hope to achieve from observing only within $\mathcal{S}$. For all $\boldsymbol{x} \in \mathcal{S}$, it is easy to see that $\eta_\mathcal{S}^2(\boldsymbol{x}) = 0$. However, the irreducible uncertainty of $\boldsymbol{x} \notin \mathcal{S}$ may be (and typically is!) strictly positive.

**Theorem 3.2** (Bound on marginal variance for ITL and VTL). *Let Assumption 3.1 hold and the data be selected by either ITL or VTL.. Assume that $f \sim \mathcal{GP}(\mu, k)$ with known mean function $\mu$*

*and kernel k, the noise $\varepsilon_{\boldsymbol{x}}$ is mutually independent and zero-mean Gaussian with known variance, and $\gamma_n$ is sublinear in $n$. Then there exists a constant $C$ such that for any $n \geq 1$ and $\boldsymbol{x} \in \mathcal{A}$,*

$$\sigma_n^2(\boldsymbol{x}) \leq \underbrace{\eta_{\mathcal{S}}^2(\boldsymbol{x})}_{irreducible} + \underbrace{C\frac{\gamma_{\mathcal{A},\mathcal{S}}(n)}{\sqrt{n}}}_{reducible}. \tag{1}$$

*Moreover, if $\boldsymbol{x} \in \mathcal{A} \cap \mathcal{S}$, there exists a constant $C'$ such that*

$$\sigma_n^2(\boldsymbol{x}) \leq C'\frac{\gamma_{\mathcal{A},\mathcal{S}}(n)}{n}. \tag{2}$$

Intuitively, Equation (1) of Theorem 3.2 can be understood as bounding an epistemic "generalization gap" (Wainwright, 2019) of the learner. The reducible uncertainty converges to zero at all prediction targets $\boldsymbol{x} \in \mathcal{A}$, e.g., for linear, Gaussian, and smooth Matérn kernels. As to be expected, a smaller target space (i.e., more targeted sampling) leads to faster convergence due to a smaller information capacity $\gamma_{\mathcal{A},\mathcal{S}}(n) \ll \gamma_n$. Equation (2) matches prior results for the setting $\mathcal{S} = \mathcal{A}$. We provide a formal proof of Theorem 3.2 in Appendix C.6.

### 3.2 Agnostic Setting

The result from the GP setting translates also to the agnostic setting, where the "ground truth" $f^\star$ may be any sufficiently regular fixed function on $\mathcal{X}$.[3] In this case, we use the model $f$ from Section 3.1 as a (misspecified) model of $f^\star$, with some kernel $k$ and zero mean function $\mu(\cdot) = 0$. We denote by $\mu_n(\boldsymbol{x}) = \mathbb{E}[f(\boldsymbol{x}) \mid \mathcal{D}_n]$ the posterior mean of $f$. W.l.o.g. we assume in the following result that the prior variance is bounded, i.e., $\mathrm{Var}[f(\boldsymbol{x})] \leq 1$.

**Theorem 3.3** (Bound on approximation error for ITL and VTL, following Abbasi-Yadkori (2013); Chowdhury & Gopalan (2017))**.** *Let Assumption 3.1 hold and the data be selected by either ITL or VTL. Pick any $\delta \in (0,1)$. Assume that $f^\star$ lies in the reproducing kernel Hilbert space $\mathcal{H}_k(\mathcal{X})$ of the kernel $k$ with norm $\|f^\star\|_k < \infty$, the noise $\varepsilon_n$ is conditionally $\rho$-sub-Gaussian, and $\gamma_n$ is sublinear in $n$. Let $\beta_n(\delta) = \|f^\star\|_k + \rho\sqrt{2(\gamma_n + 1 + \log(1/\delta))}$. Then for any $n \geq 1$ and $\boldsymbol{x} \in \mathcal{A}$, jointly with probability at least $1 - \delta$,*

$$|f^\star(\boldsymbol{x}) - \mu_n(\boldsymbol{x})| \leq \beta_n(\delta)\Big[ \underbrace{\eta_{\mathcal{S}}(\boldsymbol{x})}_{irreducible} + \underbrace{\nu_{\mathcal{A},\mathcal{S}}(n)}_{reducible} \Big]$$

*where $\nu_{\mathcal{A},\mathcal{S}}^2(n)$ denotes the reducible part of Equation (1).*

We provide a formal proof of Theorem 3.3 in Appendix C.7. Theorem 3.3 generalizes approximation error bounds of prior works to the extrapolation setting, where some prediction targets $\boldsymbol{x} \in \mathcal{A}$ lie outside the sample space $\mathcal{S}$. For prediction targets $\boldsymbol{x} \in \mathcal{A} \cap \mathcal{S}$, the irreducible uncertainty vanishes, and we recover previous results from the setting $\mathcal{S} = \mathcal{A}$.

Theorems 3.2 and 3.3 show that ITL and VTL efficiently learn $f$ at the prediction targets $\mathcal{A}$ for large classes of "sufficiently regular" functions $f$. In the following, we validate these results experimentally by showing that ITL and VTL exhibit strong empirical performance in a broad range of applications.

### 3.3 Experiments in the Gaussian Process Setting

Before demonstrating ITL and VTL on GPs to develop more intuition, we introduce a natural correlation-based baseline, which will later uncover connections to existing approaches:

$$\boldsymbol{x}_n = \arg\max_{\boldsymbol{x} \in \mathcal{S}} \sum_{\boldsymbol{x}' \in \mathcal{A}} \mathrm{Cor}[f_{\boldsymbol{x}}, f_{\boldsymbol{x}'} \mid \mathcal{D}_{n-1}]. \tag{CTL}$$

**How does the smoothness of $f$ affect ITL?** We contrast two "extreme" kernels: the *Gaussian kernel* $k(\boldsymbol{x}, \boldsymbol{x}') = \exp(-\|\boldsymbol{x} - \boldsymbol{x}'\|_2^2/2)$ and the *Laplace kernel* $k(\boldsymbol{x}, \boldsymbol{x}') = \exp(-\|\boldsymbol{x} - \boldsymbol{x}'\|_1)$. In the mean-squared sense, the Gaussian kernel yields a smooth process $f$ whereas the Laplace kernel yields a continuous but non-differentiable $f$ (Williams & Rasmussen, 2006). Figure 2 shows

---

[3]Here $f^\star(\boldsymbol{x})$ denotes the mean observation $y_{\boldsymbol{x}} = f^\star(\boldsymbol{x}) + \epsilon_{\boldsymbol{x}}$

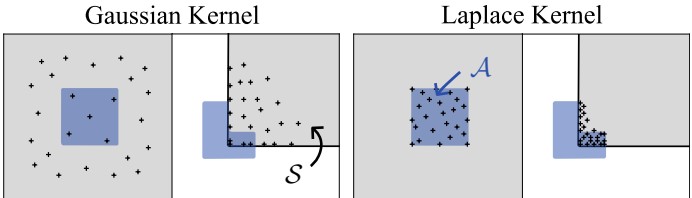

Figure 2: Initial 25 samples of ITL under a Gaussian kernel with lengthscale 1 (left) and a Laplace kernel with lengthscale 10 (right). Shown in gray is the sample space $\mathcal{S}$ and shown in blue is the target space $\mathcal{A}$. In three of the four examples, points outside the target space provide useful information.

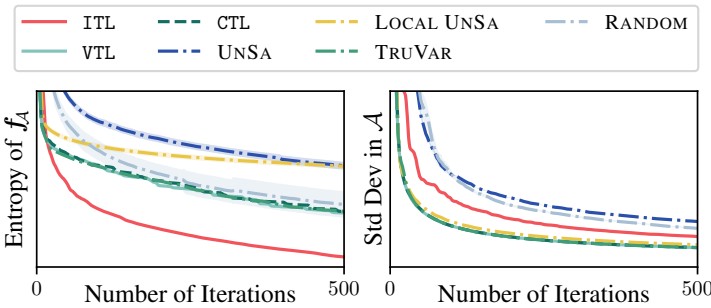

Figure 3: Entropy of $\mathbf{f}_{\mathcal{A}}$ ranging from $-3850$ to $-3725$ and the mean marginal standard deviations of $\mathbf{f}_{\mathcal{A}}$ ranging from $0$ to $0.15$. Experiment is using the Gaussian kernel of the left instance ($\mathcal{A} \subset \mathcal{S}$) from Figure 2. It can be seen that ITL and VTL outperform UNSA and RANDOM. Uncertainty bands correspond to one standard error over 10 random seeds.

how ITL adapts to the smoothness of $f$: Under the "smooth" Gaussian kernel, points outside $\mathcal{A}$ provide higher-order information. In contrast, under the "rough" Laplace kernel and if $\mathcal{A} \subseteq \mathcal{S}$, points outside $\mathcal{A}$ do not provide any additional information, and therefore are not sampled by ITL. If, however, $\mathcal{A} \not\subseteq \mathcal{S}$, information "leaks" $\mathcal{A}$ even under a Laplace kernel prior. That is, even for non-smooth functions, the point with most information need not be in $\mathcal{A}$.

**Does ITL outperform uncertainty sampling?** Uncertainty sampling (UNSA, Lewis & Catlett, 1994) is one of the most popular active learning methods. UNSA selects points $\mathbf{x}$ with high *prior* uncertainty: $\mathbf{x}_n = \arg\max_{\mathbf{x} \in \mathcal{S}} \sigma_{n-1}^2(\mathbf{x})$. This is in stark contrast to ITL and VTL which select points $\mathbf{x}$ that minimize *posterior* (epistemic) uncertainty about $\mathcal{A}$. It can be seen that UNSA is the special "undirected" case of ITL when $\mathcal{S} \subseteq \mathcal{A}$ and observation noise is homoscedastic (cf. Appendix C.1).

We compare UNSA to ITL, VTL, and CTL in Figure 3. We observe that ITL and VTL outperform UNSA which also samples points that are not informative about $\mathcal{A}$. Further, ITL and VTL outperform "local" UNSA (i.e., UNSA constrained to $\mathcal{A} \cap \mathcal{S}$) which neglects all information provided by points outside $\mathcal{A}$.[4] As one would expect, VTL has an advantage with respect to reducing the total variance of $\mathbf{f}_{\mathcal{A}}$, whereas ITL reduces the entropy of $\mathbf{f}_{\mathcal{A}}$ faster. We include ablations in Appendix H where we, in particular, observe that the advantage of ITL and VTL over UNSA increases as the volume of prediction targets shrinks in comparison to the size of domain.

## 4 Active Fine-Tuning of Neural Networks

Fine-tuning a large pre-trained model is a cost- and computation-effective approach to improve performance on a given target domain (Lee et al., 2022). While previous work has studied the effectiveness of various training procedures for fine-tuning (e.g., Eustratiadis et al., 2024), we ask: *How can we select the right data for fine-tuning to a specific task?* This *active* fine-tuning problem is an instance of the introduced "directed" transductive learning problem: Concretely, consider a supervised setting, where the function $f$ maps inputs $\mathbf{x} \in \mathcal{X}$ to outputs $y \in \mathcal{Y}$. We have access

---

[4]If $\mathcal{A} \not\subseteq \mathcal{S}$ then "local" UNSA does *not even* converge to the irreducible uncertainty.

to noisy samples from a training set $\mathcal{S}$ on $\mathcal{X}$, and we would like to learn $f$ such that our estimate minimizes a given risk measure, such as classification error, with respect to a test distribution $\mathcal{P}_{\mathcal{A}}$ on $\mathcal{X}$. The goal is to actively and efficiently sample from $\mathcal{S}$ to minimize risk with respect to $\mathcal{P}_{\mathcal{A}}$.[5] We show in this section that ITL and VTL can learn $f$ from only *few examples* from $\mathcal{S}$.

**How can we leverage the latent structure learned by the pre-trained model?** As common in related work, we approximate the (pre-trained) neural network (NN) $f(\cdot; \boldsymbol{\theta})$ as a linear function in a latent embedding space, $f(\boldsymbol{x}; \boldsymbol{\theta}) \approx \boldsymbol{\beta}^{\top} \boldsymbol{\phi}_{\boldsymbol{\theta}}(\boldsymbol{x})$, with weights $\boldsymbol{\beta} \in \mathbb{R}^{p}$ and embeddings $\boldsymbol{\phi}_{\boldsymbol{\theta}} : \mathcal{X} \to \mathbb{R}^{p}$. Common choices of embeddings include last-layer embeddings (Devlin et al., 2019; Holzmüller et al., 2023), neural tangent embeddings arising from neural tangent kernels (Jacot et al., 2018) which are motivated by their relationship to the training and fine-tuning of ultra-wide NNs (Arora et al., 2019; Lee et al., 2019; Khan et al., 2019; He et al., 2020; Malladi et al., 2023), and loss gradient embeddings (Ash et al., 2020). We provide a comprehensive overview of embeddings in Appendix J.2. Now, supposing the prior $\boldsymbol{\beta} \sim \mathcal{N}(\mathbf{0}, \boldsymbol{\Sigma})$, often with $\boldsymbol{\Sigma} = \boldsymbol{I}$ (Khan et al., 2019; He et al., 2020; Antorán et al., 2022; Wei et al., 2022), this approximation of $f$ is a Gaussian process with kernel $k(\boldsymbol{x}, \boldsymbol{x}') = \boldsymbol{\phi}_{\boldsymbol{\theta}}(\boldsymbol{x})^{\top} \boldsymbol{\Sigma} \boldsymbol{\phi}_{\boldsymbol{\theta}}(\boldsymbol{x}')$ which quantifies the similarity between points in terms of their alignment in the learned latent space. Note that the correlation $k(\boldsymbol{x}, \boldsymbol{x}') / \sqrt{k(\boldsymbol{x}, \boldsymbol{x})k(\boldsymbol{x}', \boldsymbol{x}')}$ between two points $\boldsymbol{x}, \boldsymbol{x}'$ is equal to the cosine similarity of their embeddings.

In this context, Theorem 3.2 bounds the epistemic posterior uncertainty about a prediction using the approximation $\boldsymbol{\beta}^{\top} \boldsymbol{\phi}_{\boldsymbol{\theta}}(\boldsymbol{x})$, given that the model is trained using data selected by ITL or VTL. Theorem 3.3 bounds the generalization error when using the posterior mean of $\boldsymbol{\beta}$ for prediction. This extends recent work which has studied estimators of this generalization error (Wei et al., 2022).

**Batch selection: Diversity via conditional embeddings** Efficient labeling and training necessitates a batch-wise selection of inputs. The selection of a batch of size $b > 1$ can be seen as an individual *non-adaptive* active learning problem, and significant recent work has shown that batch diversity is crucial in this setting (Ash et al., 2020; Zanette et al., 2021; Holzmüller et al., 2023; Pacchiano et al., 2024). An information-based batch-wise selection strategy is formalized by the following non-adaptive transductive active learning problem (Chen & Krause, 2013) and the greedy approximation of $B_n$ by ITL which selects elements $\boldsymbol{x}_{n,i}$ of the $n$-th batch iteratively based on $\boldsymbol{x}_{n,1:i-1}$:

$$B_n = \arg\max_{B \subseteq \mathcal{S}, |B|=b} \mathrm{I}(\boldsymbol{f}_{\mathcal{A}}; \boldsymbol{y}_B \mid \mathcal{D}_{n-1}); \qquad \boldsymbol{x}_{n,i} = \arg\max_{\boldsymbol{x} \in \mathcal{S}} \mathrm{I}(\boldsymbol{f}_{\mathcal{A}}; y_{\boldsymbol{x}} \mid \mathcal{D}_{n-1}, \boldsymbol{y}_{\boldsymbol{x}_{n,1:i-1}}). \tag{3}$$

The batch $B_n$ is diverse and informative by design. We show that under Assumption 3.1, $B'_n = \boldsymbol{x}_{n,1:b}$ yields a constant-factor approximation of $B_n$ (cf. Appendix C.3).

## 4.1 Experiments on Active Fine-Tuning

Our empirical evaluation is motivated by the following practical example: We deploy a pre-trained image classifier to user's phones who use it within their local environment. We would like to locally fine-tune a user's model to their environment. Since the users' images $\mathcal{A}$ are unlabeled, this requires selecting a small number of relevant and diverse images from the set of labeled images $\mathcal{S}$. As such, we will focus here on the setting where the points in our test set do not lie in our training set (i.e., $\mathcal{A} \cap \mathcal{S} = \emptyset$), and discuss alternative instances such as active domain adaptation in Appendix I.

**Testbeds & architectures** We use the MNIST (LeCun et al., 1998) and CIFAR-100 (Krizhevsky et al., 2009) datasets as testbeds. In both cases, we take $\mathcal{S}$ to be the training set, and we consider the task of learning the digits 3, 6, and 9 (MNIST) or the first 10 categories of CIFAR-100.[6] For MNIST, we train a simple convolutional neural network with ReLU activations, three convolutional layers with max-pooling, and two fully-connected layers. For CIFAR-100, we fine-tune an EfficientNet-B0 (Tan & Le, 2019) pre-trained on ImageNet (Deng et al., 2009), augmented by a final fully-connected layer. We train the NNs using the cross-entropy loss and the ADAM optimizer (Kingma & Ba, 2014).

**Results** In Figure 4, We compare against **(i)** active learning methods which largely aim for sample diversity but which are not directed towards the target distribution $\mathcal{P}_{\mathcal{A}}$ (e.g., BADGE; Ash et al., 2020), and **(ii)** search methods that aim to retrieve the most relevant samples from $\mathcal{S}$ with respect to the targets $\mathcal{P}_{\mathcal{A}}$ (e.g., maximizing cosine similarity to target embeddings as is common in vector databases;

---

[5]The setting with target distributions $\mathcal{P}_{\mathcal{A}}$ can be reduced to considering target sets $\mathcal{A}$ (cf. Appendix E).

[6]That is, we restrict $\mathcal{P}_{\mathcal{A}}$ to the support of points with labels $\{3, 6, 9\}$ (MNIST) or labels $\{0, \ldots, 9\}$ (CIFAR-100) and train a neural network using few examples drawn from the training set $\mathcal{S}$.

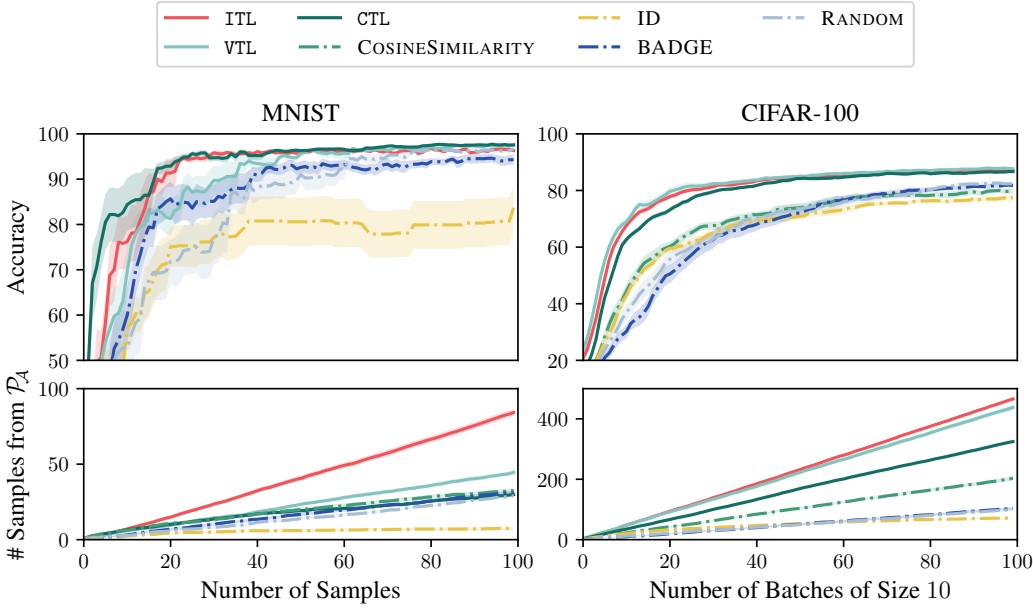

Figure 4: Active fine-tuning on MNIST (left) and CIFAR-100 (right). RANDOM selects each observation uniformly at random from $\mathcal{S}$. The batch size is 1 for MNIST and 10 for CIFAR-100. Uncertainty bands correspond to one standard error over 10 random seeds. We see that transductive active learning with ITL and VTL significantly outperforms competing methods, and in particular, retrieves substantially more samples from the support of $\mathcal{P_A}$. See Appendix J for details and ablations.

Settles & Craven, 2008; Johnson et al., 2019). INFORMATIONDENSITY (ID, Settles & Craven, 2008) is a heuristic approach aiming to combine **(i)** diversity and **(ii)** relevance. In Appendix J.5, we also compare against a wide range of additional baselines (e.g., CORESET (Sener & Savarese, 2017), TYPICLUST (Hacohen et al., 2022), PROBCOVER (Yehuda et al., 2022), etc.) that fall into one of the categories **(i)** and **(ii)**, and which perform similar to the baselines listed here.

We observe that ITL, VTL, and CTL consistently and significantly outperform random sampling from $\mathcal{S}$ as well as all baselines. We see that relevance-based methods such as COSINESIMILARITY have an initial advantage over RANDOM but for batch sizes larger than 1 they quickly fall behind due to diminishing informativeness of the selected data. In contrast, diversity-based methods such as BADGE are more competitive with RANDOM but do not explicitly aim to retrieve relevant samples.

Remarkably, transductive active learning outperforms random data selection even in the MNIST experiment where the model is randomly initialized. This suggests that the learned embeddings can be informative for data selection even in the early stages of training, bootstrapping the learning progress.

**Balancing sample relevance and diversity** Our proposed methods unify approaches to coverage (promoting *diverse* samples) and search (aiming for *relevant* samples with respect to a given query $\mathcal{A}$) which leads to the significant improvement upon the state-of-the-art in Figure 4. Notably, for a batch size and query size of 1 and if correlations are non-negative, ITL, VTL, CTL, and the canonical cosine similarity are equivalent. CTL can be seen as a direct generalization of cosine similarity-based retrieval to batch and query sizes larger than one. In contrast to CTL, ITL and VTL may also sample points which exhibit a strong negative correlation (which is also informative).

We observe empirically that ITL obtains samples from $\mathcal{P_A}$ at more than twice the rate of COSINESIMILARITY, which translates to a significant improvement in accuracy in more difficult learning tasks, while requiring fewer (labeled) samples from $\mathcal{S}$. This phenomenon manifests for both MNIST and CIFAR-100, as well as imbalanced datasets $\mathcal{S}$ or imbalanced reference samples from $\mathcal{P_A}$ (cf. Appendix J.6). The improvement in accuracy appears to increase in the large-data regime, where the learning tasks become more difficult. Akin to a previously identified scaling trend with size of the pre-training dataset (Tamkin et al., 2022), this suggests a potential scaling trend where the improvement of ITL over random batch selection grows as models are fine-tuned on a larger pool of data.

**Towards task-driven few-shot learning**   Being able to efficiently and automatically select data may allow dynamic few-shot fine-tuning to individual tasks (Vinyals et al., 2016; Hardt & Sun, 2024), e.g., fine-tuning the model to each test point / query / prompt. Such task-driven few-shot learning can be seen as a form of "memory recall" akin to associative memory (Hopfield, 1982). Our results are a first indication that task-driven learning can lead to substantial performance gains, and we believe that this is a promising direction for future studies.

# 5    Safe Bayesian Optimization

Another practical problem that can be cast as "directed" learning is safe Bayesian optimization (Safe BO, Sui et al., 2015; Berkenkamp et al., 2021) which has applications in natural science (Cooper & Netoff, 2022) and robotics (Wischnewski et al., 2019; Sukhija et al., 2023; Widmer et al., 2023). Safe BO solves the following optimization problem

$$\max_{\boldsymbol{x} \in \mathcal{S}^\star} f^\star(\boldsymbol{x}) \quad \text{where} \quad \mathcal{S}^\star = \{\boldsymbol{x} \in \mathcal{X} \mid g^\star(\boldsymbol{x}) \geq 0\} \tag{4}$$

which can be generalized to multiple constraints. The functions $f^\star$ and $g^\star$, and hence also the "safe set" $\mathcal{S}^\star$, are unknown and have to be actively learned from data. However, it is crucial that the data collection does not violate the constraint, i.e., $\boldsymbol{x}_n \in \mathcal{S}^\star, \forall n \geq 1$.

**Safe Bayesian optimization as Transductive Active Learning**   In the agnostic setting from Section 3.2, GPs $f$ and $g$ can be used as well-calibrated models of the ground truths $f^\star$ and $g^\star$, and we denote lower- and upper-confidence bounds by $l_n^f(\boldsymbol{x}), l_n^g(\boldsymbol{x})$ and $u_n^f(\boldsymbol{x}), u_n^g(\boldsymbol{x})$, respectively. These confidence bounds induce a *pessimistic* safe set $\mathcal{S}_n \stackrel{\text{def}}{=} \{\boldsymbol{x} \mid l_n^g(\boldsymbol{x}) \geq 0\}$ and an *optimistic* safe set $\widehat{\mathcal{S}}_n \stackrel{\text{def}}{=} \{\boldsymbol{x} \mid u_n^g(\boldsymbol{x}) \geq 0\}$ which satisfy $\mathcal{S}_n \subseteq \mathcal{S}^\star \subseteq \widehat{\mathcal{S}}_n$ with high probability at all times. Similarly, the set of *potential maximizers*

$$\mathcal{A}_n \stackrel{\text{def}}{=} \{\boldsymbol{x} \in \widehat{\mathcal{S}}_n \mid u_n^f(\boldsymbol{x}) \geq \max_{\boldsymbol{x}' \in \mathcal{S}_n} l_n^f(\boldsymbol{x}')\} \tag{5}$$

contains the solution to Equation (4) at all times with high probability.

The (simple) regret $r_n(\mathcal{S}) \stackrel{\text{def}}{=} \max_{\boldsymbol{x} \in \mathcal{S}} f^\star(\boldsymbol{x}) - f^\star(\widehat{\boldsymbol{x}}_n)$ with $\widehat{\boldsymbol{x}}_n \stackrel{\text{def}}{=} \arg\max_{\boldsymbol{x} \in \mathcal{S}_n} l_n^f(\boldsymbol{x})$ measures the worst-case performance of a decision rule. To achieve small regret, one faces an *exploration-expansion* dilemma wherein one needs to explore points that are known-to-be-safe, i.e., lie in the estimated safe set $\mathcal{S}_n$, and might be optimal, while at the same time discovering new safe points by "expanding" $\mathcal{S}_n$. Accordingly, a natural choice for the target space of Safe BO is $\mathcal{A}_n$ since it captures both exploration and expansion *simultaneously*.[7] To prevent constraint violation, the sample space is restricted to the pessimistic safe set $\mathcal{S}_n$. In Safe BO, both the target space and sample space change with each round $n$, and we generalize our theoretical results from Section 3 in Appendix C to this setting.

**Theorem 5.1** (Convergence to safe optimum)**.** *Pick any $\epsilon > 0$, $\delta \in (0, 1)$. Assume that $f^\star$, $g^\star$ lie in the reproducing kernel Hilbert space $\mathcal{H}_k(\mathcal{X})$ of the kernel $k$, and that the noise $\varepsilon_n$ is conditionally $\rho$-sub-Gaussian. Then, we have with probability at least $1 - \delta$,*

*Safety: for all $n \geq 1$,   $\boldsymbol{x}_n \in \mathcal{S}^\star$.*

*Moreover, assume $\mathcal{S}_0 \neq \emptyset$ and denote with $\mathcal{R}$ the largest reachable safe set starting from $\mathcal{S}_0$. Then, the convergence of reducible uncertainty implies that there exists $n^\star > 0$ such that with probability at least $1 - \delta$,*

*Optimality: for all $n \geq n^\star$,   $r_n(\mathcal{R}) \leq \epsilon$.*

We provide a formal proof in Appendix C.8. Central to the proof is the application of Theorem 3.3 to show that the safety of parameters *outside* the safe set $\mathcal{S}_n$ can be inferred efficiently. In Section 3, we outline settings where the reducible uncertainty converges which is the case for a very general class of functions, and for such instances Theorem 5.1 guarantees optimality in the largest reachable safe set $\mathcal{R}$. $\mathcal{R}$ represents the largest set any safe learning algorithm can explore without violating the safety constraints (with high probability) during learning (cf. Definition C.29). Our guarantees are similar to those of other Safe BO algorithms (Berkenkamp et al., 2021) but require fewer assumptions and generalize to continuous domains. We obtain Theorem 5.1 from a more general result (Theorem C.34) which can be specialized to yield "free" novel convergence guarantees for problems other than Bayesian optimization, such as level set estimation, by choosing an appropriate target space.

---

[7] An alternative possibility is to weigh each point in $\mathcal{A}_n$ according to how likely it is to be the safe optimum. Which approach performs better is task-dependent, and we include a detailed discussion in Appendix K.1.

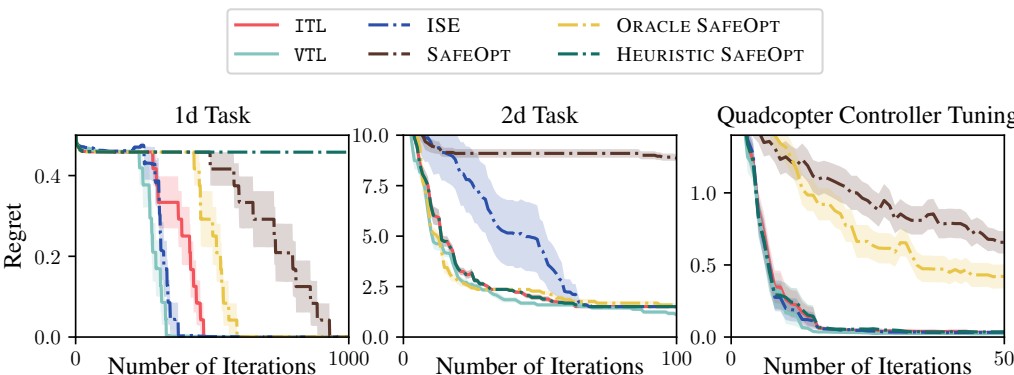

Figure 5: We compare ITL and VTL to ORACLE SAFEOPT, which has oracle knowledge of the Lipschitz constants, SAFEOPT, where the Lipschitz constants are estimated from the GP, as well as HEURISTIC SAFEOPT and ISE, and observe that ITL and VTL systematically perform well. We compare against additional baselines in Appendix K.1. The regret is evaluated with respect to the ground truth objective $f^\star$ and constraint $g^\star$, and averaged over 10 (in synthetic experiments) and 25 (in the quadcopter experiment) random seeds. Additional details can be found in Appendix K.4.

## 5.1 Experiments on Safe Bayesian Optimization

We evaluate two synthetic experiments for a 1d and 2d parameter space, respectively (cf. Appendix K.4 for details), which demonstrate the various shortcomings of existing Safe BO baselines. Additionally, as third experiment, we safely tune the controller of a quadcopter.

**Safe controller tuning for a quadcopter** We consider a quadcopter with unknown dynamics; $s_{t+1} = T(s_t, u_t)$ where $u_t \in \mathbb{R}^{d_u}$ is the control signal and $s_t \in \mathbb{R}^{d_s}$ is the state at time $t$. The inputs $u_t$ are calculated through a deterministic function of the state $\pi : \mathcal{S} \to \mathcal{U}$ which we call the policy. The policy is parameterized via parameters $x \in \mathcal{X}$, e.g., PID controller gains, such that $u_t = \pi_x(s_t)$. The goal is to find the optimal parameters with respect to an unknown objective $f^\star$ while satisfying some unknown constraint(s) $g^\star(x) \geq 0$, e.g., the quadcopter does not fall on the ground. This is a typical Safe BO problem which is widely applied for safe controller learning in robotics (Berkenkamp et al., 2021; Baumann et al., 2021; Widmer et al., 2023).

**Results** We compare ITL and VTL to SAFEOPT (Berkenkamp et al., 2021), which is undirected, i.e., expands in all directions including ones that are known-to-be suboptimal, and ISE (Bottero et al., 2022), which is solely expansionist — does not trade-off expansion-exploration. We provide a detailed discussion of baselines in Appendix K.2. In all our experiments, summarized in Figure 5, we observe that ITL and VTL systematically perform well, i.e., better or on par with the state-of-the-art. We attribute this to its directed exploration and less conservative expansion over SAFEOPT (cf. 1d task and quadcopter experiment), and natural trade-off between expansion and exploration as opposed to ISE (see 2d task). Generally, VTL has a slight advantage over ITL, which is because VTL minimizes marginal variances (as opposed to entropy), which are decisive for expanding the safe set. While ITL and VTL do not violate constraints, we observe that other methods that do not explicitly enforce safety such as EIC (Gardner et al., 2014) lead to constraint violation (cf. Appendix K.4.2).

## 6 Related Work

**(Inductive) active learning** The special case of transductive active learning where $\mathcal{A} = \mathcal{S} = \mathcal{X}$ has been widely studied. We refer to this special instance as *inductive* active learning, since the goal is to extract as much information as possible as opposed to making predictions on a specific target set.

Several works have previously found entropy-based decision rules to be useful for inductive active learning (Krause & Guestrin, 2007; Guo & Greiner, 2007; Krause et al., 2008) and semi-supervised learning (Grandvalet & Bengio, 2004). The variance-based VTL has previously been proposed by Cohn (1993) in the special case of inductive active learning without proving theoretical guarantees. VTL was then recently re-derived by Shoham & Avron (2023) along other experimental design

criteria under the lens of minimizing risk for inductive one-shot learning in overparameterized models. Substantial work on active learning has studied entropy-based criteria in *parameter-space*, most notably BALD (MacKay, 1992; Houlsby et al., 2011; Gal et al., 2017; Kirsch et al., 2019), which selects $\boldsymbol{x}_n = \arg\max_{\boldsymbol{x} \in \mathcal{X}} \mathrm{I}(\boldsymbol{\theta}; y_{\boldsymbol{x}} \mid \mathcal{D}_{n-1})$, where $\boldsymbol{\theta}$ is the random parameter vector of a parametric model (e.g., obtained via Bayesian deep learning). Such methods are inherently inductive in the sense that they do not facilitate learning on specific prediction targets.

**Transductive active learning** In contrast, ITL operates in *output-space* where it is straightforward to specify prediction targets, and which is computationally easier. Special cases of ITL when $\mathcal{S} = \mathcal{X}$ and $|\mathcal{A}| = 1$ have been proposed in the foundational work of MacKay (1992) on "directed" output-space active learning. As generalization to larger target spaces, MacKay (1992) proposed mean-marginal ITL,

$$\boldsymbol{x}_n = \arg\max_{\boldsymbol{x} \in \mathcal{S}} \sum_{\boldsymbol{x}' \in \mathcal{A}} \mathrm{I}(f_{\boldsymbol{x}'}; y_{\boldsymbol{x}} \mid \mathcal{D}_{n-1}), \tag{MM-ITL}$$

for which we derive analogous versions of Theorems 3.2 and 3.3 in Appendix D.3. We note that similarly to VTL, MM-ITL disregards the mutual dependence of points in the target space $\mathcal{A}$ and differs from VTL only in a different weighting of the posterior marginal variances of the prediction targets (cf. Appendix D.3). Recently, Bickford Smith et al. (2023) generalized MM-ITL by treating the prediction target as a random variable, and Kothawade et al. (2021) and Bickford Smith et al. (2024) demonstrated the use of output-space decision rules for image classification tasks in a pre-training context.

**Influence functions** measure the change in a model's prediction when a single data point is removed from the training data (Cook, 1977; Koh & Liang, 2017; Pruthi et al., 2019). Influence functions have been used for data selection in settings closely related to the transductive active fine-tuning of neural networks proposed in this work (Xia et al., 2024). They select data that reduces a first-order Taylor approximation to the test loss after fine-tuning a neural network, which corresponds to maximizing cosine similarity to the prediction targets in a loss-gradient embedding space. We show in our experiments that transductive active learning can substantially outperform COSINESIMILARITY. We attribute this primarily to influence functions implicitly assuming that the influence of selected data adds linearly (i.e., two equally scored data points are expected to doubly improve the model performance, Xu & Kazantsev, 2019, Section 3.2). This assumption does not hold in practice as seen, e.g., by simply duplicating data. The same limitation applies to the related approach of datamodels (Ilyas et al., 2022).

**Other work on directed active learning** Directed active learning methods have been proposed for the problem of determining the optimum of an unknown function, also known as best-arm identification (Audibert et al., 2010) or pure exploration bandits (Bubeck et al., 2009). Entropy search methods (Hennig & Schuler, 2012; Hernández-Lobato et al., 2014) are widely used and select $\boldsymbol{x}_n = \arg\max_{\boldsymbol{x} \in \mathcal{X}} \mathrm{I}(\boldsymbol{x}^*; y_{\boldsymbol{x}} \mid \mathcal{D}_{n-1})$ in *input-space* where $\boldsymbol{x}^* = \arg\max_{\boldsymbol{x}} f_{\boldsymbol{x}}$. Similarly to ITL, *output-space* entropy search methods (Hoffman & Ghahramani, 2015; Wang & Jegelka, 2017), which select $\boldsymbol{x}_n = \arg\max_{\boldsymbol{x} \in \mathcal{X}} \mathrm{I}(f^*; y_{\boldsymbol{x}} \mid \mathcal{D}_{n-1})$ with $f^* = \max_{\boldsymbol{x}} f_{\boldsymbol{x}}$, are more computationally tractable. In fact, output-space entropy search is a special case of ITL with a stochastic target space (cf. Equation (47) in Appendix K.1). Bogunovic et al. (2016) analyze TRUVAR in the context of Bayesian optimization and level set estimation. TRUVAR is akin to VTL with a similar notion of "target space", but their algorithm and analysis rely on a threshold scheme which requires that $\mathcal{A} \subseteq \mathcal{S}$. Fiez et al. (2019) introduce the *transductive linear bandit* problem, which is a special case of transductive active learning limited to a linear function class and with the objective of determining the maximum within an initial candidate set.[8] We mention additional more loosely related works in Appendix A.

# 7 Conclusion

We investigated the generalization of active learning to settings with concrete prediction targets and/or with limited information due to constrained sample spaces. This provides a flexible framework, applicable also to other domains than were discussed (such as recommender systems, molecular design, robotics, etc.) by varying the choice of target space and sample space. Further, we proved novel generalization bounds which may be of independent interest for active learning. Finally, we demonstrated across broad applications that sampling *relevant and diverse* points (as opposed to only one of the two) leads to a substantial improvement upon the state-of-the-art.

---

[8]The transductive bandit problem can be solved analogously to Safe BO, by maintaining the set $\mathcal{A}_n$.

## Acknowledgements

Many thanks to Armin Lederer, Johannes Kirschner, Jonas Rothfuss, Lars Lorch, Manish Prajapat, Nicolas Emmenegger, Parnian Kassraie, and Scott Sussex for their insightful feedback on different versions of this manuscript, as well as Anton Baumann for helpful discussions. We further thank Freddie Bickford Smith for a constructive discussion regarding the relationship between our work and prior work.

This project was supported in part by the European Research Council (ERC) under the European Union's Horizon 2020 research and Innovation Program Grant agreement no. 815943, the Swiss National Science Foundation under NCCR Automation, grant agreement 51NF40 180545, and by a grant of the Hasler foundation (grant no. 21039).

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

# Appendices

A general principle of "transductive learning" was already formulated by the famous computer scientist Vladimir Vapnik in the 20th century. Vapnik proposes the following "imperative for a complex world":

*When solving a problem of interest, do not solve a more general problem as an intermediate step. Try to get the answer that you really need but not a more general one.*

– Vapnik (1982)

These appendices provide additional background, proofs, experiment details, and ablation studies.

## Contents

# A  Additional Related Work

The general principle of non-active "transductive learning" was introduced by Vapnik (1982). The notion of "target" from transductive active learning is akin to the notion of "task" in curriculum learning (Bengio et al., 2009; Graves et al., 2017; Soviany et al., 2022). The study of settings where the irreducible uncertainty is zero is related to the study of estimability in experimental design (Graybill, 1961; Mutny & Krause, 2022). In feature selection, selecting features that maximize information gain with respect to a to-be-predicted label is a standard approach (Peng et al., 2005; Vergara & Estévez, 2014; Beraha et al., 2019) which is akin to ITL (cf. Appendix D). The themes of relevance and diversity are also important for efficient in-context learning (e.g., Ye et al., 2023; Kumari et al., 2024) and data pruning (Zheng et al., 2023). Transductive active learning is complimentary to other learning methodologies, such as semi-supervised learning (Gao et al., 2020), self-supervised learning (Shwartz-Ziv & LeCun, 2023; Balestriero et al., 2023), and meta-learning (Kaddour et al., 2020; Rothfuss et al., 2023).

# B  Background

## B.1  Information Theory

Throughout this work, $\log$ denotes the natural logarithm. Given random vectors $\boldsymbol{x}$ and $\boldsymbol{y}$, we denote by

$$\mathrm{H}[\boldsymbol{x}] \overset{\mathrm{def}}{=} \mathbb{E}_{p(\boldsymbol{x})}[-\log p(\boldsymbol{x})],$$
$$\mathrm{H}[\boldsymbol{x} \mid \boldsymbol{y}] \overset{\mathrm{def}}{=} \mathbb{E}_{p(\boldsymbol{x},\boldsymbol{y})}[-\log p(\boldsymbol{x} \mid \boldsymbol{y})], \quad \text{and}$$
$$\mathrm{I}(\boldsymbol{x};\boldsymbol{y}) \overset{\mathrm{def}}{=} \mathrm{H}[\boldsymbol{x}] - \mathrm{H}[\boldsymbol{x} \mid \boldsymbol{y}]$$

the (differential) entropy, conditional entropy, and information gain, respectively (Cover, 1999).[9]

The *multivariate information gain* (Murphy, 2023) between random vectors $\boldsymbol{x}, \boldsymbol{y}, \boldsymbol{z}$ is given by

$$\mathrm{I}(\boldsymbol{x};\boldsymbol{y};\boldsymbol{z}) \overset{\mathrm{def}}{=} \mathrm{I}(\boldsymbol{x};\boldsymbol{y}) - \mathrm{I}(\boldsymbol{x};\boldsymbol{y} \mid \boldsymbol{z}) \tag{6}$$
$$= \mathrm{I}(\boldsymbol{x};\boldsymbol{y}) + \mathrm{I}(\boldsymbol{x};\boldsymbol{z}) - \mathrm{I}(\boldsymbol{x};\boldsymbol{y},\boldsymbol{z}). \tag{7}$$

When $\mathrm{I}(\boldsymbol{x};\boldsymbol{y};\boldsymbol{z}) \neq 0$ it is said that $\boldsymbol{y}$ and $\boldsymbol{z}$ *interact* regarding their information about $\boldsymbol{x}$. If the interaction is positive, it is said that the information of $\boldsymbol{z}$ about $\boldsymbol{x}$ is *redundant* given $\boldsymbol{y}$. Conversely, if the interaction is negative, it is said that the information of $\boldsymbol{z}$ about $\boldsymbol{x}$ is *synergistic* with $\boldsymbol{y}$. The notion of synergy is akin to the frequentist notion of "suppressor variables" in linear regression (Das & Kempe, 2008).

## B.2  Gaussian Processes

The stochastic process $f$ is a Gaussian process (GP, Williams & Rasmussen (2006)), denoted $f \sim \mathcal{GP}(\mu, k)$, with mean function $\mu$ and kernel $k$ if for any finite subset $X = \{\boldsymbol{x}_1, \ldots, \boldsymbol{x}_n\} \subseteq \mathcal{X}$, $\boldsymbol{f}_X \sim \mathcal{N}(\boldsymbol{\mu}_X, \boldsymbol{K}_{XX})$ is jointly Gaussian with mean vector $\boldsymbol{\mu}_X(i) = \mu(\boldsymbol{x}_i)$ and covariance matrix $\boldsymbol{K}_{XX}(i,j) = k(\boldsymbol{x}_i, \boldsymbol{x}_j)$.

In the following, we formalize the assumptions from the GP setting (cf. Section 3.1).

**Assumption B.1** (Gaussian prior). We assume that $f \sim \mathcal{GP}(\mu, k)$ with known mean function $\mu$ and kernel $k$.

**Assumption B.2** (Gaussian noise). We assume that the noise $\varepsilon_{\boldsymbol{x}}$ is mutually independent and zero-mean Gaussian with known variance $\rho^2(\boldsymbol{x}) > 0$. We write $\boldsymbol{P}_X = \mathrm{diag}\, \rho^2(\boldsymbol{x}_1), \ldots, \rho^2(\boldsymbol{x}_n)$.

Under Assumptions B.1 and B.2, the posterior distribution of $f$ after observing points $X$ is $\mathcal{GP}(\mu_n, k_n)$ with

$$\mu_n(\boldsymbol{x}) = \mu(\boldsymbol{x}) + \boldsymbol{K}_{\boldsymbol{x}X}(\boldsymbol{K}_{XX} + \boldsymbol{P}_X)^{-1}(\boldsymbol{y}_X - \boldsymbol{\mu}_X),$$
$$k_n(\boldsymbol{x}, \boldsymbol{x}') = k(\boldsymbol{x}, \boldsymbol{x}') - \boldsymbol{K}_{\boldsymbol{x}X}(\boldsymbol{K}_{XX} + \boldsymbol{P}_X)^{-1}\boldsymbol{K}_{X\boldsymbol{x}'},$$
$$\sigma_n^2(\boldsymbol{x}) = k_n(\boldsymbol{x}, \boldsymbol{x}).$$

---

[9]One has to be careful to ensure that $\mathrm{I}(\boldsymbol{x};\boldsymbol{y})$ exists, i.e., $|\mathrm{I}(\boldsymbol{x};\boldsymbol{y})| < \infty$. We will assume that this is the case throughout this work. When $\boldsymbol{x}$ and $\boldsymbol{y}$ are jointly Gaussian, this is satisfied when the noise variance $\rho^2$ is positive.

For Gaussian random vectors $\boldsymbol{f}$ and $\boldsymbol{y}$, the entropy is $\mathrm{H}[\boldsymbol{f}] = \frac{n}{2}\log(2\pi e) + \frac{1}{2}\log|\mathrm{Var}[\boldsymbol{f}]|$, the information gain is $\mathrm{I}(\boldsymbol{f};\boldsymbol{y}) = \frac{1}{2}(\log|\mathrm{Var}[\boldsymbol{y}]| - \log|\mathrm{Var}[\boldsymbol{y} \mid \boldsymbol{f}]|)$, and

$$\gamma_n = \max_{\substack{X \subseteq \mathcal{X} \\ |X| \leq n}} \frac{1}{2}\log\left|\boldsymbol{I} + \boldsymbol{P}_X^{-1}\boldsymbol{K}_{XX}\right|.$$

## C  Proofs

We will write

- $\sigma^2 \stackrel{\mathrm{def}}{=} \max_{\boldsymbol{x} \in \mathcal{X}} \sigma_0^2(\boldsymbol{x})$, and
- $\tilde{\sigma}^2 \stackrel{\mathrm{def}}{=} \max_{\boldsymbol{x} \in \mathcal{X}} \sigma_0^2(\boldsymbol{x}) + \rho^2(\boldsymbol{x})$.

The following is a brief overview of the structure of this section:

1. Appendix C.1 relates ITL in the inductive learning setting ($\mathcal{S} \subseteq \mathcal{A}$) to prior work.
2. Appendix C.2 relates the designs selected by ITL and VTL to the optimal designs for corresponding non-adaptive objectives.
3. Appendix C.3 shows that batch selection via ITL or VTL leads to informative and diverse batches, utilizing the results from Appendix C.2.
4. Appendix C.4 introduces measures of synergies that generalize the submodularity assumption (cf. Assumption 3.1).
5. Appendix C.5 proves key results on the convergence of the ITL and VTL objectives.
6. Appendix C.6 proves Theorem 3.2 (convergence in GP setting).
7. Appendix C.7 proves Theorem 3.3 (convergence in agnostic setting).
8. Appendix C.8 proves Theorem 5.1 (convergence in safe BO application).
9. Appendix C.9 includes useful facts.

### C.1  Undirected Case of ITL

We briefly examine the important special case of ITL where $\mathcal{S} \subseteq \mathcal{A}$. In this setting, for all $\boldsymbol{x} \in \mathcal{S}$, the decision rule of ITL simplifies to

$$\mathrm{I}(\boldsymbol{f}_{\mathcal{A}}; y_{\boldsymbol{x}} \mid \mathcal{D}_n) \stackrel{(i)}{=} \mathrm{I}(\boldsymbol{f}_{\mathcal{A}\setminus\{\boldsymbol{x}\}}; y_{\boldsymbol{x}} \mid f_{\boldsymbol{x}}, \mathcal{D}_n) + \mathrm{I}(f_{\boldsymbol{x}}; y_{\boldsymbol{x}} \mid \mathcal{D}_n)$$
$$\stackrel{(ii)}{=} \mathrm{I}(f_{\boldsymbol{x}}; y_{\boldsymbol{x}} \mid \mathcal{D}_n)$$
$$= \mathrm{H}[y_{\boldsymbol{x}} \mid \mathcal{D}_n] - \mathrm{H}[\varepsilon_{\boldsymbol{x}}]$$

where $(i)$ follows from the chain rule of information gain and $\boldsymbol{x} \in \mathcal{S} \subseteq \mathcal{A}$; and $(ii)$ follows from the conditional independence $\boldsymbol{f}_{\mathcal{A}} \perp y_{\boldsymbol{x}} \mid f_{\boldsymbol{x}}$.

If additionally $f$ is a GP then

$$\mathrm{H}[y_{\boldsymbol{x}} \mid \mathcal{D}_n] - \mathrm{H}[\varepsilon_{\boldsymbol{x}}] = \frac{1}{2}\log\left(1 + \frac{\mathrm{Var}[f_{\boldsymbol{x}} \mid \mathcal{D}_n]}{\mathrm{Var}[\varepsilon_{\boldsymbol{x}}]}\right).$$

This decision rule has also been termed *total information gain* (MacKay, 1992). When $\mathcal{S} \subseteq \mathcal{A}$ and observation noise is homoscedastic, this decision rule is equivalent to uncertainty sampling.

### C.2  Non-adaptive Data Selection & Submodularity

Recall the non-myopic information gain $\psi_{\mathcal{A}}(X) = \mathrm{I}(\boldsymbol{f}_{\mathcal{A}}; \boldsymbol{y}_X)$ (ITL) and variance reduction $\psi_{\mathcal{A}}(X) = \mathrm{tr}\,\mathrm{Var}[\boldsymbol{f}_{\mathcal{A}}] - \mathrm{tr}\,\mathrm{Var}[\boldsymbol{f}_{\mathcal{A}} \mid \boldsymbol{y}_X]$ (VTL) objective functions from Assumption 3.1. In this section, we will relate the designs selected by ITL and VTL to the optimal designs for these objectives. To this end, consider the non-adaptive optimization problem

$$X^{\star} = \arg\max_{\substack{X \subseteq \mathcal{S} \\ |X|=k}} \psi_{\mathcal{A}}(X).$$

**Lemma C.1.** *For both* ITL *and* VTL, $\psi_{\mathcal{A}}$ *is non-negative and monotone.*

*Proof.* For ITL, $\psi_{\mathcal{A}}(X) \geq 0$ follows from the non-negativity of mutual information. To conclude monotonicity, note that for any $X' \subseteq X \subseteq \mathcal{S}$,

$$\mathrm{I}(\boldsymbol{f}_{\mathcal{A}}; \boldsymbol{y}_{X'}) = \mathrm{H}[\boldsymbol{f}_{\mathcal{A}}] - \mathrm{H}[\boldsymbol{f}_{\mathcal{A}} \mid \boldsymbol{y}_{X'}] \leq \mathrm{H}[\boldsymbol{f}_{\mathcal{A}}] - \mathrm{H}[\boldsymbol{f}_{\mathcal{A}} \mid \boldsymbol{y}_X] = \mathrm{I}(\boldsymbol{f}_{\mathcal{A}}; \boldsymbol{y}_X)$$

due to monotonicity of conditional entropy (which is also called the "information never hurts" principle).

For VTL, recall that $\mathrm{tr}\,\mathrm{Var}[\boldsymbol{f}_{\mathcal{A}} \mid \boldsymbol{y}_X] \leq \mathrm{tr}\,\mathrm{Var}[\boldsymbol{f}_{\mathcal{A}} \mid \boldsymbol{y}_{X'}]$ for any $X' \subseteq X \subseteq \mathcal{S}$ (with an implicit expectation over $\boldsymbol{y}_X, \boldsymbol{y}_{X'}$). Non-negativity and monotonicity of $\psi_{\mathcal{A}}$ then follow analogously to ITL. $\quad\square$

**Lemma C.2.** *The marginal gain* $\Delta_{\mathcal{A}}(\boldsymbol{x} \mid X) \overset{\text{def}}{=} \psi_{\mathcal{A}}(X \cup \{\boldsymbol{x}\}) - \psi_{\mathcal{A}}(X)$ *of* $\boldsymbol{x} \in \mathcal{S}$ *given* $X \subseteq \mathcal{S}$ *is the* ITL *and* VTL *objective, respectively.*

*Proof.* For ITL,

$$\begin{aligned}
\Delta_{\mathcal{A}}(\boldsymbol{x} \mid X) &= \mathrm{I}(\boldsymbol{f}_{\mathcal{A}}; \boldsymbol{y}_X, y_{\boldsymbol{x}}) - \mathrm{I}(\boldsymbol{f}_{\mathcal{A}}; \boldsymbol{y}_X) \\
&= \mathrm{H}[\boldsymbol{f}_{\mathcal{A}} \mid \boldsymbol{y}_X] - \mathrm{H}[\boldsymbol{f}_{\mathcal{A}} \mid \boldsymbol{y}_X, y_{\boldsymbol{x}}] \\
&= \mathrm{I}(\boldsymbol{f}_{\mathcal{A}}; y_{\boldsymbol{x}} \mid \boldsymbol{y}_X)
\end{aligned}$$

which is precisely the ITL objective.

For VTL,

$$\begin{aligned}
\Delta_{\mathcal{A}}(\boldsymbol{x} \mid X) &= \mathrm{tr}\,\mathrm{Var}[\boldsymbol{f}_{\mathcal{A}} \mid \boldsymbol{y}_X] - \mathrm{tr}\,\mathrm{Var}[\boldsymbol{f}_{\mathcal{A}} \mid \boldsymbol{y}_X, y_{\boldsymbol{x}}] \\
&= -\mathrm{tr}\,\mathrm{Var}[\boldsymbol{f}_{\mathcal{A}} \mid \boldsymbol{y}_X, y_{\boldsymbol{x}}] + \mathrm{const}
\end{aligned}$$

which is precisely the VTL objective. $\quad\square$

**Definition C.3** (Submodularity). $\psi_{\mathcal{A}}$ *is submodular if and only if for all* $\boldsymbol{x} \in \mathcal{S}$ *and* $X' \subseteq X \subseteq \mathcal{S}$,

$$\Delta_{\mathcal{A}}(\boldsymbol{x} \mid X') \geq \Delta_{\mathcal{A}}(\boldsymbol{x} \mid X).$$

**Theorem C.4** (Nemhauser et al. (1978)). *Let Assumption 3.1 hold. For any* $n \geq 1$, *if* ITL *or* VTL *selected* $\boldsymbol{x}_{1:n}$, *respectively, then*

$$\psi_{\mathcal{A}}(\boldsymbol{x}_{1:n}) \geq (1 - 1/e) \max_{\substack{X \subseteq \mathcal{S} \\ |X| \leq n}} \psi_{\mathcal{A}}(X).$$

*Proof.* This is a special case of a canonical result from non-negative monotone submodular function maximization (Nemhauser et al., 1978; Krause & Golovin, 2014). $\quad\square$

## C.3 Batch Diversity: Batch Selection as Non-adaptive Data Selection

Recall the non-adaptive optimization problem

$$B_{n,k} = \underset{\substack{B \subseteq \mathcal{S} \\ |B| = k}}{\arg\max}\, \mathrm{I}(\boldsymbol{f}_{\mathcal{A}}; \boldsymbol{y}_B \mid \mathcal{D}_{n-1})$$

from Equation (3) with batch size $k > 0$, and denote by $B'_{n,k} = \boldsymbol{x}_{n,1:k}$ the greedy approximation from Equation (3). The selection of an individual batch can be seen as a single non-adaptive optimization problem with marginal gain

$$\begin{aligned}
\Delta_n(\boldsymbol{x} \mid B) &= \mathrm{I}(\boldsymbol{f}_{\mathcal{A}}; \boldsymbol{y}_B, y_{\boldsymbol{x}} \mid \mathcal{D}_{n-1}) - \mathrm{I}(\boldsymbol{f}_{\mathcal{A}}; \boldsymbol{y}_B \mid \mathcal{D}_{n-1}) \\
&= \mathrm{H}[\boldsymbol{f}_{\mathcal{A}} \mid \mathcal{D}_{n-1}, \boldsymbol{y}_B] - \mathrm{H}[\boldsymbol{f}_{\mathcal{A}} \mid \mathcal{D}_{n-1}, \boldsymbol{y}_B, y_{\boldsymbol{x}}] \\
&= \mathrm{I}(\boldsymbol{f}_{\mathcal{A}}; y_{\boldsymbol{x}} \mid \mathcal{D}_{n-1}, \boldsymbol{y}_B)
\end{aligned}$$

and which is precisely the objective function of ITL from Equation (3). Hence, the approximation guarantees from Theorems C.4 and C.11 apply. The derivation is analogous for VTL.

Prior work has shown that the greedy solution $B'_n$ is also competitive with a fully sequential "batchless" decision rule (Chen & Krause, 2013; Esfandiari et al., 2021).

## C.4 Measures of Synergies & Approximate Submodularity

We will now show that "downstream synergies", if present, can be seen as a source of learning complexity, which is orthogonal to the information capacity $\gamma_n$.

**Example C.5.** Consider the example where $f$ is a stochastic process of three random variables $X, Y, Z$ where $X$ and $Y$ are Bernoulli ($p = \frac{1}{2}$), and $Z$ is the XOR of $X$ and $Y$. Suppose that observations are exact (i.e., $\varepsilon_n = 0$), that the target space $\mathcal{A}$ comprises the output variable $Z$ while the sample space $\mathcal{S}$ comprises the input variables $X$ and $Y$. Observing any single $X$ or $Y$ yields no information about $Z$: $\mathrm{I}(Z; X) = \mathrm{I}(Z; Y) = 0$, however, observing both inputs jointly perfectly determines $Z$: $\mathrm{I}(Z; X, Y) = 1$. Thus, $\gamma_n(\mathcal{A}; \mathcal{S}) = 1$ if $n \geq 2$ and $\gamma_n(\mathcal{A}; \mathcal{S}) = 0$ else.

Learning about $Z$ in examples of this kind is difficult for agents that make decisions greedily, since the next action (observing $X$ or $Y$) yields no signal about its long-term usefulness. We call a sequence of observations, such as $\{X, Y\}$, *synergistic* since its combined information value is larger than the individual values. The prevalence of synergies is not captured by the information capacity $\gamma_n(\mathcal{A}; \mathcal{S})$ since it measures only the joint information gain of $n$ samples within $\mathcal{S}$. Instead, the prevalence of synergies is captured by the sequence $\Gamma_n \stackrel{\text{def}}{=} \max_{\boldsymbol{x} \in \mathcal{S}} \Delta_{\mathcal{A}}(\boldsymbol{x} \mid \boldsymbol{x}_{1:n})$, which measures the maximum information gain of $y_{n+1}$. If $\Gamma_n > \Gamma_{n-1}$ at any round $n$, this indicates a synergy. The following key object measures the additional complexity due to synergies.

**Definition C.6** (Task complexity). For $n \geq 1$, assuming $\Gamma_i > 0$ for all $1 \leq i \leq n$, we define the *task complexity* as

$$\alpha_{\mathcal{A}, \mathcal{S}}(n) \stackrel{\text{def}}{=} \max_{i \in \{0, \ldots, n-1\}} \frac{\Gamma_{n-1}}{\Gamma_i}.$$

Note that $\alpha_{\mathcal{A}, \mathcal{S}}(n)$ is large only if the information gain of $y_n$ is larger than that of a previous observation $y_i$. Intuitively, if $\alpha_{\mathcal{A}, \mathcal{S}}(n)$ is large, the agent had to discover the *implicit* intermediate observations $y_1, \ldots, y_{n-1}$ that lead to downstream synergies. We will subsequently formalize the intimate connections of the task complexity to synergies and submodularity. Note that in the GP setting, $\alpha_{\mathcal{A}, \mathcal{S}}(n)$ can be computed online by keeping track of the smallest $\Gamma_i$ during previous rounds $i$. Further, note that $\alpha_{\mathcal{A}, \mathcal{S}}(n) \leq 1$ if $\psi_{\mathcal{A}}$ is submodular.

### C.4.1 The Information Ratio

Another object will prove useful in our analysis of synergies.

Consider an alternative multiplicative interpretation of the multivariate information gain (cf. Equation (7)), which we call the *information ratio* of $X \subseteq \mathcal{S}$ given $D \subseteq \mathcal{S}$, $|X|, |D| < \infty$:

$$\bar{\kappa}(X \mid D) \stackrel{\text{def}}{=} \frac{\sum_{\boldsymbol{x} \in X} \Delta_{\mathcal{A}}(\boldsymbol{x} \mid D)}{\Delta_{\mathcal{A}}(X \mid D)} \in [0, \infty). \tag{8}$$

Observe that $\bar{\kappa}(X \mid D)$ measures the synergy properties of $\boldsymbol{y}_X$ with respect to $\boldsymbol{f}_{\mathcal{A}}$ given $\boldsymbol{y}_D$ in a multiplicative sense. That is, if $\bar{\kappa}(X \mid D) > 1$ then information in $\boldsymbol{y}_X$ is redundant, whereas if $\bar{\kappa}(X \mid D) < 1$ then information in $\boldsymbol{y}_X$ is synergistic, and if $\bar{\kappa}(X \mid D) = 1$ then $\boldsymbol{y}_X$ do not mutually interact with respect to $\boldsymbol{f}_{\mathcal{A}}$ (all given $\boldsymbol{y}_D$). In the degenerate case where $\Delta_{\mathcal{A}}(X \mid D) = 0$ (which implies $\sum_{\boldsymbol{x} \in X} \Delta_{\mathcal{A}}(\boldsymbol{x} \mid D) = 0$) we therefore let $\bar{\kappa}(X \mid D) = 1$.

**The information ratio of ITL is strictly positive in the Gaussian case**    We prove the following straightforward lower bound to the information ratio of ITL.

**Lemma C.7.** *Let $X, D \subseteq \mathcal{S}, |X|, |D| < \infty$. If $\boldsymbol{f}_{\mathcal{A}}$ and $\boldsymbol{y}_{X \cup D}$ are jointly Gaussian then $\bar{\kappa}(X \mid D) > 0$.*

*Proof.* W.l.o.g. assume $D = \emptyset$. We let $X = \{\boldsymbol{x}_1, \ldots, \boldsymbol{x}_k\}$ and prove lower and upper bound separately. We assume w.l.o.g. that $\mathrm{I}(\boldsymbol{f}_{\mathcal{A}}; \boldsymbol{y}_X) > 0$ which implies $|\mathrm{Var}[\boldsymbol{f}_{\mathcal{A}} \mid \boldsymbol{y}_X]| < |\mathrm{Var}[\boldsymbol{f}_{\mathcal{A}}]|$. Thus, there exists some $i$ such that $\boldsymbol{f}_{\mathcal{A}}$ and $y_{\boldsymbol{x}_i}$ are dependent, so $|\mathrm{Var}[\boldsymbol{f}_{\mathcal{A}} \mid y_{\boldsymbol{x}_i}]| < |\mathrm{Var}[\boldsymbol{f}_{\mathcal{A}}]|$ which implies $\mathrm{I}(\boldsymbol{f}_{\mathcal{A}}; y_{\boldsymbol{x}_i}) > 0$. We therefore conclude that $\bar{\kappa}(X) > 0$. $\qquad \square$

The following example shows that this lower bound is tight.

**Example C.8** (Synergies of Gaussian random variables, inspired by Section 3 of Barrett (2015))**.**
Consider the three random variables $X$, $Y$, and $Z$ (think $\mathcal{A} = \{X\}$ and $\mathcal{S} = \{Y, Z\}$) which are
jointly Gaussian with mean vector $\mathbf{0}$ and covariance matrix

$$\boldsymbol{\Sigma} = \begin{bmatrix} 1 & a & a \\ a & 1 & 0 \\ a & 0 & 1 \end{bmatrix}, \qquad \text{for } 2a^2 < 1$$

where the constraint on $a$ is to ensure that $\boldsymbol{\Sigma}$ is positive definite. Computing the mutual information,
we have

$$\mathrm{I}(X;Y) = \mathrm{I}(X;Z) = -\frac{1}{2}\log(1 - a^2)$$

and $\mathrm{I}(X;Y,Z) = -\frac{1}{2}\log(1 - 2a^2)$. Therefore,

$$\frac{\mathrm{I}(X;Y) + \mathrm{I}(X;Z)}{\mathrm{I}(X;Y,Z)} = \frac{\log(1 - 2a^2 + a^4)}{\log(1 - 2a^2)} < 1.$$

Note that

$$\lim_{a \to \frac{1}{\sqrt{2}}} \frac{\log(1 - 2a^2 + a^4)}{\log(1 - 2a^2)} = 0,$$

and hence — perhaps unintuitively — even if $Y$ and $Z$ are uncorrelated, their information about
$X$ may be arbitrarily synergistic.

### C.4.2 The Submodularity of the Special "Undirected" Case of ITL

In the inductive active learning problem considered in most prior works, where $\mathcal{S} \subseteq \mathcal{A}$ and $f$ is a
Gaussian process, it holds for ITL that $\alpha_{\mathcal{A},\mathcal{S}}(n) = 1$ since all learning targets appear *explicitly* in $\mathcal{S}$:

**Lemma C.9.** *Let $\mathcal{S} \subseteq \mathcal{A}$. Then $\psi_{\mathcal{A}}$ of ITL is submodular.*

*Proof.* Fix any $\boldsymbol{x} \in \mathcal{S}$ and $X' \subseteq X \subseteq \mathcal{S}$. Let $\bar{X} \stackrel{\mathrm{def}}{=} X \setminus X'$. By the definition of conditional
information gain, we have

$$\Delta_{\mathcal{A}}(\boldsymbol{x} \mid X) = \mathrm{I}(y_{\boldsymbol{x}}; \boldsymbol{f}_{\mathcal{A}} \mid \boldsymbol{y}_X) = \mathrm{I}(y_{\boldsymbol{x}}; \boldsymbol{f}_{\mathcal{A}}, \boldsymbol{y}_{X'} \mid \boldsymbol{y}_{\bar{X}}) - \mathrm{I}(y_{\boldsymbol{x}}; \boldsymbol{y}_{X'} \mid \boldsymbol{y}_{\bar{X}}).$$

Since for any $\boldsymbol{x} \in \mathcal{S}$ and $X \subseteq \mathcal{S}$, $y_{\boldsymbol{x}} \perp \boldsymbol{y}_X \mid \boldsymbol{f}_{\mathcal{A}}$, this simplifies to

$$\mathrm{I}(y_{\boldsymbol{x}}; \boldsymbol{f}_{\mathcal{A}} \mid \boldsymbol{y}_X) = \mathrm{I}(y_{\boldsymbol{x}}; \boldsymbol{f}_{\mathcal{A}} \mid \boldsymbol{y}_{\bar{X}}) - \mathrm{I}(y_{\boldsymbol{x}}; \boldsymbol{y}_{X'} \mid \boldsymbol{y}_{\bar{X}}).$$

It then follows from $\mathrm{I}(y_{\boldsymbol{x}}; \boldsymbol{y}_{X'} \mid \boldsymbol{y}_{\bar{X}}) \geq 0$ that

$$\Delta_{\mathcal{A}}(\boldsymbol{x} \mid X) = \mathrm{I}(y_{\boldsymbol{x}}; \boldsymbol{f}_{\mathcal{A}} \mid \boldsymbol{y}_X) \leq \mathrm{I}(y_{\boldsymbol{x}}; \boldsymbol{f}_{\mathcal{A}} \mid \boldsymbol{y}_{\bar{X}}) = \Delta_{\mathcal{A}}(\boldsymbol{x} \mid X').$$

$\square$

This implies that $\alpha_{\mathcal{A},\mathcal{S}}(n) \leq 1$ for any $n$ and $\bar{\kappa}(X \mid D) \geq 1$ for any $X, D \subseteq \mathcal{S}$ when $\mathcal{S} \subseteq \mathcal{A}$.

### C.4.3 The Submodularity Ratio

Building upon the theory of maximizing non-negative monotone submodular functions (Nemhauser
et al., 1978; Krause & Golovin, 2014), Das & Kempe (2018) define the following notion of
"approximate" submodularity:

**Definition C.10** (Submodularity ratio)**.** The *submodularity ratio* of $\psi_{\mathcal{A}}$ up to cardinality $n \geq 1$ is

$$\kappa_{\mathcal{A}}(n) \stackrel{\mathrm{def}}{=} \min_{\substack{D \subseteq \boldsymbol{x}_{1:n} \\ X \subseteq \mathcal{S}: |X| \leq n \\ D \cap X = \emptyset}} \bar{\kappa}(X \mid D), \tag{9}$$

where they define $\frac{0}{0} \equiv 1$. $\psi_{\mathcal{A}}$ is said to be $\kappa$-*weakly submodular* for some $\kappa > 0$ if $\inf_{n \in \mathbb{N}} \kappa_{\mathcal{A}}(n) \geq \kappa$.

As a special case of Theorem 6 from Das & Kempe (2018), applying that $\psi_{\mathcal{A}}$ is non-negative and
monotone, we obtain the following result.

**Theorem C.11** (Das & Kempe (2018))**.** *For any $n \geq 1$, if ITL or VTL selected $\boldsymbol{x}_{1:n}$, respectively,
then*

$$\psi_{\mathcal{A}}(\boldsymbol{x}_{1:n}) \geq (1 - e^{-\kappa_{\mathcal{A}}(n)}) \max_{\substack{X \subseteq \mathcal{S} \\ |X| \leq n}} \psi_{\mathcal{A}}(X).$$

If $\psi_{\mathcal{A}}$ is submodular, it is implied that $\kappa_{\mathcal{A}}(n) \geq 1$ for all $n \geq 1$ in which case Theorem C.11 recovers
Theorem C.4.

## C.5 Convergence of Marginal Gain

Our following analysis allows for changing target spaces $\mathcal{A}_n$ and sample spaces $\mathcal{S}_n$ (cf. Section 5), and to this end, we redefine $\Gamma_n \overset{\text{def}}{=} \max_{\boldsymbol{x} \in \mathcal{S}_n} \Delta_{\mathcal{A}_n}(\boldsymbol{x} \mid \boldsymbol{x}_{1:n})$. The following theorems show that the marginal gains of ITL and VTL converge to zero, and will serve as the main tool for establishing Theorems 3.2 and 3.3. We will abbreviate $\alpha_{\mathcal{A},\mathcal{S}}(n)$ by $\alpha_n$.

**Theorem C.12** (Convergence of Marginal Gain for ITL). *Assume that Assumptions B.1 and B.2 are satisfied. Fix any integers $n_1 > n_0 \geq 0$, $\Delta = n_1 - n_0 + 1$ such that for all $i \in \{n_0, \ldots, n_1 - 1\}$, $\mathcal{A}_{i+1} \subseteq \mathcal{A}_i$ and $\mathcal{S} \overset{\text{def}}{=} \mathcal{S}_{i+1} = \mathcal{S}_i$. Further, assume $|\mathcal{A}_{n_0}| < \infty$. Then, if the sequence $\{\boldsymbol{x}_{i+1}\}_{i=n_0}^{n_1}$ was generated by* ITL,*

$$\Gamma_{n_1} \leq \alpha_{n_1} \frac{\gamma_\Delta}{\Delta}. \tag{10}$$

*Moreover, if $n_0 = 0$,*

$$\Gamma_{n_1} \leq \alpha_{n_1} \frac{\gamma_{\mathcal{A}_0,\mathcal{S}}(\Delta)}{\Delta}. \tag{11}$$

*Proof.* We have

$$\Gamma_{n_1} = \frac{1}{\Delta} \sum_{i=n_0}^{n_1} \Gamma_{n_1}$$

$$\overset{(i)}{\leq} \frac{\alpha_{n_1}}{\Delta} \sum_{i=n_0}^{n_1} \Gamma_i$$

$$= \frac{\alpha_{n_1}}{\Delta} \sum_{i=n_0}^{n_1} \max_{\boldsymbol{x} \in \mathcal{S}} \mathrm{I}(\boldsymbol{f}_{\mathcal{A}_i}; y_{\boldsymbol{x}} \mid \boldsymbol{y}_{1:i})$$

$$\overset{(ii)}{=} \frac{\alpha_{n_1}}{\Delta} \sum_{i=n_0}^{n_1} \mathrm{I}(\boldsymbol{f}_{\mathcal{A}_i}; y_{\boldsymbol{x}_{i+1}} \mid \mathcal{D}_i)$$

$$\overset{(iii)}{\leq} \frac{\alpha_{n_1}}{\Delta} \sum_{i=n_0}^{n_1} \mathrm{I}(\boldsymbol{f}_{\mathcal{A}_{n_0}}; y_{\boldsymbol{x}_{i+1}} \mid \mathcal{D}_i)$$

$$\overset{(iv)}{=} \frac{\alpha_{n_1}}{\Delta} \sum_{i=n_0}^{n_1} \mathrm{I}(\boldsymbol{f}_{\mathcal{A}_{n_0}}; y_{\boldsymbol{x}_{i+1}} \mid \boldsymbol{y}_{\boldsymbol{x}_{n_0+1:i}}, \mathcal{D}_{n_0})$$

$$\overset{(v)}{=} \frac{\alpha_{n_1}}{\Delta} \mathrm{I}(\boldsymbol{f}_{\mathcal{A}_{n_0}}; \boldsymbol{y}_{\boldsymbol{x}_{n_0+1:n_1+1}} \mid \mathcal{D}_{n_0})$$

$$\leq \frac{\alpha_{n_1}}{\Delta} \max_{\substack{X \subseteq \mathcal{S} \\ |X|=\Delta}} \mathrm{I}(\boldsymbol{f}_{\mathcal{A}_{n_0}}; \boldsymbol{y}_X \mid \mathcal{D}_{n_0})$$

$$\overset{(vi)}{\leq} \frac{\alpha_{n_1}}{\Delta} \max_{\substack{X \subseteq \mathcal{S} \\ |X|=\Delta}} \mathrm{I}(\boldsymbol{f}_X; \boldsymbol{y}_X \mid \mathcal{D}_{n_0})$$

$$\overset{(vii)}{\leq} \frac{\alpha_{n_1}}{\Delta} \max_{\substack{X \subseteq \mathcal{S} \\ |X|=\Delta}} \mathrm{I}(\boldsymbol{f}_X; \boldsymbol{y}_X)$$

$$= \alpha_{n_1} \frac{\gamma_\Delta}{\Delta}$$

where $(i)$ follows from the definition of the task complexity $\alpha_{n_1}$ (cf. Definition C.6); $(ii)$ uses the objective of ITL and that the posterior variance of Gaussians is independent of the realization and only depends on the *location* of observations; $(iii)$ uses $\mathcal{A}_{i+1} \subseteq \mathcal{A}_i$ and monotonicity of information gain; $(iv)$ uses that the posterior variance of Gaussians is independent of the realization and only depends on the *location* of observations; $(v)$ uses the chain rule of information gain; $(vi)$ uses $\boldsymbol{y}_X \perp \boldsymbol{f}_{\mathcal{A}_{n_0}} \mid \boldsymbol{f}_X$ and the data processing inequality. The conditional independence follows from the assumption that the observation noise is independent. Similarly, $\boldsymbol{y}_X \perp \mathcal{D}_{n_0} \mid \boldsymbol{f}_X$ which implies $(vii)$.

If $n_0 = 0$, then the bound before line $(vi)$ simplifies to $\alpha_{n_1} \gamma_{\mathcal{A}_0,\mathcal{S}}(\Delta)/\Delta$. □

The result for VTL is stated, for simplicity, only for the case where the target space and sample space are fixed.

**Theorem C.13** (Convergence of Marginal Gain for VTL). *Assume that Assumptions B.1 and B.2 are satisfied. Then for any $n \geq 1$, if the sequence $\{\boldsymbol{x}_i\}_{i=1}^n$ is generated by* VTL,

$$\Gamma_{n-1} \leq \frac{2\sigma^2 \alpha_n}{n} \sum_{\boldsymbol{x}' \in \mathcal{A}} \gamma_{\{\boldsymbol{x}'\},\mathcal{S}}(n). \tag{12}$$

We remark that $\sum_{\boldsymbol{x}' \in \mathcal{A}} \gamma_{\{\boldsymbol{x}'\},\mathcal{S}}(n) \leq |\mathcal{A}| \gamma_{\mathcal{A},\mathcal{S}}(n)$.

*Proof.* We have

$$
\begin{aligned}
\Gamma_{n-1} &= \frac{1}{n} \sum_{i=0}^{n-1} \Gamma_{n-1} \\
&\overset{(i)}{\leq} \frac{\alpha_n}{n} \sum_{i=0}^{n-1} \Gamma_i \\
&= \frac{\alpha_n}{n} \sum_{i=0}^{n-1} \left[ \operatorname{tr} \operatorname{Var}[\boldsymbol{f}_\mathcal{A} \mid \boldsymbol{y}_{1:i}] - \min_{\boldsymbol{x} \in \mathcal{S}} \operatorname{tr} \operatorname{Var}[\boldsymbol{f}_\mathcal{A} \mid \boldsymbol{y}_{1:i}, y_{\boldsymbol{x}}] \right] \\
&\overset{(ii)}{=} \frac{\alpha_n}{n} \sum_{i=0}^{n-1} \left[ \operatorname{tr} \operatorname{Var}[\boldsymbol{f}_\mathcal{A} \mid \mathcal{D}_i] - \operatorname{tr} \operatorname{Var}[\boldsymbol{f}_\mathcal{A} \mid \mathcal{D}_{i+1}] \right] \\
&\overset{(iii)}{\leq} \frac{\sigma^2 \alpha_n}{n} \sum_{\boldsymbol{x}' \in \mathcal{A}} \sum_{i=0}^{n-1} \log\left( \frac{\operatorname{Var}[f_{\boldsymbol{x}'} \mid \mathcal{D}_n]}{\operatorname{Var}[f_{\boldsymbol{x}'} \mid \mathcal{D}_{n+1}]} \right) \\
&= \frac{2\sigma^2 \alpha_n}{n} \sum_{\boldsymbol{x}' \in \mathcal{A}} \sum_{i=0}^{n-1} \operatorname{I}\left( f_{\boldsymbol{x}'}; y_{\boldsymbol{x}_{n+1}} \mid \mathcal{D}_n \right) \\
&\overset{(iv)}{=} \frac{2\sigma^2 \alpha_n}{n} \sum_{\boldsymbol{x}' \in \mathcal{A}} \sum_{i=0}^{n-1} \operatorname{I}\left( f_{\boldsymbol{x}'}; y_{\boldsymbol{x}_{n+1}} \mid \boldsymbol{y}_{\boldsymbol{x}_{1:n}} \right) \\
&\overset{(v)}{=} \frac{2\sigma^2 \alpha_n}{n} \sum_{\boldsymbol{x}' \in \mathcal{A}} \operatorname{I}\left( f_{\boldsymbol{x}'}; \boldsymbol{y}_{\boldsymbol{x}_{1:n}} \right) \\
&\leq \frac{2\sigma^2 \alpha_n}{n} \sum_{\boldsymbol{x}' \in \mathcal{A}} \max_{\substack{X \subseteq \mathcal{S} \\ |X|=n}} \operatorname{I}(f_{\boldsymbol{x}'}; \boldsymbol{y}_X) \\
&= \frac{2\sigma^2 \alpha_n}{n} \sum_{\boldsymbol{x}' \in \mathcal{A}} \gamma_{\{\boldsymbol{x}'\},\mathcal{S}}(n)
\end{aligned}
$$

where $(i)$ follows from the definition of the task complexity $\alpha_{n_1}$ (cf. Definition C.6); $(ii)$ follows from the VTL decision rule and that the posterior variance of Gaussians is independent of the realization and only depends on the *location* of observations; $(iii)$ follows from Lemma C.38 and monotonicity of variance; $(iv)$ uses that the posterior variance of Gaussians is independent of the realization and only depends on the *location* of observations; and $(v)$ uses the chain rule of mutual information. The remainder of the proof is analogous to the proof of Theorem C.12 (cf. Appendix C.5). □

**Keeping track of the task complexity online** In general, the task complexity $\alpha_n$ may be larger than one in the "directed" setting (i.e., when $\mathcal{S} \not\subseteq \mathcal{A}$). However, note that $\alpha_n$ can easily be evaluated online by keeping track of the smallest $\Gamma_i$ during previous rounds $i$.

### C.6 Proof of Theorem 3.2

We will now prove Theorem 3.2. We first prove the convergence of marginal variance within $\mathcal{S}$ for ITL, before proving the convergence outside $\mathcal{S}$ in Appendix C.6.1.

**Lemma C.14** (Uniform convergence of marginal variance within $\mathcal{S}$ for ITL). *Assume that Assumptions B.1 and B.2 are satisfied. For any $n \geq 0$ and $x \in \mathcal{A} \cap \mathcal{S}$,*

$$\sigma_n^2(x) \leq 2\tilde{\sigma}^2 \cdot \Gamma_n. \tag{13}$$

*Proof.* We have

$$\sigma_n^2(x) = \text{Var}[f_x \mid \mathcal{D}_n] - \underbrace{\text{Var}[f_x \mid f_x, \mathcal{D}_n]}_{0}$$

$$\overset{(i)}{=} \text{Var}[y_x \mid \mathcal{D}_n] - \rho^2(x) - (\text{Var}[y_x \mid f_x, \mathcal{D}_n] - \rho^2(x))$$

$$= \text{Var}[y_x \mid \mathcal{D}_n] - \text{Var}[y_x \mid f_x, \mathcal{D}_n]$$

$$\overset{(ii)}{\leq} \tilde{\sigma}^2 \log\left(\frac{\text{Var}[y_x \mid \mathcal{D}_n]}{\text{Var}[y_x \mid f_x, \mathcal{D}_n]}\right)$$

$$= 2\tilde{\sigma}^2 \cdot \text{I}(f_x; y_x \mid \mathcal{D}_n)$$

$$\overset{(iii)}{\leq} 2\tilde{\sigma}^2 \cdot \text{I}(f_\mathcal{A}; y_x \mid \mathcal{D}_n)$$

$$\overset{(iv)}{\leq} 2\tilde{\sigma}^2 \cdot \max_{x' \in \mathcal{S}} \text{I}(f_\mathcal{A}; y_{x'} \mid \mathcal{D}_n)$$

$$\overset{(v)}{=} 2\tilde{\sigma}^2 \cdot \max_{x' \in \mathcal{S}} \text{I}(f_\mathcal{A}; y_{x'} \mid y_{1:n})$$

$$= 2\tilde{\sigma}^2 \cdot \Gamma_n$$

where $(i)$ follows from the noise assumption (cf. Assumption B.2); $(ii)$ follows from Lemma C.38 and using monotonicity of variance; $(iii)$ follows from $x \in \mathcal{A}$ and monotonicity of information gain; $(iv)$ follows from $x \in \mathcal{S}$; and $(v)$ uses that the posterior variance of Gaussians is independent of the realization and only depends on the *location* of observations. □

### C.6.1 Convergence outside $\mathcal{S}$ for ITL

We will now show convergence of marginal variance to the irreducible uncertainty for points outside the sample space.

Our proof roughly proceeds as follows: We construct an "approximate Markov boundary" of $x$ in $\mathcal{S}$, and show (1) that the size of this Markov boundary is independent of $n$, and (2) that a small uncertainty reduction within the Markov boundary implies that the marginal variances at the Markov boundary and(!) $x$ are small.

**Definition C.15** (Approximate Markov boundary). For any $\epsilon > 0$, $n \geq 0$, and $x \in \mathcal{X}$, we denote by $B_{n,\epsilon}(x)$ the smallest (multi-)subset of $\mathcal{S}$ such that

$$\text{Var}[f_x \mid \mathcal{D}_n, y_{B_{n,\epsilon}(x)}] \leq \eta_\mathcal{S}^2(x) + \epsilon. \tag{14}$$

We call $B_{n,\epsilon}(x)$ an *$\epsilon$-approximate Markov boundary* of $x$ in $\mathcal{S}$.

Equation (14) is akin to the notion of the smallest Markov blanket in $\mathcal{S}$ of some $x \in \mathcal{X}$ (called a *Markov boundary*) which is the smallest set $\mathcal{B} \subseteq \mathcal{S}$ such that $f_x \perp f_\mathcal{S} \mid f_\mathcal{B}$.

**Lemma C.16** (Existence of an approximate Markov boundary). *For any $\epsilon > 0$, let $k$ be the smallest integer satisfying*

$$\frac{\gamma_k}{k} \leq \frac{\epsilon \lambda_{\min}(K_{\mathcal{S}\mathcal{S}})}{2|\mathcal{S}|\sigma^2\tilde{\sigma}^2}. \tag{15}$$

*Then, for any $n \geq 0$ and $x \in \mathcal{X}$, there exists an $\epsilon$-approximate Markov boundary $B_{n,\epsilon}(x)$ of $x$ in $\mathcal{S}$ with size at most $k$.*

Lemma C.16 shows that for any $\epsilon > 0$ there exists a universal constant $b_\epsilon$ (with respect to $n$ and $x$) such that

$$|B_{n,\epsilon}(x)| \leq b_\epsilon \qquad \forall n \geq 0, x \in \mathcal{X}. \tag{16}$$

We defer the proof of Lemma C.16 to Appendix C.6.3 where we also provide an algorithm to compute $B_{n,\epsilon}(x)$.

**Lemma C.17.** *For any $\epsilon > 0$, $n \geq 0$, and $\boldsymbol{x} \in \mathcal{X}$,*

$$\sigma_n^2(\boldsymbol{x}) \leq 2\sigma^2 \cdot \mathrm{I}(f_{\boldsymbol{x}}; \boldsymbol{y}_{B_{n,\epsilon}(\boldsymbol{x})} \mid \mathcal{D}_n) + \eta_{\mathcal{S}}^2(\boldsymbol{x}) + \epsilon \tag{17}$$

*where $B_{n,\epsilon}(\boldsymbol{x})$ is an $\epsilon$-approximate Markov boundary of $\boldsymbol{x}$ in $\mathcal{S}$.*

*Proof.* We have

$$
\begin{aligned}
\sigma_n^2(\boldsymbol{x}) &= \mathrm{Var}[f_{\boldsymbol{x}} \mid \mathcal{D}_n] - \eta_{\mathcal{S}}^2(\boldsymbol{x}) + \eta_{\mathcal{S}}^2(\boldsymbol{x}) \\
&\overset{(i)}{\leq} \mathrm{Var}[f_{\boldsymbol{x}} \mid \mathcal{D}_n] - \mathrm{Var}[f_{\boldsymbol{x}} \mid \boldsymbol{y}_{B_{n,\epsilon}(\boldsymbol{x})}, \mathcal{D}_n] + \eta_{\mathcal{S}}^2(\boldsymbol{x}) + \epsilon \\
&\overset{(ii)}{\leq} \sigma^2 \log\left(\frac{\mathrm{Var}[f_{\boldsymbol{x}} \mid \mathcal{D}_n]}{\mathrm{Var}[f_{\boldsymbol{x}} \mid \boldsymbol{y}_{B_{n,\epsilon}(\boldsymbol{x})}, \mathcal{D}_n]}\right) + \eta_{\mathcal{S}}^2(\boldsymbol{x}) + \epsilon \\
&= 2\sigma^2 \cdot \mathrm{I}(f_{\boldsymbol{x}}; \boldsymbol{y}_{B_{n,\epsilon}(\boldsymbol{x})} \mid \mathcal{D}_n) + \eta_{\mathcal{S}}^2(\boldsymbol{x}) + \epsilon
\end{aligned}
$$

where $(i)$ follows from the defining property of an $\epsilon$-approximate Markov boundary (cf. Equation (14)); and $(ii)$ follows from Lemma C.38 and using monotonicity of variance. $\qquad \square$

**Lemma C.18.** *For any $\epsilon > 0$, $n \geq 0$, and $\boldsymbol{x} \in \mathcal{A}$,*

$$\mathrm{I}(f_{\boldsymbol{x}}; \boldsymbol{y}_{B_{n,\epsilon}(\boldsymbol{x})} \mid \mathcal{D}_n) \leq \frac{b_\epsilon}{\bar{\kappa}_n(B_{n,\epsilon}(\boldsymbol{x}))}\Gamma_n \tag{18}$$

*where $B_{n,\epsilon}(\boldsymbol{x})$ is an $\epsilon$-approximate Markov boundary of $\boldsymbol{x}$ in $\mathcal{S}$, $|B_{n,\epsilon}(\boldsymbol{x})| \leq b_\epsilon$, and where $\bar{\kappa}_n(\cdot) \overset{\mathrm{def}}{=} \bar{\kappa}(\cdot \mid \boldsymbol{x}_{1:n})$ denotes the information ratio from Equation (8).*

We remark that $\bar{\kappa}_n(\cdot) > 0$ as is shown in Lemma C.7, and hence, the right-hand side of the inequality is well-defined.

*Proof.* We use the abbreviated notation $B = B_{n,\epsilon}(\boldsymbol{x})$. We have

$$
\begin{aligned}
\mathrm{I}(f_{\boldsymbol{x}}; \boldsymbol{y}_B \mid \mathcal{D}_n) &\overset{(i)}{\leq} \mathrm{I}(\boldsymbol{f}_{\mathcal{A}}; \boldsymbol{y}_B \mid \mathcal{D}_n) \\
&\overset{(ii)}{\leq} \frac{1}{\bar{\kappa}_{n,b_\epsilon}} \sum_{\tilde{\boldsymbol{x}} \in B} \mathrm{I}(\boldsymbol{f}_{\mathcal{A}}; y_{\tilde{\boldsymbol{x}}} \mid \mathcal{D}_n) \\
&\overset{(iii)}{\leq} \frac{b_\epsilon}{\bar{\kappa}_{n,b_\epsilon}} \max_{\tilde{\boldsymbol{x}} \in B} \mathrm{I}(\boldsymbol{f}_{\mathcal{A}}; y_{\tilde{\boldsymbol{x}}} \mid \mathcal{D}_n) \\
&\overset{(iv)}{\leq} \frac{b_\epsilon}{\bar{\kappa}_{n,b_\epsilon}} \max_{\tilde{\boldsymbol{x}} \in \mathcal{S}} \mathrm{I}(\boldsymbol{f}_{\mathcal{A}}; y_{\tilde{\boldsymbol{x}}} \mid \mathcal{D}_n) \\
&\overset{(v)}{=} \frac{b_\epsilon}{\bar{\kappa}_{n,b_\epsilon}} \max_{\tilde{\boldsymbol{x}} \in \mathcal{S}} \mathrm{I}(\boldsymbol{f}_{\mathcal{A}}; y_{\tilde{\boldsymbol{x}}} \mid \boldsymbol{y}_{1:n}) \\
&= \frac{b_\epsilon}{\bar{\kappa}_{n,b_\epsilon}}\Gamma_n
\end{aligned}
$$

where $(i)$ follows from monotonicity of mutual information; $(ii)$ follows from the definition of the information ratio $\bar{\kappa}_{n,b_\epsilon}$ (cf. Equation (8)); $(iii)$ follows from $b \leq b_\epsilon$; $(iv)$ follows from $B \subseteq \mathcal{S}$; and $(v)$ uses that the posterior variance of Gaussians is independent of the realization and only depends on the *location* of observations. $\qquad \square$

*Proof of Theorem 3.2 for* ITL . The case where $\boldsymbol{x} \in \mathcal{A} \cap \mathcal{S}$ is shown by Lemma C.14 with $C = 2\tilde{\sigma}^2$.

To prove the more general result, fix any $\boldsymbol{x} \in \mathcal{A}$ and $\epsilon > 0$. By Lemma C.16, there exists an $\epsilon$-approximate Markov boundary $B_{n,\epsilon}(\boldsymbol{x})$ of $\boldsymbol{x}$ in $\mathcal{S}$ such that $|B_{n,\epsilon}(\boldsymbol{x})| \leq b_\epsilon$. We have

$$
\begin{aligned}
\sigma_n^2(\boldsymbol{x}) &\overset{(i)}{\leq} 2\sigma^2 \cdot \mathrm{I}(f_{\boldsymbol{x}}; \boldsymbol{y}_{B_{n,\epsilon}(\boldsymbol{x})} \mid \mathcal{D}_n) + \eta_{\mathcal{S}}^2(\boldsymbol{x}) + \epsilon \\
&\overset{(ii)}{\leq} \frac{2\sigma^2 b_\epsilon}{\bar{\kappa}_n(B_{n,\epsilon}(\boldsymbol{x}))}\Gamma_n + \eta_{\mathcal{S}}^2(\boldsymbol{x}) + \epsilon
\end{aligned}
$$

where $(i)$ follows from Lemma C.17; and $(ii)$ follows from Lemma C.18.

Let $\epsilon = c\frac{\gamma_{\sqrt{n}}}{\sqrt{n}}$ with $c = 2|\mathcal{S}|\sigma^2\tilde{\sigma}^2/\lambda_{\min}(\boldsymbol{K}_{\mathcal{S}\mathcal{S}})$. Then, by Equation (15), $b_\epsilon$ can be bounded for instance by $\sqrt{n}$. Together with Theorem C.12 this implies for ITL that

$$\sigma_n^2(\boldsymbol{x}) \le \eta_{\mathcal{S}}^2(\boldsymbol{x}) + 2\sigma^2\sqrt{n}\,\Gamma_n + c\gamma_{\sqrt{n}}/\sqrt{n}$$
$$\le \eta_{\mathcal{S}}^2(\boldsymbol{x}) + c'\gamma_n/\sqrt{n}$$

for a constant $c'$, e.g., $c' = 2\sigma^2 + c$. $\qquad\square$

### C.6.2 Convergence outside $\mathcal{S}$ for VTL

*Proof of Theorem 3.2 for* VTL . Analogously to Lemma C.17, we have

$$\sigma_n^2(\boldsymbol{x}) = \mathrm{Var}[f_{\boldsymbol{x}} \mid \mathcal{D}_n] - \eta_{\mathcal{S}}^2(\boldsymbol{x}) + \eta_{\mathcal{S}}^2(\boldsymbol{x})$$
$$\stackrel{(i)}{\le} \mathrm{Var}[f_{\boldsymbol{x}} \mid \mathcal{D}_n] - \mathrm{Var}[f_{\boldsymbol{x}} \mid \boldsymbol{y}_{B_{n,\epsilon}(\boldsymbol{x})}, \mathcal{D}_n] + \eta_{\mathcal{S}}^2(\boldsymbol{x}) + \epsilon$$

where $(i)$ follows from the defining property of an $\epsilon$-approximate Markov boundary (cf. Equation (14)). Further, we have

$$\mathrm{Var}[f_{\boldsymbol{x}} \mid \mathcal{D}_n] - \mathrm{Var}[f_{\boldsymbol{x}} \mid \boldsymbol{y}_{B_{n,\epsilon}(\boldsymbol{x})}, \mathcal{D}_n]$$
$$\stackrel{(i)}{\le} \sum_{\tilde{\boldsymbol{x}} \in B_{n,\epsilon}(\boldsymbol{x})} (\mathrm{Var}[f_{\boldsymbol{x}} \mid \mathcal{D}_n] - \mathrm{Var}[f_{\boldsymbol{x}} \mid y_{\tilde{\boldsymbol{x}}}, \mathcal{D}_n])$$
$$\stackrel{(ii)}{\le} \sum_{\tilde{\boldsymbol{x}} \in B_{n,\epsilon}(\boldsymbol{x})} (\mathrm{tr}\,\mathrm{Var}[\boldsymbol{f}_{\mathcal{A}} \mid \boldsymbol{y}_{1:n}] - \mathrm{tr}\,\mathrm{Var}[\boldsymbol{f}_{\mathcal{A}} \mid y_{\tilde{\boldsymbol{x}}}, \boldsymbol{y}_{1:n}])$$
$$\stackrel{(iii)}{\le} b_\epsilon \Gamma_n$$

where $(i)$ follows from the submodularity of $\psi_{\mathcal{A}}$; $(ii)$ uses that the posterior variance of Gaussians is independent of the realization and only depends on the *location* of observations; and $(iii)$ follows from the definition of $\Gamma_n$ and Lemma C.16.

The remainder of the proof is analogous to the result for ITL, using Theorem C.13 to bound $\Gamma_n$. $\qquad\square$

### C.6.3 Existence of an Approximate Markov Boundary

We now derive Lemma C.16 which shows the existence of an approximate Markov boundary of $\boldsymbol{x}$ in $\mathcal{S}$.

**Lemma C.19.** *For any $S \subseteq \mathcal{S}$ and $k \ge 0$, there exists $B \subseteq S$ with $|B| = k$ such that for all $\boldsymbol{x}' \in S$,*

$$\mathrm{Var}[f_{\boldsymbol{x}'} \mid \boldsymbol{y}_B] \le 2\tilde{\sigma}^2\frac{\gamma_k}{k}. \tag{19}$$

*Proof.* We choose $B \subseteq S$ greedily using the acquisition function

$$\tilde{\boldsymbol{x}}_k \stackrel{\mathrm{def}}{=} \arg\max_{\tilde{\boldsymbol{x}} \in S} \mathrm{I}(\boldsymbol{f}_S; y_{\tilde{\boldsymbol{x}}} \mid \boldsymbol{y}_{B_{k-1}})$$

where $B_k = \tilde{\boldsymbol{x}}_{1:k}$. Note that this is the "undirected" special case of ITL, and hence, we have

$$\mathrm{Var}\big[f_{\boldsymbol{x}'} \mid \boldsymbol{y}_{B_k}\big] \stackrel{(i)}{\le} 2\tilde{\sigma}^2\Gamma_k$$
$$\stackrel{(ii)}{\le} 2\tilde{\sigma}^2\frac{\gamma_k}{k}$$

where $(i)$ is due to Lemma C.14; and $(ii)$ is due to Theorem C.12 and $\alpha_{S,S}(k) \le 1$. $\qquad\square$

**Lemma C.20.** *Given any $\epsilon > 0$ and $B \subseteq S \subseteq \mathcal{S}$ with $|S| < \infty$, such that for any $\boldsymbol{x}' \in S$,*

$$\mathrm{Var}[f_{\boldsymbol{x}'} \mid \boldsymbol{y}_B] \le \frac{\epsilon\lambda_{\min}(\boldsymbol{K}_{SS})}{|S|\sigma^2}. \tag{20}$$

*Then for any $\boldsymbol{x} \in \mathcal{X}$,*

$$\mathrm{Var}[f_{\boldsymbol{x}} \mid \boldsymbol{y}_B] \le \mathrm{Var}[f_{\boldsymbol{x}} \mid \boldsymbol{f}_S] + \epsilon. \tag{21}$$

*Proof.* We will denote the right-hand side of Equation (20) by $\epsilon'$. We have

$$\mathrm{Var}[f_{\boldsymbol{x}} \mid \boldsymbol{y}_B]$$

$$\overset{(i)}{=} \mathbb{E}_{\boldsymbol{f}_S}[\mathrm{Var}_{f_{\boldsymbol{x}}}[f_{\boldsymbol{x}} \mid \boldsymbol{f}_S, \boldsymbol{y}_B] \mid \boldsymbol{y}_B]$$
$$\qquad + \mathrm{Var}_{\boldsymbol{f}_S}[\mathbb{E}_{f_{\boldsymbol{x}}}[f_{\boldsymbol{x}} \mid \boldsymbol{f}_S, \boldsymbol{y}_B] \mid \boldsymbol{y}_B]$$

$$\overset{(ii)}{=} \mathrm{Var}_{f_{\boldsymbol{x}}}[f_{\boldsymbol{x}} \mid \boldsymbol{f}_S, \boldsymbol{y}_B] + \mathrm{Var}_{\boldsymbol{f}_S}[\mathbb{E}_{f_{\boldsymbol{x}}}[f_{\boldsymbol{x}} \mid \boldsymbol{f}_S, \boldsymbol{y}_B] \mid \boldsymbol{y}_B]$$

$$\overset{(iii)}{=} \underbrace{\mathrm{Var}_{f_{\boldsymbol{x}}}[f_{\boldsymbol{x}} \mid \boldsymbol{f}_S]}_{\text{irreducible uncertainty}} + \underbrace{\mathrm{Var}_{\boldsymbol{f}_S}[\mathbb{E}_{f_{\boldsymbol{x}}}[f_{\boldsymbol{x}} \mid \boldsymbol{f}_S] \mid \boldsymbol{y}_B]}_{\text{reducible (epistemic) uncertainty}}$$

where $(i)$ follows from the law of total variance; $(ii)$ uses that the conditional variance of a Gaussian depends only on the location of observations and not on their value; and $(iii)$ follows from $f_{\boldsymbol{x}} \perp \boldsymbol{y}_B \mid \boldsymbol{f}_S$ since $B \subseteq S$. It remains to bound the reducible uncertainty.

Let $h_{\boldsymbol{x}} : \mathbb{R}^d \to \mathbb{R}$, $\boldsymbol{f}_S \mapsto \mathbb{E}[f_{\boldsymbol{x}} \mid \boldsymbol{f}_S]$ where we write $d \overset{\mathrm{def}}{=} |S|$. Using the formula for the GP posterior mean, we have

$$h_{\boldsymbol{x}}(\boldsymbol{f}_S) = \mathbb{E}[f_{\boldsymbol{x}}] + \boldsymbol{z}^\top (\boldsymbol{f}_S - \mathbb{E}[\boldsymbol{f}_S])$$

where $\boldsymbol{z} \overset{\mathrm{def}}{=} \boldsymbol{K}_{SS}^{-1} \boldsymbol{K}_{S\boldsymbol{x}}$. Because $h$ is a linear function in $\boldsymbol{f}_S$ we have for the reducible uncertainty that

$$\mathrm{Var}_{\boldsymbol{f}_S}[h_{\boldsymbol{x}}(\boldsymbol{f}_S) \mid \boldsymbol{y}_B] = \boldsymbol{z}^\top \mathrm{Var}[\boldsymbol{f}_S \mid \boldsymbol{y}_B] \boldsymbol{z}$$

$$\overset{(i)}{\leq} d \cdot \boldsymbol{z}^\top \mathrm{diag}\, \mathrm{Var}[\boldsymbol{f}_S \mid \boldsymbol{y}_B] \boldsymbol{z}$$

$$\overset{(ii)}{\leq} \epsilon' d \, \boldsymbol{z}^\top \boldsymbol{z}$$

$$= \epsilon' d \, \boldsymbol{K}_{\boldsymbol{x}S} \boldsymbol{K}_{SS}^{-1} \boldsymbol{K}_{SS}^{-1} \boldsymbol{K}_{S\boldsymbol{x}}$$

$$\leq \frac{\epsilon' d}{\lambda_{\min}(\boldsymbol{K}_{SS})} \boldsymbol{K}_{\boldsymbol{x}S} \boldsymbol{K}_{SS}^{-1} \boldsymbol{K}_{S\boldsymbol{x}}$$

$$\overset{(iii)}{\leq} \frac{\epsilon' d \sigma^2}{\lambda_{\min}(\boldsymbol{K}_{SS})}$$

where $(i)$ follows from Lemma C.37; $(ii)$ follows from the assumption that $\mathrm{Var}[f_{\boldsymbol{x}'} \mid \boldsymbol{y}_B] \leq \epsilon'$ for all $\boldsymbol{x}' \in S$; and $(iii)$ follows from

$$\boldsymbol{K}_{\boldsymbol{x}S} \boldsymbol{K}_{SS}^{-1} \boldsymbol{K}_{S\boldsymbol{x}} \leq \boldsymbol{K}_{\boldsymbol{x}\boldsymbol{x}} = \sigma^2$$

since $\boldsymbol{K}_{\boldsymbol{x}\boldsymbol{x}} - \boldsymbol{K}_{\boldsymbol{x}S} \boldsymbol{K}_{SS}^{-1} \boldsymbol{K}_{S\boldsymbol{x}} \geq 0$. $\qquad \square$

*Proof of Lemma C.16.* Let $B \subseteq \mathcal{S}$ be the set of size $k$ generated by Lemma C.19 to satisfy $\mathrm{Var}[f_{\boldsymbol{x}'} \mid \boldsymbol{y}_B] \leq 2\tilde{\sigma}^2 \gamma_k / k$ for all $\boldsymbol{x}' \in \mathcal{S}$. We have for any $\boldsymbol{x} \in \mathcal{X}$,

$$\mathrm{Var}[f_{\boldsymbol{x}} \mid \mathcal{D}_n, \boldsymbol{y}_B] \overset{(i)}{\leq} \mathrm{Var}[f_{\boldsymbol{x}} \mid \boldsymbol{y}_B]$$

$$\overset{(ii)}{\leq} \mathrm{Var}[f_{\boldsymbol{x}} \mid \boldsymbol{f}_{\mathcal{S}}] + \epsilon$$

where $(i)$ follows from monotonicity of variance; and $(ii)$ follows from Lemma C.20; using $|\mathcal{S}| < \infty$ and the condition on $k$. $\qquad \square$

We remark that Lemma C.19 provides an algorithm (just "undirected" ITL!) to compute an approximate Markov boundary, and the set $B$ returned by this algorithm is a valid approximate Markov boundary for all $\boldsymbol{x} \in \mathcal{X}$. One can simply swap-in ITL with target space $\{\boldsymbol{x}\}$ for "undirected" ITL to obtain tighter (but instance-dependent) bounds on the size of the approximate Markov boundary.

### C.6.4 Generalization to Continuous $\mathcal{S}$ for Finite Dimensional RKHSs

In this subsection we generalize Theorem 3.2 to continuous sample spaces $\mathcal{S}$. We will make the following assumption:

**Assumption C.21.** The RKHS of the kernel $k$ is finite dimensional. In other words, the kernel $k$ can be expressed as $k(\boldsymbol{x}, \boldsymbol{x}') = \boldsymbol{\phi}(\boldsymbol{x})^\top \boldsymbol{\phi}(\boldsymbol{x}')$ for some feature map $\boldsymbol{\phi} : \mathcal{X} \to \mathbb{R}^d$ with $d < \infty$.

In the following, we will denote the design matrix of the sample space $\mathcal{S}$ by $\boldsymbol{\Phi} \overset{\text{def}}{=} [\boldsymbol{\phi}(\boldsymbol{x}) : \boldsymbol{x} \in \mathcal{S}]^\top \in \mathbb{R}^{|\mathcal{S}| \times d}$, and we denote by $\boldsymbol{\Pi}_{\boldsymbol{\Phi}}$ its orthogonal projection onto the orthogonal complement of the span of $\boldsymbol{\Phi}$. In particular, it holds that

1. $\boldsymbol{\Pi}_{\boldsymbol{\Phi}} \boldsymbol{v} = \boldsymbol{0}$ for all $\boldsymbol{v} \in \text{span}\,\boldsymbol{\Phi}$, and
2. $\boldsymbol{\Pi}_{\boldsymbol{\Phi}} \boldsymbol{v} = \boldsymbol{v}$ for all $\boldsymbol{v} \in (\text{span}\,\boldsymbol{\Phi})^\perp$.

Especially, $\boldsymbol{v} \in \ker \boldsymbol{\Pi}_{\boldsymbol{\Phi}}$ if and only if $\boldsymbol{v} \in \text{span}\,\boldsymbol{\Phi}$. This projection can be computed as follows:

**Lemma C.22.** *It holds that*

$$\boldsymbol{\Pi}_{\boldsymbol{\Phi}} = \boldsymbol{I} - \boldsymbol{\Phi}^\top (\boldsymbol{\Phi}\boldsymbol{\Phi}^\top)^{-1} \boldsymbol{\Phi}. \tag{22}$$

*Proof.* $\boldsymbol{\Phi}^\top (\boldsymbol{\Phi}\boldsymbol{\Phi}^\top)^{-1} \boldsymbol{\Phi}$ is the orthogonal projection onto the span of $\boldsymbol{\Phi}$ (see, e.g., Strang, 2016, page 211). $\qquad\square$

**Lemma C.23.** *Under Assumption C.21, the irreducible uncertainty $\eta_{\mathcal{S}}^2(\boldsymbol{x})$ of $\boldsymbol{x} \in \mathcal{X}$ is*

$$\eta_{\mathcal{S}}^2(\boldsymbol{x}) = \|\boldsymbol{\phi}(\boldsymbol{x})\|_{\boldsymbol{\Pi}_{\boldsymbol{\Phi}}}^2 \tag{23}$$

*where $\|\boldsymbol{v}\|_{\boldsymbol{A}} = \sqrt{\boldsymbol{v}^\top \boldsymbol{A} \boldsymbol{v}}$ denotes the Mahalanobis distance.*

*Proof.* This is an immediate consequence of the formula for the conditional variance of multivariate Gaussians (cf. Appendix B.2), applied to the linear kernel. $\qquad\square$

Lemmas C.22 and C.23 imply that $\eta_{\mathcal{S}}^2(\boldsymbol{x}^\|) = 0$ for all $\boldsymbol{x}^\| \in \mathcal{X}$ with $\boldsymbol{\phi}(\boldsymbol{x}^\|) \in \text{span}\,\boldsymbol{\Phi}$. That is, the irreducible uncertainty is zero for points in the span of the sample space. In contrast, for points $\boldsymbol{x}^\perp$ with $\boldsymbol{\phi}(\boldsymbol{x}^\perp) \in (\text{span}\,\boldsymbol{\Phi})^\perp$, the irreducible uncertainty equals the initial uncertainty: $\eta_{\mathcal{S}}^2(\boldsymbol{x}^\perp) = \sigma_0^2(\boldsymbol{x}^\perp)$. The irreducible uncertainty of any other point $\boldsymbol{x}$ can be computed by simple decomposition of $\boldsymbol{\phi}(\boldsymbol{x})$ into parallel and orthogonal components.

Assuming that Assumption C.21 holds and given any (non-finite) $\mathcal{S} \subseteq \mathcal{X}$, there exists a basis $\Omega_{\mathcal{S}} \subseteq \mathcal{X}$ in the space of embeddings $\boldsymbol{\phi}(\cdot)$ such that $\text{span}\,\mathcal{S} = \text{span}\,\Omega_{\mathcal{S}}$ and $|\Omega_{\mathcal{S}}| \leq d$. The generalized existence of an approximate Markov boundary for continuous domains can then be shown analogously to Lemma C.16:

**Lemma C.24** (Existence of an approximate Markov boundary for a continuous domain)**.** *Let $\mathcal{S}$ be any (continuous) subset of $\mathcal{X}$ and let Assumption C.21 hold with $d < \infty$. Further, for any $\epsilon > 0$, let $k$ be the smallest integer satisfying*

$$\frac{\gamma_k}{k} \leq \frac{\epsilon \lambda_{\min}(\boldsymbol{K}_{\Omega_{\mathcal{S}} \Omega_{\mathcal{S}}})}{2d\sigma^2 \tilde{\sigma}^2}. \tag{24}$$

*Then, for any $n \geq 0$ and $\boldsymbol{x} \in \mathcal{X}$, there exists an $\epsilon$-approximate Markov boundary $B_{n,\epsilon}(\boldsymbol{x})$ of $\boldsymbol{x}$ in $\mathcal{S}$ with size at most $k$.*

*Proof sketch.* The proof follows analogously to Lemma C.16 by conditioning on the finite set $\Omega_{\mathcal{S}}$ as opposed to $\mathcal{S}$. $\qquad\square$

## C.7 Proof of Theorem 3.3

We first formalize the assumptions of Theorem 3.3:

**Assumption C.25** (Regularity of $f^\star$)**.** We assume that $f^\star$ is in a reproducing kernel Hilbert space $\mathcal{H}_k(\mathcal{X})$ associated with a kernel $k$ and has bounded norm, that is, $\|f^\star\|_k \leq B$ for some finite $B \in \mathbb{R}$.

**Assumption C.26** (Sub-Gaussian noise)**.** We further assume that each $\varepsilon_n$ from the noise sequence $\{\varepsilon_n\}_{n=1}^\infty$ is conditionally zero-mean $\rho(\boldsymbol{x}_n)$-sub-Gaussian with known constants $\rho(\boldsymbol{x}) > 0$ for all $\boldsymbol{x} \in \mathcal{X}$. Concretely,

$$\forall n \geq 1, \lambda \in \mathbb{R} : \quad \mathbb{E}\big[e^{\lambda \varepsilon_n} \,\big|\, \mathcal{D}_{n-1}\big] \leq \exp\left(\frac{\lambda^2 \rho^2(\boldsymbol{x}_n)}{2}\right)$$

where $\mathcal{D}_{n-1}$ corresponds to the $\sigma$-algebra generated by the random variables $\{\boldsymbol{x}_i, \epsilon_i\}_{i=1}^{n-1}$ and $\boldsymbol{x}_n$.

We make use of the following foundational result, showing that under the above two assumptions the (misspecified) Gaussian process model from Section 3.1 is an all-time well-calibrated model of $f^\star$:

**Lemma C.27** (Well-calibrated confidence intervals; Abbasi-Yadkori (2013); Chowdhury & Gopalan (2017))**.** *Pick $\delta \in (0,1)$ and let Assumptions C.25 and C.26 hold. Let*

$$\beta_n(\delta) = \|f^\star\|_k + \rho\sqrt{2(\gamma_n + 1 + \log(1/\delta))}$$

*where $\rho = \max_{\boldsymbol{x} \in \mathcal{X}} \rho(\boldsymbol{x})$.[10] Then, for all $\boldsymbol{x} \in \mathcal{X}$ and $n \geq 0$ jointly with probability at least $1 - \delta$,*

$$|f^\star(\boldsymbol{x}) - \mu_n(\boldsymbol{x})| \leq \beta_n(\delta) \cdot \sigma_n(\boldsymbol{x})$$

*where $\mu_n(\boldsymbol{x})$ and $\sigma_n^2(\boldsymbol{x})$ are mean and variance (as defined in Appendix B.2) of the GP posterior of $f(\boldsymbol{x})$ conditional on the observations $\mathcal{D}_n$, pretending that $\varepsilon_i$ is Gaussian with variance $\rho^2(\boldsymbol{x}_i)$.*

The proof of Theorem 3.3 is a straightforward application of Lemma C.27 and Theorem 3.2:

*Proof of Theorem 3.3.* By Theorem 3.2, we have that for all $\boldsymbol{x} \in \mathcal{A}$,

$$\sigma_n(\boldsymbol{x}) \leq \sqrt{\eta_{\mathcal{S}}^2(\boldsymbol{x}) + \nu_{\mathcal{A},\mathcal{S}}^2(n)} \leq \eta_{\mathcal{S}}(\boldsymbol{x}) + \nu_{\mathcal{A},\mathcal{S}}(n).$$

The result then follows by application of Lemma C.27. $\qquad\square$

## C.8 Proof of Theorem 5.1

In this section, we derive our main result on Safe BO. In Appendix C.8.1, we give the definition of the reachable safe set $\mathcal{R}$ and derive the conditions under which convergence to the reachable safe set is guaranteed. Then, in Appendix C.8.2, we prove Theorem 5.1.

**Notation** In the agnostic setting from Section 3.2 (i.e., under Assumptions C.25 and C.26), Lemma C.27 provides us with the following $(1 - \delta)$-confidence intervals (CIs)

$$\mathcal{C}_n(\boldsymbol{x}) \overset{\text{def}}{=} \mathcal{C}_{n-1}(\boldsymbol{x}) \cap [\mu_n(\boldsymbol{x}) \pm \beta_n(\delta) \cdot \sigma_n(\boldsymbol{x})] \tag{25}$$

where $\mathcal{C}_{-1}(\boldsymbol{x}) = \mathbb{R}$. We write $u_n(\boldsymbol{x}) \overset{\text{def}}{=} \max \mathcal{C}_n(\boldsymbol{x})$, $l_n(\boldsymbol{x}) \overset{\text{def}}{=} \min \mathcal{C}_n(\boldsymbol{x})$, and $w_n(\boldsymbol{x}) \overset{\text{def}}{=} u_n(\boldsymbol{x}) - l_n(\boldsymbol{x})$ for its upper bound, lower bound, and width, respectively.

We learn separate statistical models $f$ and $\{g_1, \ldots, g_q\}$ for the ground truth objective $f^\star$ and ground truth constraints $\{g_1^\star, \ldots, g_q^\star\}$. We write $\mathcal{I} \overset{\text{def}}{=} \{f, 1, \ldots, q\}$ and collect the constraints in $\mathcal{I}_s \overset{\text{def}}{=} \{1, \ldots, q\}$. Without loss of generality, we assume that the confidence intervals include the ground truths with probability at least $1 - \delta$ jointly for all $i \in \mathcal{I}$.[11] For $i \in \mathcal{I}$, denote by $u_{n,i}, l_{n,i}, w_{n,i}, \eta_i, \beta_{n,i}$ the respective quantities. In the following, we do not explicitly denote the dependence of $\beta_n$ on $\delta$.

To improve clarity, we will refer to the set of potential maximizers defined in Equation (5) as $\mathcal{M}_n$ and denote by $\mathcal{A}_n$ an arbitrary target space.

We point out the following corollary:

**Corollary C.28** (Safety)**.** *With high probability, jointly for any $n \geq 0$ and any $i \in \mathcal{I}_s$,*

$$\forall \boldsymbol{x} \in \mathcal{S}_n : g_i^\star(\boldsymbol{x}) \geq 0. \tag{26}$$

### C.8.1 Convergence to Reachable Safe Set

**Definition C.29** (Reachable safe set)**.** Given any pessimistic safe set $\mathcal{S} \subseteq \mathcal{X}$ and any $\epsilon \geq 0$ and $\beta \geq 0$, we define the *reachable safe set* up to $(\epsilon, \beta)$-slack and its closure as

$$\mathcal{R}_{\epsilon,\beta}(\mathcal{S}) \overset{\text{def}}{=} \mathcal{S} \cup \{\boldsymbol{x} \in \mathcal{X} \setminus \mathcal{S} \mid$$
$$g_i^\star(\boldsymbol{x}) - \beta(\eta_i(\boldsymbol{x}; \mathcal{S}) + \epsilon) \geq 0 \text{ for all } i \in \mathcal{I}_s\}$$
$$\bar{\mathcal{R}}_{\epsilon,\beta}(\mathcal{S}) \overset{\text{def}}{=} \lim_{n \to \infty} (\mathcal{R}_{\epsilon,\beta})^n(\mathcal{S})$$

where $(\mathcal{R}_{\epsilon,\beta})^n$ denotes the $n$-th composition of $\mathcal{R}_{\epsilon,\beta}$ with itself.

---

[10]$\beta_n(\delta)$ can be tightened adaptively (Emmenegger et al., 2023).

[11]This can be achieved by taking a union bound and rescaling $\delta$.

*Remark* C.30. Convergence of the safe set to the closure of the reachability operator can only be guaranteed for finite safe sets ($|\mathcal{S}^\star| < \infty$). The following proofs readily generalize to continuous domains by considering convergence within the $k$-th composition of the reachability operator with itself for some $k < \infty$. In this case the sample complexity grows with $k$ rather than $|\mathcal{S}^\star|$. The only required modification is to lift the assumption of Theorem C.12 that information is gained only while safe sets remain constant (i.e., $\mathcal{S}_{i+1} = \mathcal{S}_i$ for all $i$). This assumption is straightforward to lift since for any $n \geq 0$ and $T \geq 1$,

$$\max_{\boldsymbol{x} \in \mathcal{S}_n} \Delta_{\mathcal{A}}(\boldsymbol{x} \mid \boldsymbol{x}_{1:n+T}) \leq \frac{1}{T} \sum_{t=1}^{T} \max_{\boldsymbol{x} \in \mathcal{S}_n} \Delta_{\mathcal{A}}(\boldsymbol{x} \mid \boldsymbol{x}_{1:n+t}) \leq \frac{1}{T} \sum_{t=1}^{T} \max_{\boldsymbol{x} \in \mathcal{S}_{n+t}} \Delta_{\mathcal{A}}(\boldsymbol{x} \mid \boldsymbol{x}_{1:n+t}) \leq \frac{\gamma_T}{T},$$

using submodularity for the first inequality and the monotonicity of the safe set for the second inequality. In particular, this shows that one continues learning about points in the original safe set — even as the safe set grows.

We denote by $\mathcal{S}_0$ the initial pessimistic safe set induced by the (prior) statistical model $g$ (cf. Section 5) and write $\bar{\mathcal{R}}_{\epsilon,\beta} \stackrel{\text{def}}{=} \bar{\mathcal{R}}_{\epsilon,\beta}(\mathcal{S}_0)$.

**Lemma C.31** (Properties of the reachable safe set). *For all $\mathcal{S}, \mathcal{S}' \subseteq \mathcal{X}$, $\epsilon \geq 0$, and $\beta \geq 0$:*

*(i) $\mathcal{S}' \subseteq \mathcal{S} \implies \mathcal{R}_{\epsilon,\beta}(\mathcal{S}') \subseteq \mathcal{R}_{\epsilon,\beta}(\mathcal{S})$,*

*(ii) $\mathcal{R}_{\epsilon,\beta}(\mathcal{S}) \subseteq \mathcal{S} \implies \bar{\mathcal{R}}_{\epsilon,\beta}(\mathcal{S}) \subseteq \mathcal{S}$, and*

*(iii) $\mathcal{R}_{0,0}(\emptyset) = \bar{\mathcal{R}}_{0,0} = \mathcal{S}^\star$.*

*Proof (adapted from lemma 7.1 of Berkenkamp et al. (2021)).*

1. Let $\boldsymbol{x} \in \mathcal{R}_{\epsilon,\beta}(\mathcal{S}')$. If $\boldsymbol{x} \in \mathcal{S}$ then $\boldsymbol{x} \in \mathcal{R}_{\epsilon,\beta}(\mathcal{S})$, so let $\boldsymbol{x} \notin \mathcal{S}$. Then, by definition, for all $i \in \mathcal{I}_s$, $f_i^\star(\boldsymbol{x}) - \beta \eta_i(\boldsymbol{x}; \mathcal{S}') - \epsilon \geq 0$. By the monotonicity of variance, $\eta_i(\boldsymbol{x}; \mathcal{S}') \geq \eta_i(\boldsymbol{x}; \mathcal{S})$ for all $i \in \mathcal{I}$, and hence $f_i^\star(\boldsymbol{x}) - \beta \eta_i(\boldsymbol{x}; \mathcal{S}) - \epsilon \geq 0$ for all $i \in \mathcal{I}_s$. It follows that $\boldsymbol{x} \in \mathcal{R}_{\epsilon,\beta}(\mathcal{S})$.

2. By the monotonicity of variance, $\eta_i(\boldsymbol{x}; \mathcal{R}_{\epsilon,\beta}(\mathcal{S})) \geq \eta_i(\boldsymbol{x}; \mathcal{S})$ for all $\boldsymbol{x} \in \mathcal{X}$ and $i \in \mathcal{I}$. Thus, by definition of the safe region, we have that $\mathcal{R}_{\epsilon,\beta}(\mathcal{R}_{\epsilon,\beta}(\mathcal{S})) \subseteq \mathcal{S}$. The result follows by taking the limit.

3. The result follows directly from the definition of the true safe set $\mathcal{S}^\star$ (cf. Equation (4)). $\qquad\square$

Clearly, we cannot expand the safe set beyond $\bar{\mathcal{R}}_{0,0}$. The following is our main intermediate result, showing that either we expand the safe set at some point or the uncertainty converges to the irreducible uncertainty.

**Lemma C.32.** *Given any $n_0 \geq 0$, $\epsilon > 0$, let $n'$ be the smallest integer such that $\nu_{n',\tilde{\epsilon}^2} \leq \tilde{\epsilon}$ where $\tilde{\epsilon} = \epsilon/2$. Let $\beta_{n_0+n'} = \max_{i \in \mathcal{I}_s} \beta_{n_0+n',i}$. Assume that the sequence of target spaces is monotonically decreasing, i.e., $\mathcal{A}_{n+1} \subseteq \mathcal{A}_n$. Then, we have with high probability (at least) one of*

$$\Big( \forall \boldsymbol{x} \in \mathcal{A}_{n_0+n'}, \ \forall i \in \mathcal{I} :$$
$$w_{n_0+n',i}(\boldsymbol{x}) \leq \beta_{n_0+n'} [\eta_i(\boldsymbol{x}; \mathcal{S}_{n_0+n'}) + \epsilon]$$
$$and \quad \mathcal{A}_{n_0+n'} \cap \mathcal{R}_{\epsilon,\beta_{n_0+n'}}(\mathcal{S}_{n_0+n'}) \subseteq \mathcal{S}_{n_0+n'} \Big)$$

*or $|\mathcal{S}_{n_0+n'+1}| > |\mathcal{S}_{n_0}|$.*

*Proof.* Suppose that $|\mathcal{S}_{n_0+n'+1}| = |\mathcal{S}_{n_0}|$. Then, by Theorem 3.3 (using that the sequence of target spaces is monotonically decreasing), for any $\boldsymbol{x} \in \mathcal{A}_{n_0+n'}$ and $i \in \mathcal{I}$,

$$w_{n_0+n',i}(\boldsymbol{x}) \leq \beta_{n_0+n'} [\eta_i(\boldsymbol{x}; \mathcal{S}_{n_0+n'}) + \epsilon].$$

As $\mathcal{S}_{n_0+n'+1} = \mathcal{S}_{n_0+n'}$ we have for all $\boldsymbol{x} \in \mathcal{A}_{n_0+n'} \setminus \mathcal{S}_{n_0+n'}$ and $i \in \mathcal{I}_s$, with high probability that

$$0 > l_{n_0+n',i}(\boldsymbol{x}) \geq g_i^\star(\boldsymbol{x}) - w_{n_0+n',i}(\boldsymbol{x})$$
$$\geq g_i^\star(\boldsymbol{x}) - \beta_{n_0+n'} [\eta_i(\boldsymbol{x}; \mathcal{S}_{n_0+n'}) + \epsilon].$$

It follows that $\mathcal{A}_{n_0+n'} \cap \mathcal{R}_{\epsilon,\beta_{n_0+n'}}(\mathcal{S}_{n_0+n'}) \subseteq \mathcal{S}_{n_0+n'}$. $\qquad\square$

To gather more intuition about the above lemma, consider the target space

$$\mathcal{E}_n \overset{\text{def}}{=} \widehat{\mathcal{S}}_n \setminus \mathcal{S}_n. \tag{27}$$

We call $\mathcal{E}_n$ the *potential expanders* since it contains all points which might be safe, but are not yet known to be safe. Under this target space, the above lemma simplifies slightly:

**Lemma C.33.** *For any $n \geq 0$ and $\epsilon, \beta \geq 0$, if $\mathcal{E}_n \subseteq \mathcal{A}_n$ then with high probability,*

$$\mathcal{S}_n \cup (\mathcal{A}_n \cap \mathcal{R}_{\epsilon,\beta}(\mathcal{S}_n)) = \mathcal{R}_{\epsilon,\beta}(\mathcal{S}_n).$$

*Proof.* With high probability, $\mathcal{R}_{\epsilon,\beta}(\mathcal{S}_n) \subseteq \widehat{\mathcal{S}}_n = \mathcal{S}_n \cup \mathcal{E}_n$. The lemma is a direct consequence. $\square$

The above lemmas can be combined to yield our main result of this subsection, establishing the convergence of ITL to the reachable safe set.

**Theorem C.34** (Convergence to reachable safe set). *For any $\epsilon > 0$, let $n'$ be the smallest integer satisfying the condition of Lemma C.32, and define $n^\star \overset{\text{def}}{=} (|\mathcal{S}^\star| + 1)n'$. Let $\bar{\beta}_{n^\star} \geq \beta_{n,i}$ for all $n \leq n^\star, i \in \mathcal{I}_s$. Assume that the sequence of target spaces is monotonically decreasing, i.e., $\mathcal{A}_{n+1} \subseteq \mathcal{A}_n$. Then, the following inequalities hold jointly with probability at least $1 - \delta$:*

*(i)* $\forall n \geq 0, \ \forall i \in \mathcal{I}_s : g_i^\star(\boldsymbol{x}_n) \geq 0,$

*safety*

*(ii)* $\mathcal{A}_{n^\star} \cap \bar{\mathcal{R}}_{\epsilon,\bar{\beta}_{n^\star}} \subseteq \mathcal{S}_{n^\star} \subseteq \bar{\mathcal{R}}_{0,0} = \mathcal{S}^\star,$

*convergence to safe region*

*(iii)* $\forall \boldsymbol{x} \in \mathcal{A}_{n^\star}, \ \forall i \in \mathcal{I} : w_{n^\star,i}(\boldsymbol{x}) \leq \bar{\beta}_{n^\star} \eta_i(\boldsymbol{x}; \bar{\mathcal{R}}_{\epsilon,\bar{\beta}_{n^\star}}) + \epsilon,$

*convergence of width*

*(iv)* $\forall \boldsymbol{x} \in \bar{\mathcal{R}}_{\epsilon,\bar{\beta}_{n^\star}}, \ \forall i \in \mathcal{I} : \eta_i(\boldsymbol{x}; \bar{\mathcal{R}}_{\epsilon,\bar{\beta}_{n^\star}}) = 0.$

*convergence of width within safe region*

*Proof.* $(i)$ is a direct consequence of Corollary C.28. $\mathcal{S}_{n^\star} \subseteq \mathcal{S}^\star$ follows directly from the pessimistic safe set $\mathcal{S}_{n^\star}$ from $(ii)$ being a subset of the true safe set $\mathcal{S}^\star$. $(iv)$ follows directly from the definition of irreducible uncertainty. Thus, it remains to establish $\mathcal{A}_{n^\star} \cap \bar{\mathcal{R}}_{\epsilon,\bar{\beta}_{n^\star}} \subseteq \mathcal{S}_{n^\star}$ and $(iii)$.

Recall that with high probability $|\mathcal{S}_n| \in [0, |\mathcal{S}^\star|]$ for all $n \geq 0$. Thus, the size of the pessimistic safe set can increase at most $|\mathcal{S}^\star|$ many times. By Lemma C.32, using the assumption on $n'$, the size of the pessimistic safe set increases at least once every $n'$ iterations, or else:

$$\forall \boldsymbol{x} \in \mathcal{A}_{n_0+n'}, \ \forall i \in \mathcal{I} : w_{n_0+n',i}(\boldsymbol{x}) \leq \beta_{n_0+n'}[\eta_i(\boldsymbol{x}; \mathcal{S}_{n_0+n'}) + \epsilon]$$
$$\text{and} \quad \mathcal{A}_{n_0+n'} \cap \mathcal{R}_{\epsilon,\beta_{n_0+n'}}(\mathcal{S}_{n_0+n'}) \subseteq \mathcal{S}_{n_0+n'}. \tag{28}$$

Because the safe set can expand at most $|\mathcal{S}^\star|$ many times, Equation (28) occurs eventually for some $n_0 \leq |\mathcal{S}^\star|n'$. In this case, since $\bar{\beta}_{n^\star} \geq \beta_{n_0+n'}$ and $\mathcal{A}_{n^\star} \subseteq \mathcal{A}_{n_0+n'}$ (as $n_0 + n' \leq n^\star$) we have that

$$\mathcal{A}_{n^\star} \cap \mathcal{R}_{\epsilon,\bar{\beta}_{n^\star}}(\mathcal{S}_{n_0+n'}) \subseteq \mathcal{A}_{n_0+n'} \cap \mathcal{R}_{\epsilon,\beta_{n_0+n'}}(\mathcal{S}_{n_0+n'})$$
$$\subseteq \mathcal{S}_{n_0+n'}.$$

By Lemma C.31 (ii), this implies

$$\mathcal{A}_{n^\star} \cap \bar{\mathcal{R}}_{\epsilon,\bar{\beta}_{n^\star}} \subseteq \mathcal{S}_{n_0+n'} \subseteq \mathcal{S}_{n^\star}.$$

$\square$

We emphasize that Theorem C.34 holds for arbitrary target spaces $\mathcal{A}_n$. If additionally, $\mathcal{E}_n \subseteq \mathcal{A}_n$ for all $n \geq 0$ then by Lemma C.33, Theorem C.34 (ii) strengthens to $\bar{\mathcal{R}}_{\epsilon,\bar{\beta}_{n^\star}} \subseteq \mathcal{S}_{n^\star}$. Intuitively, $\mathcal{E}_n \subseteq \mathcal{A}_n$ ensures that one aims to expand the safe set in *all* directions. Conversely, if $\mathcal{E}_n \not\subseteq \mathcal{A}_n$ then one aims only to expand the safe set in the direction of $\mathcal{A}_n$ (or not at all if $\mathcal{A}_n \subseteq \mathcal{S}_n$).

**"Free" convergence guarantees in many applications** Theorem C.34 can be specialized to yield convergence guarantees in various settings by choosing an appropriate target space $\mathcal{A}_n$. Straightforward application of Theorem C.34 (informally) requires that the sequence of target spaces is monotonically decreasing (i.e., $\mathcal{A}_{n+1} \subseteq \mathcal{A}_n$), and that each target space $\mathcal{A}_n$ is an "over-approximation" of the actual set of targeted points (such as the set of optimas in the Bayesian optimization setting). We discuss two such applications in the following.

1. *Pure expansion:* For example, for the target space $\mathcal{E}_n$, Theorem C.34 bounds the convergence of the safe set to the reachable safe set. In this case, the transductive active learning problem corresponds to the "pure expansion" setting, also addressed by the ISE baseline discussed in Section 5. The ISE baseline, however, does not establish convergence guarantees of the kind of Theorem C.34. Note that $\mathcal{E}_n$ satisfies the (informal) requirements laid out previously, since it is monotonically decreasing by definition, and with high probability, any point $\boldsymbol{x} \in \mathcal{S}^\star$ that is not in $\mathcal{S}_n$ is contained within $\mathcal{E}_n$.

2. *Level set estimation:* Given any $\tau \in \mathbb{R}$, we denote the (safe) $\tau$-level set of $f^\star$ by $\mathcal{L}^\tau \stackrel{\text{def}}{=} \{\boldsymbol{x} \in \mathcal{S}^\star \mid f^\star(\boldsymbol{x}) = \tau\}$. We define the *potential level set* as

$$\mathcal{L}_n^\tau \stackrel{\text{def}}{=} \{\boldsymbol{x} \in \widehat{\mathcal{S}}_n \mid l_n^f(\boldsymbol{x}) \leq \tau \leq u_n^f(\boldsymbol{x})\}. \tag{29}$$

That is, $\mathcal{L}_n^\tau$ is the subset of the optimistic safe set $\widehat{\mathcal{S}}_n$ where the $\tau$-level set of $f^\star$ may be located. Analogously to the potential expanders, it is straightforward to show that $\mathcal{L}_n^\tau$ over-approximates the true $\tau$-level set and is monotonically decreasing.

We remark that our guarantees from this section also apply to the standard ("unsafe") setting where $\mathcal{S}^\star = \mathcal{S}_0 = \mathcal{X}$.

### C.8.2 Convergence to Safe Optimum

In this section, we specialize Theorem C.34 for the case that the target space contains the potential maximizers $\mathcal{M}_n$ (cf. Equation (5)). It is straightforward to see that the sequence $\mathcal{M}_n$ is monotonically decreasing (i.e., $\mathcal{M}_{n+1} \subseteq \mathcal{M}_n$). The following lemma shows that the potential maximizers over-approximate the set of safe maxima $\mathcal{X}^* \stackrel{\text{def}}{=} \arg\max_{\boldsymbol{x} \in \mathcal{S}^\star} f^\star(\boldsymbol{x})$.

**Lemma C.35** (Potential maximizers over-approximate safe maxima)**.** *For all $n \geq 0$ and with probability at least $1 - \delta$,*

 *(i) $\boldsymbol{x} \in \mathcal{X}^*$ implies $\boldsymbol{x} \in \mathcal{M}_n$ and*

 *(ii) $\boldsymbol{x} \notin \mathcal{M}_n$ implies $\boldsymbol{x} \notin \mathcal{X}^*$.*

*Proof.* If $\boldsymbol{x} \notin \mathcal{M}_n$ then

$$u_{n,f}(\boldsymbol{x}) < \max_{\boldsymbol{x}' \in \mathcal{S}_n} l_{n,f}(\boldsymbol{x}') \leq \max_{\boldsymbol{x}' \in \mathcal{S}^\star} l_{n,f}(\boldsymbol{x}')$$

where we used $\mathcal{S}_n \subseteq \mathcal{S}^\star$ with high probability, which directly implies with high probability that $\boldsymbol{x} \notin \mathcal{X}^*$.

For the other direction, if $\boldsymbol{x} \in \mathcal{X}^*$ then

$$u_{n,f}(\boldsymbol{x}) \geq \max_{\boldsymbol{x}' \in \mathcal{S}^\star} l_{n,f}(\boldsymbol{x}') \geq \max_{\boldsymbol{x}' \in \mathcal{S}_n} l_{n,f}(\boldsymbol{x}')$$

with high probability. $\qquad\square$

We denote the set of optimal actions which are safe up to $(\epsilon, \beta)$-slack by

$$\mathcal{X}^*_{\epsilon,\beta} \stackrel{\text{def}}{=} \arg\max_{\boldsymbol{x} \in \bar{\mathcal{R}}_{\epsilon,\beta}} f^\star(\boldsymbol{x}),$$

and by $f^*_{\epsilon,\beta}$ the maximum value attained by $f^\star$ at any of the points in $\mathcal{X}^*_{\epsilon,\beta}$. The regret can be expressed as

$$r_n(\bar{\mathcal{R}}_{\epsilon,\beta}) = f^*_{\epsilon,\beta} - f^\star(\widehat{\boldsymbol{x}}_n)$$

The following theorem formalizes Theorem 5.1 and establishes convergence to the safe optimum.

**Theorem C.36** (Convergence to safe optimum). *For any $\epsilon > 0$, let $n'$ be the smallest integer satisfying the condition of Lemma C.32, and define $n^\star \stackrel{\text{def}}{=} (|\mathcal{S}^\star| + 1)n'$. Let $\bar{\beta}_{n^\star} \geq \beta_{n,i}$ for all $n \leq n^\star, i \in \mathcal{I}_s$. Then, the following inequalities hold jointly with probability at least $1 - \delta$:*

*(i) $\forall n \geq 0, \ \forall i \in \mathcal{I}_s : g_i^\star(\boldsymbol{x}_n) \geq 0$,*

*safety*

*(ii) $\forall n \geq n^\star : r_n(\bar{\mathcal{R}}_{\epsilon,\bar{\beta}_{n^\star}}) \leq \epsilon$.*

*convergence to safe optimum*

*Proof.* Fix any $\boldsymbol{x}^* \in \mathcal{X}_{\epsilon,\bar{\beta}_{n^\star}}^* \subseteq \bar{\mathcal{R}}_{\epsilon,\bar{\beta}_{n^\star}}$. Assume w.l.o.g. that $\boldsymbol{x}^* \in \mathcal{M}_{n^\star}$.[12] Then, with high probability,

$$
\begin{aligned}
f_{\epsilon,\bar{\beta}_{n^\star}}^* = f^\star(\boldsymbol{x}^*) &\leq u_{n^\star,f}(\boldsymbol{x}^*) \\
&= l_{n^\star,f}(\boldsymbol{x}^*) + w_{n^\star,f}(\boldsymbol{x}^*) \\
&\stackrel{(i)}{\leq} l_{n^\star,f}(\widehat{\boldsymbol{x}}_{n^\star}) + w_{n^\star,f}(\boldsymbol{x}^*) \\
&\leq f^\star(\widehat{\boldsymbol{x}}_{n^\star}) + w_{n^\star,f}(\boldsymbol{x}^*) \\
&\stackrel{(ii)}{\leq} f^\star(\widehat{\boldsymbol{x}}_{n^\star}) + \epsilon
\end{aligned}
$$

where $(i)$ follows from the definition of $\widehat{\boldsymbol{x}}_n$; and $(ii)$ follows from Theorem C.34 and noting that $\boldsymbol{x}^* \in \mathcal{M}_{n^\star} \cap \bar{\mathcal{R}}_{\epsilon,\bar{\beta}_{n^\star}}$.

We have shown that $f^\star(\widehat{\boldsymbol{x}}_{n^\star}) \geq f_{\epsilon,\bar{\beta}_{n^\star}}^* - \epsilon$, which implies $r_{n^\star}(\bar{\mathcal{R}}_{\epsilon,\bar{\beta}_{n^\star}}) \leq \epsilon$. Since the upper- and lower-confidence bounds are monotonically decreasing / increasing, respectively, we have that for all $n \geq n^\star, r_n(\bar{\mathcal{R}}_{\epsilon,\bar{\beta}_{n^\star}}) \leq \epsilon$. $\qquad\square$

## C.9 Useful Facts and Inequalities

We denote by $\preceq$ the Loewner partial ordering of symmetric matrices.

**Lemma C.37.** *Let $\boldsymbol{A} \in \mathbb{R}^{n \times n}$ be a positive definite matrix with diagonal $\boldsymbol{D}$. Then, $\boldsymbol{A} \preceq n\boldsymbol{D}$.*

*Proof.* Equivalently, one can show $n\boldsymbol{D} - \boldsymbol{A} \succeq \boldsymbol{0}$. We write $\boldsymbol{A} \stackrel{\text{def}}{=} \boldsymbol{D}^{1/2}\boldsymbol{Q}\boldsymbol{D}^{1/2}$, and thus, $\boldsymbol{Q} = \boldsymbol{D}^{-1/2}\boldsymbol{A}\boldsymbol{D}^{-1/2}$ is a positive definite symmetric matrix with all diagonal elements equal to $1$. It remains to show that

$$
n\boldsymbol{D} - \boldsymbol{A} = \boldsymbol{D}^{1/2}(n\boldsymbol{I} - \boldsymbol{Q})\boldsymbol{D}^{1/2} \succeq \boldsymbol{0}.
$$

Note that $\sum_{i=1}^n \lambda_i(\boldsymbol{Q}) = \operatorname{tr} \boldsymbol{Q} = n$, and hence, all eigenvalues of $\boldsymbol{Q}$ belong to $(0, n)$. $\qquad\square$

**Lemma C.38.** *If $a, b \in (0, M]$ for some $M > 0$ and $b \geq a$ then*

$$
b - a \leq M \cdot \log\left(\frac{b}{a}\right). \tag{30}
$$

*If additionally, $a \geq M'$ for some $M' > 0$ then*

$$
b - a \geq M' \cdot \log\left(\frac{b}{a}\right). \tag{31}
$$

*Proof.* Let $f(x) \stackrel{\text{def}}{=} \log x$. By the mean value theorem, there exists $c \in (a, b)$ such that

$$
\frac{1}{c} = f'(c) = \frac{f(b) - f(a)}{b - a} = \frac{\log b - \log a}{b - a} = \frac{\log(\frac{b}{a})}{b - a}.
$$

Thus,

$$
b - a = c \cdot \log\left(\frac{b}{a}\right) < M \cdot \log\left(\frac{b}{a}\right).
$$

---

[12]Otherwise, with high probability, $f^\star(\widehat{\boldsymbol{x}}_{n^\star}) > f_{\epsilon,\bar{\beta}_{n^\star}}^*$.

Under the additional condition that $a \geq M'$, we obtain

$$b - a = c \cdot \log\left(\frac{b}{a}\right) > M' \cdot \log\left(\frac{b}{a}\right).$$

$\square$

# D    Interpretations & Approximations of Principle (†)

We give a brief overview of interpretations and approximations of ITL, as well as alternative decision rules adhering to the fundamental principle (†).

The discussed interpretations of (†) differ mainly in how they quantify the "uncertainty" about $\mathcal{A}$. In the GP setting, this "uncertainty" is captured by the covariance matrix $\boldsymbol{\Sigma}$ of $\boldsymbol{f}_{\mathcal{A}}$, and we consider two main ways of "scalarizing" $\boldsymbol{\Sigma}$:

1. the total (marginal) variance $\mathrm{tr}\,\boldsymbol{\Sigma}$, and
2. the "generalized variance" $|\boldsymbol{\Sigma}|$.

The generalized variance — which was originally suggested by Wilks (1932) as a generalization of variance to multiple dimensions — takes into account correlations. In contrast, the total variance discards all correlations between points in $\mathcal{A}$.

All discussed decision rules following principle (†) (i.e., ITL, VTL, MM-ITL) differ only in their weighting of the points in $\mathcal{A}$, and they coincide when $|\mathcal{A}| = 1$.

## D.1    Interpretations of ITL

We briefly discuss three interpretations of ITL.

**Minimizing generalized variance**    In the GP setting, ITL can be equivalently characterized as minimizing generalized posterior variance:

$$\begin{aligned}
\boldsymbol{x}_n &= \arg\max_{\boldsymbol{x} \in \mathcal{S}} \mathrm{I}(\boldsymbol{f}_{\mathcal{A}}; y_{\boldsymbol{x}} \mid \mathcal{D}_n) \\
&= \arg\max_{\boldsymbol{x} \in \mathcal{S}} \frac{1}{2} \log\left(\frac{|\mathrm{Var}[\boldsymbol{f}_{\mathcal{A}} \mid \mathcal{D}_{n-1}]|}{|\mathrm{Var}[\boldsymbol{f}_{\mathcal{A}} \mid \mathcal{D}_{n-1}, y_{\boldsymbol{x}}]|}\right) \\
&= \arg\min_{\boldsymbol{x} \in \mathcal{S}} |\mathrm{Var}[\boldsymbol{f}_{\mathcal{A}} \mid \mathcal{D}_{n-1}, y_{\boldsymbol{x}}]|.
\end{aligned} \tag{32}$$

**Maximizing relevance and minimizing redundancy**    An alternative interpretation of ITL is

$$\mathrm{I}(\boldsymbol{f}_{\mathcal{A}}; y_{\boldsymbol{x}} \mid \mathcal{D}_n) = \underbrace{\mathrm{I}(\boldsymbol{f}_{\mathcal{A}}; y_{\boldsymbol{x}})}_{\text{relevance}} - \underbrace{\mathrm{I}(\boldsymbol{f}_{\mathcal{A}}; y_{\boldsymbol{x}}; \mathcal{D}_n)}_{\text{redundancy}} \tag{33}$$

where $\mathrm{I}(\boldsymbol{f}_{\mathcal{A}}; y_{\boldsymbol{x}}; \mathcal{D}_n) = \mathrm{I}(\boldsymbol{f}_{\mathcal{A}}; y_{\boldsymbol{x}}) - \mathrm{I}(\boldsymbol{f}_{\mathcal{A}}; y_{\boldsymbol{x}} \mid \mathcal{D}_n)$ denotes the *multivariate information gain* (cf. Appendix B). In this way, ITL can be seen as maximizing observation relevance while minimizing observation redundancy. This interpretation is common in the literature on feature selection (Peng et al., 2005; Vergara & Estévez, 2014; Beraha et al., 2019).

**Steepest descent in measure spaces**    ITL can be seen as performing steepest descent in the space of probability measures over $\boldsymbol{f}_{\mathcal{A}}$, with the KL divergence as metric:

$$\mathrm{I}(\boldsymbol{f}_{\mathcal{A}}; y_{\boldsymbol{x}} \mid \mathcal{D}_n) = \mathbb{E}_{y_{\boldsymbol{x}}}[\mathrm{KL}(p(\boldsymbol{f}_{\mathcal{A}} \mid \mathcal{D}_n, y_{\boldsymbol{x}}) \| p(\boldsymbol{f}_{\mathcal{A}} \mid \mathcal{D}_n))].$$

That is, ITL finds the observation yielding the "largest update" to the current density.

## D.2    Interpretations of VTL

Quantifying the uncertainty about $\boldsymbol{f}_{\mathcal{A}}$ by the marginal variance of points in $\mathcal{A}$ rather than entropy (or generalized variance), the principle (†) leads to VTL. Note that if $|\mathcal{A}| = 1$, then VTL is equivalent to ITL. Unlike the similar, but more technical, TRUVAR algorithm proposed by Bogunovic et al. (2016), VTL does not require truncated variances, and hence, VTL can be applied to constrained settings (where $\mathcal{A} \not\subseteq \mathcal{S}$) as well.

**Relationship to ITL**   Note that the ITL criterion in the GP setting can be expressed as

$$\boldsymbol{x}_n = \operatorname*{arg\,min}_{\boldsymbol{x}\in\mathcal{S}} \operatorname{tr} \log \operatorname{Var}[\boldsymbol{f}_{\mathcal{A}} \mid \mathcal{D}_{n-1}, y_{\boldsymbol{x}}] \tag{34}$$

where for a positive semi-definite matrix $\boldsymbol{A}$ with spectral decomposition $\boldsymbol{A} = \boldsymbol{V}\boldsymbol{\Lambda}\boldsymbol{V}^\top$ we write $\log \boldsymbol{A} = \boldsymbol{V} \log \boldsymbol{\Lambda}\boldsymbol{V}^\top$ for the logarithmic matrix function. To derive Equation (34) we use that $\log |\boldsymbol{A}| = \sum_i \log \lambda_i(\boldsymbol{A}) = \operatorname{tr}\log \boldsymbol{A}$. Hence, ITL and VTL are identical up to a different weighting of the eigenvalues of the posterior covariance matrix.

**Minimizing a bound to the approximation error**   Chowdhury & Gopalan (2017) (page 19) bound the approximation error $|f^\star(\boldsymbol{x}) - \mu_n(\boldsymbol{x})|$ by

$$\underbrace{|\boldsymbol{k}_t(\boldsymbol{x})^\top(\boldsymbol{K}_t + \boldsymbol{P}_t)^{-1}\boldsymbol{\varepsilon}_{1:t}|}_{\text{variance}} + \underbrace{|f^\star(\boldsymbol{x}) - \boldsymbol{k}_t(\boldsymbol{x})^\top(\boldsymbol{K}_t + \boldsymbol{P}_t)^{-1}\boldsymbol{f}_{1:t}|}_{\text{bias}}$$

where $\boldsymbol{k}_t(\boldsymbol{x}) \overset{\text{def}}{=} \boldsymbol{K}_{\boldsymbol{x}\boldsymbol{x}_{1:t}}$, $\boldsymbol{K}_t \overset{\text{def}}{=} \boldsymbol{K}_{\boldsymbol{x}_{1:t}\boldsymbol{x}_{1:t}}$, and $\boldsymbol{P}_t \overset{\text{def}}{=} \boldsymbol{P}_{\boldsymbol{x}_{1:t}}$. Similar to a standard bias-variance decomposition, the first term measures variance and the second term measures bias. Following Lemma C.27, VTL can be seen as greedily minimizing this bound to the approximation error (i.e., both bias and variance).

**Maximizing correlation to prediction targets weighted by their variance**   It can be shown (see the proof below) that the VTL decision rule is equivalent to

$$\boldsymbol{x}_n = \operatorname*{arg\,max}_{\boldsymbol{x}\in\mathcal{S}} \sum_{\boldsymbol{x}'\in\mathcal{A}} \operatorname{Var}[f_{\boldsymbol{x}'} \mid \mathcal{D}_{n-1}] \cdot \operatorname{Cor}[f_{\boldsymbol{x}'}, y_{\boldsymbol{x}} \mid \mathcal{D}_{n-1}]^2. \tag{35}$$

That is, VTL maximizes the squared correlation between the next observation and the prediction targets, weighted by their variance. Intuitively, prediction targets are weighted by their variance since more can be learned about a prediction target with higher variance. This is precisely what leads to the "diverse" sample selection, and is akin to "uncertainty sampling" among the prediction targets and then selecting the observation which is most correlated with the selected prediction target.

*Proof.* Starting with the VTL objective, we have

$$
\begin{aligned}
\operatorname*{arg\,min}_{\boldsymbol{x}\in\mathcal{S}} \sum_{\boldsymbol{x}'\in\mathcal{A}} \operatorname{Var}[f_{\boldsymbol{x}'} \mid \mathcal{D}_n, y_{\boldsymbol{x}}] &= \operatorname*{arg\,min}_{\boldsymbol{x}\in\mathcal{S}} \sum_{\boldsymbol{x}'\in\mathcal{A}} \left( \operatorname{Var}[f_{\boldsymbol{x}'} \mid \mathcal{D}_n] - \frac{\operatorname{Cov}[f_{\boldsymbol{x}'}, y_{\boldsymbol{x}} \mid \mathcal{D}_n]^2}{\operatorname{Var}[y_{\boldsymbol{x}} \mid \mathcal{D}_n]} \right) \\
&= \operatorname*{arg\,max}_{\boldsymbol{x}\in\mathcal{S}} \sum_{\boldsymbol{x}'\in\mathcal{A}} \frac{\operatorname{Var}[f_{\boldsymbol{x}'} \mid \mathcal{D}_n] \cdot \operatorname{Cov}[f_{\boldsymbol{x}'}, y_{\boldsymbol{x}} \mid \mathcal{D}_n]^2}{\operatorname{Var}[f_{\boldsymbol{x}'} \mid \mathcal{D}_n] \cdot \operatorname{Var}[y_{\boldsymbol{x}} \mid \mathcal{D}_n]} + \text{const} \\
&= \operatorname*{arg\,max}_{\boldsymbol{x}\in\mathcal{S}} \sum_{\boldsymbol{x}'\in\mathcal{A}} \operatorname{Var}[f_{\boldsymbol{x}'} \mid \mathcal{D}_n] \cdot \operatorname{Cor}[f_{\boldsymbol{x}'}, y_{\boldsymbol{x}} \mid \mathcal{D}_n]^2 + \text{const}.
\end{aligned}
$$

$\square$

### D.3   Mean Marginal ITL

MacKay (1992) previously proposed "mean-marginal" ITL (MM-ITL) in the setting where $\mathcal{S} = \mathcal{X}$, which selects

$$\boldsymbol{x}_n = \operatorname*{arg\,max}_{\boldsymbol{x}\in\mathcal{S}} \sum_{\boldsymbol{x}'\in\mathcal{A}} \mathrm{I}(f_{\boldsymbol{x}'}; y_{\boldsymbol{x}} \mid \mathcal{D}_{n-1}) \tag{36}$$

and which simplifies in the GP setting to

$$
\begin{aligned}
\boldsymbol{x}_n &= \operatorname*{arg\,max}_{\boldsymbol{x}\in\mathcal{S}} \frac{1}{2} \sum_{\boldsymbol{x}'\in\mathcal{A}} \log\left( \frac{\operatorname{Var}[f_{\boldsymbol{x}'} \mid \mathcal{D}_{n-1}]}{\operatorname{Var}[f_{\boldsymbol{x}'} \mid \mathcal{D}_{n-1}, y_{\boldsymbol{x}}]} \right) \\
&= \operatorname*{arg\,min}_{\boldsymbol{x}\in\mathcal{S}} \sum_{\boldsymbol{x}'\in\mathcal{A}} \log \operatorname{Var}[f_{\boldsymbol{x}'} \mid \mathcal{D}_{n-1}, y_{\boldsymbol{x}}] \\
&= \operatorname*{arg\,min}_{\boldsymbol{x}\in\mathcal{S}} \operatorname{tr} \log \operatorname{diag} \operatorname{Var}[\boldsymbol{f}_{\mathcal{A}} \mid \mathcal{D}_{n-1}, y_{\boldsymbol{x}}]. \tag{37}
\end{aligned}
$$

Analogously to the derivation of Equation (34), this can also be expressed as

$$\boldsymbol{x}_n = \underset{\boldsymbol{x} \in \mathcal{S}}{\arg\min} \left| \operatorname{diag} \operatorname{Var}[\boldsymbol{f}_{\mathcal{A}} \mid \mathcal{D}_{n-1}, y_{\boldsymbol{x}}] \right|. \tag{38}$$

Effectively, MM-ITL ignores the mutual interaction between points in $\mathcal{A}$. As can be seen from Equation (37) and as is also mentioned by MacKay (1992), MM-ITL is equivalent to VTL up to a different weighting of the points in $\mathcal{A}$: instead of minimizing the average posterior variance (as in VTL), MM-ITL minimizes the average posterior log-variance. Under the lens of principle (†), this can be seen as minimizing the average marginal entropy of predictions within the target space:

$$\boldsymbol{x}_n = \underset{\boldsymbol{x} \in \mathcal{S}}{\arg\min} \sum_{\boldsymbol{x}' \in \mathcal{A}} \operatorname{H}[f_{\boldsymbol{x}'} \mid \mathcal{D}_{n-1}, y_{\boldsymbol{x}}].$$

We remark that MM-ITL is a special case of EPIG (Bickford Smith et al., 2023, Appendix E.2).

**Not a generalization of uncertainty sampling** Unlike ITL, MM-ITL is not a generalization of uncertainty sampling. The reason is precisely that MM-ITL ignores the mutual interaction between points in $\mathcal{A}$. Consider the example where $\mathcal{X} = \mathcal{S} = \mathcal{A} = \{1, \ldots, 10\}$ where $\boldsymbol{f}_{1:9}$ are highly correlated while $f_{10}$ is mostly independent of the other points. Visually, imagine a smooth function (i.e., under a Gaussian kernel) with points 1 through 9 close to each other and point 10 far away. Further, suppose that point 10 has a slightly larger marginal variance than the others. Then, MM-ITL would select one of the points $1:9$ since this leads to the largest reduction in the marginal (log-)variances (i.e., to a small posterior "uncertainty").[13] In contrast, ITL selects the point with the largest prior marginal variance (cf. Appendix C.1), point 10, since this leads to the largest reduction in entropy.[14]

**Similarity to VTL** Observe that the following decision rule is equivalent to VTL:

$$\boldsymbol{x}_n = \underset{\boldsymbol{x} \in \mathcal{S}}{\arg\max} \operatorname{tr} \operatorname{Var}[\boldsymbol{f}_{\mathcal{A}} \mid \mathcal{D}_{n-1}] - \operatorname{tr} \operatorname{Var}[\boldsymbol{f}_{\mathcal{A}} \mid \mathcal{D}_{n-1}, y_{\boldsymbol{x}}].$$

By Lemma C.38, for any $\boldsymbol{x} \in \mathcal{S}$, this objective value can be tightly lower- and upper-bounded (up to constant-factors) by

$$\sum_{\boldsymbol{x}' \in \mathcal{A}} \log\left( \frac{\operatorname{Var}[f_{\boldsymbol{x}'} \mid \mathcal{D}_{n-1}]}{\operatorname{Var}[f_{\boldsymbol{x}'} \mid \mathcal{D}_{n-1}, y_{\boldsymbol{x}}]} \right)$$

$$= 2 \sum_{\boldsymbol{x}' \in \mathcal{A}} \operatorname{I}(f_{\boldsymbol{x}'}; y_{\boldsymbol{x}} \mid \mathcal{D}_{n-1}) \qquad \text{(see MM-ITL)}$$

$$\overset{(i)}{=} - \sum_{\boldsymbol{x}' \in \mathcal{A}} \log\left( 1 - \operatorname{Cor}[f_{\boldsymbol{x}'}, y_{\boldsymbol{x}} \mid \mathcal{D}_{n-1}]^2 \right) \tag{39}$$

where $(i)$ is detailed in example 8.5.1 of Cover (1999). Thus, VTL and MM-ITL are closely related.

**Experiments** In our experiments with Gaussian processes from Figures 2 and 6, we observe that MM-ITL performs similarly to VTL and CTL.

**Convergence of uncertainty** We derive a convergence guarantee for MM-ITL which is analogous to the guarantees for ITL from Theorem C.12 and for VTL from Theorem C.13. We will assume for simplicity that $\Gamma_n$ is monotonically decreasing in $n$ (i.e., $\alpha_n \leq 1$).

**Theorem D.1** (Convergence of uncertainty reduction of MM-ITL). *Assume that Assumptions B.1 and B.2 are satisfied. Then for any $n \geq 1$, if $\Gamma_0 \geq \cdots \geq \Gamma_{n-1}$ and the sequence $\{\boldsymbol{x}_i\}_{i=1}^{n}$ is generated by* MM-ITL*, then*

$$\Gamma_{n-1} \leq \frac{1}{n} \sum_{\boldsymbol{x}' \in \mathcal{A}} \gamma_n(\{\boldsymbol{x}'\}; \mathcal{S}). \tag{40}$$

*Proof.* We have

$$\Gamma_{n-1} = \frac{1}{n} \sum_{i=0}^{n-1} \Gamma_{n-1}$$

---

[13]This is because the observation reduces uncertainty not just about the observed point itself.

[14]Because points $\boldsymbol{f}_{1:9}$ are highly correlated, $\operatorname{H}[\boldsymbol{f}_{1:9}]$ is already "small".

$$\overset{(i)}{\leq} \frac{1}{n} \sum_{i=0}^{n-1} \Gamma_i$$

$$= \frac{1}{n} \sum_{i=0}^{n-1} \max_{\boldsymbol{x} \in \mathcal{S}} \sum_{\boldsymbol{x}' \in \mathcal{A}} \mathrm{I}(f_{\boldsymbol{x}'}; y_{\boldsymbol{x}} \mid \mathcal{D}_n)$$

$$\overset{(ii)}{=} \frac{1}{n} \sum_{i=0}^{n-1} \sum_{\boldsymbol{x}' \in \mathcal{A}} \mathrm{I}(f_{\boldsymbol{x}'}; y_{\boldsymbol{x}_{n+1}} \mid \mathcal{D}_n)$$

$$\overset{(iii)}{=} \frac{1}{n} \sum_{\boldsymbol{x}' \in \mathcal{A}} \sum_{i=0}^{n-1} \mathrm{I}(f_{\boldsymbol{x}'}; y_{\boldsymbol{x}_{n+1}} \mid \boldsymbol{y}_{\boldsymbol{x}_{1:n}})$$

$$\overset{(iv)}{=} \frac{1}{n} \sum_{\boldsymbol{x}' \in \mathcal{A}} \mathrm{I}(f_{\boldsymbol{x}'}; \boldsymbol{y}_{\boldsymbol{x}_{1:n}})$$

$$\leq \frac{1}{n} \sum_{\boldsymbol{x}' \in \mathcal{A}} \max_{\substack{X \subseteq \mathcal{S} \\ |X|=n}} \mathrm{I}(f_{\boldsymbol{x}'}; \boldsymbol{y}_X)$$

$$= \frac{1}{n} \sum_{\boldsymbol{x}' \in \mathcal{A}} \gamma_n(\{\boldsymbol{x}'\}; \mathcal{S})$$

where $(i)$ follows by assumption; $(ii)$ follows from the MM-ITL decision rule; $(iii)$ uses that the posterior variance of Gaussians is independent of the realization and only depends on the *location* of observations; and $(iv)$ uses the chain rule of mutual information. The remainder of the proof is analogous to the proof of Theorem C.12 (cf. Appendix C.5). $\qquad\square$

Noting that

$$\mathrm{I}(f_{\boldsymbol{x}'}; y_{\boldsymbol{x}} \mid \mathcal{D}_{n-1}) \leq \sum_{\boldsymbol{x}' \in \mathcal{A}} \mathrm{I}(f_{\boldsymbol{x}'}; y_{\boldsymbol{x}} \mid \mathcal{D}_{n-1})$$

for any $n \geq 1$, $\boldsymbol{x} \in \mathcal{X}$, and $\boldsymbol{x}' \in \mathcal{A}$, Theorem 3.2 can be readily rederived for MM-ITL (cf. Lemmas C.14 and C.18). Hence, the posterior marginal variances of MM-ITL can be bounded uniformly in terms of $\Gamma_n$ analogously to ITL.

### D.4 Correlation-based Transductive Learning

We will briefly look at the CTL (*Correlation-based TL*) decision rule

$$\boldsymbol{x}_n = \arg\max_{\boldsymbol{x} \in \mathcal{S}} \sum_{\boldsymbol{x}' \in \mathcal{A}} \mathrm{Cor}[f_{\boldsymbol{x}}, f_{\boldsymbol{x}'} \mid \mathcal{D}_{n-1}] \tag{41}$$

which permits no interpretation under principle (†). However, if all correlations are non-negative (such as for the standard Gaussian and Matérn kernels), CTL is closely related to ITL, VTL, and MM-ITL (cf. Equations (35) and (39)). In this case, if $|\mathcal{A}| = 1$, then all decision rules coincide.

If, on the other hand, correlations may be negative then there is a crucial difference between CTL and the decision rules motivated from principle (†). Namely, decision rules following (†) exhibit a preference for points with high *absolute* correlation to prediction targets as opposed to CTL which prefers points with high *positive* correlation. This stems from the intuitive fact that points with a strong negative correlation are equally informative as points with a strong positive correlation. Nevertheless, we observe in our experiments that (even for a linear kernel which does not ensure non-negative correlations) points selected by ITL and VTL are typically positively correlated with prediction targets.

### D.5 Summary

We have seen that ITL, VTL, and MM-ITL can be seen as different interpretations of the same fundamental principle (†), with the approximations CTL. If $|\mathcal{A}| = 1$ and correlations are non-negative, then all four decision rules are equivalent. CTL prefers points with high positive

correlation whereas the other decision rules prefer points with high absolute correlation. ITL is the only decision rule that takes into account the mutual dependence between points in $\mathcal{A}$, and VTL and MM-ITL differ only in their weighting of the posterior marginal variances of points in $\mathcal{A}$.

# E   Stochastic Target Spaces

When the target space $\mathcal{A}$ is large, it may be computationally infeasible to compute the exact objective. A natural approach to address this issue is to approximate the target space by a smaller set of size $K$.

**Discretizing the target space**   One possibility is to discretize the target space $\mathcal{A}$. Compact target spaces can be addressed, e.g., via discretization arguments which are common in the Bayesian optimization literature (see, e.g., appendix C.1 of Srinivas et al. (2009)). That is, if the target space can be covered approximately using a finite (possibly large) set of points, the guarantees of Theorem 3.2 extend directly. This, however, can be impractical when the required size of discretization for sufficiently small approximation error is large. In the following, we briefly discuss a natural alternative approach based on sampling points from $\mathcal{A}$.

**Target distributions**   Let $\mathcal{A} \subseteq \mathcal{X}$ be a (possibly continuous) target space, and let $\mathcal{P}_{\mathcal{A}}$ be a probability distribution supported on $\mathcal{A}$. In iteration $n$, a subset $A_n$ of $K$ points is sampled independently from $\mathcal{A}$ according to the distribution $\mathcal{P}_{\mathcal{A}}$ and the objective is computed on this subset. Formally, this amounts to a single-sample Monte Carlo approximation of

$$\boldsymbol{x}_n \in \arg\max_{\boldsymbol{x} \in \mathcal{S}} \mathbb{E}_{A \overset{\text{iid}}{\sim} \mathcal{P}_{\mathcal{A}}}[\mathrm{I}(\boldsymbol{f}_A; y_{\boldsymbol{x}} \mid \mathcal{D}_{n-1})]. \tag{42}$$

The convergence guarantees from Appendix C can be generalized to the setting of stochastic target spaces by estimating how often points "near" a specified prediction target $\boldsymbol{x} \in \mathcal{A}$ are sampled.

**Definition E.1** ($\gamma$-ball at $\boldsymbol{x}$). Given $\boldsymbol{x} \in \mathcal{A}$ and any $\gamma \geq 0$, we call the set

$$B_\gamma(\boldsymbol{x}) \overset{\text{def}}{=} \{\boldsymbol{x}' \in \mathcal{X} \mid \|\boldsymbol{x} - \boldsymbol{x}'\| \leq \gamma\}$$

the $\gamma$-ball at $\boldsymbol{x}$. Further, we call $\mathcal{P}_{\mathcal{A}}(B_\gamma(\boldsymbol{x}))$ the weight of that ball.

**Proposition E.2** (sketch). *Given any $n \geq 1, K \geq 1, \gamma > 0$, and $\boldsymbol{x} \in \mathcal{A}$, suppose that $B_\gamma(\boldsymbol{x})$ has weight $p > 0$. Assume that the* ITL *objective is $L_I$-Lipschitz continuous. Then, with probability at least $1 - \exp(-(1-p)n/(8K))$,*

$$\sigma_n^2(\boldsymbol{x}) \lesssim \eta_{\mathcal{S}}^2(\boldsymbol{x}) + CL_I\gamma\frac{\gamma_{k(n)}}{\sqrt{k(n)}}$$

*where $k(n) \overset{\text{def}}{=} Kpn/2$.*

*Proof sketch.* Let $Y_i \sim \text{Binom}(K, p)$ denote the random variable counting the number of occurrences of a point from $B_\gamma(\boldsymbol{x})$ in $A_i$. Moreover, we write $X_i \overset{\text{def}}{=} \mathbb{1}\{B_\gamma(\boldsymbol{x}) \cap A_i \neq \emptyset\}$. Note that

$$\nu \overset{\text{def}}{=} \mathbb{E}X_i = \mathbb{P}(B_\gamma(\boldsymbol{x}) \cap A_i \neq \emptyset) = 1 - \mathbb{P}(Y_i = 0) = 1 - (1-p)^K \approx Kp$$

where the approximation stems from a first-order truncation of the Bernoulli series. Let $X \overset{\text{def}}{=} \sum_{i=1}^n X_i$ with $\mathbb{E}X = n\nu \approx Kpn$.

Using the assumed Lipschitz-continuity of the objective, we know that $\mathrm{I}(\boldsymbol{f}_{A'}; y_{\boldsymbol{x}} \mid \mathcal{D}_{n-1}) \leq L_I\gamma\mathrm{I}(\boldsymbol{f}_A; y_{\boldsymbol{x}} \mid \mathcal{D}_{n-1})$ where $A' \overset{\text{def}}{=} (A \setminus \{\boldsymbol{x}_\gamma\}) \cup \{\boldsymbol{x}\}$ and $\boldsymbol{x}_\gamma$ is the point from the $\gamma$-ball at $\boldsymbol{x}$. The bound then follows analogously to Theorem 3.2.

Finally, by Chernoff's bound, at least $Kpn/2$ iterations contain a point from $B_\gamma(\boldsymbol{x})$ with probability at least $1 - \exp(-Kpn/8)$. $\qquad\square$

This strategy can also be used to generalize the VTL, CTL, and MM-ITL objectives to stochastic target spaces.

# F   Closed-form Decision Rules

Below, we list the closed-form expressions for the ITL and VTL objectives. In the following, $k_n$ denotes the kernel conditional on $\mathcal{D}_n$.

**ITL**

$$\mathrm{I}(\boldsymbol{f}_{\mathcal{A}}; y_{\boldsymbol{x}} \mid \mathcal{D}_{n-1}) = \frac{1}{2} \log\left( \frac{\mathrm{Var}[y_{\boldsymbol{x}} \mid \mathcal{D}_{n-1}]}{\mathrm{Var}[y_{\boldsymbol{x}} \mid \boldsymbol{f}_{\mathcal{A}}, \mathcal{D}_{n-1}]} \right) \tag{43}$$

$$= \frac{1}{2} \log\left( \frac{k_{n-1}(\boldsymbol{x}, \boldsymbol{x}) + \rho^2}{\hat{k}_{n-1}(\boldsymbol{x}, \boldsymbol{x}) + \rho^2} \right)$$

where $\hat{k}_n(\boldsymbol{x}, \boldsymbol{x}) = k_n(\boldsymbol{x}, \boldsymbol{x}) - \boldsymbol{k}_n(\boldsymbol{x}, \mathcal{A}) \boldsymbol{K}_n(\mathcal{A}, \mathcal{A})^{-1} \boldsymbol{k}_n(\mathcal{A}, \boldsymbol{x})$.

**VTL**

$$\mathrm{tr}\, \mathrm{Var}[\boldsymbol{f}_{\mathcal{A}} \mid \mathcal{D}_{n-1}, y_{\boldsymbol{x}}] = \sum_{\boldsymbol{x}' \in \mathcal{A}} \left( k_{n-1}(\boldsymbol{x}', \boldsymbol{x}') - \frac{k_{n-1}(\boldsymbol{x}, \boldsymbol{x}')^2}{k_{n-1}(\boldsymbol{x}, \boldsymbol{x}) + \rho^2} \right).$$

## G   Computational Complexity

Evaluating the acquisition function of ITL in round $n$ requires computing for each $\boldsymbol{x} \in \mathcal{S}$,

$$\mathrm{I}(\boldsymbol{f}_{\mathcal{A}}; y_{\boldsymbol{x}} \mid \mathcal{D}_n)$$

$$= \frac{1}{2} \log\left( \frac{|\mathrm{Var}[\boldsymbol{f}_{\mathcal{A}} \mid \mathcal{D}_n]|}{|\mathrm{Var}[\boldsymbol{f}_{\mathcal{A}} \mid y_{\boldsymbol{x}}, \mathcal{D}_n]|} \right) \qquad \text{(forward)}$$

$$= \frac{1}{2} \log\left( \frac{\mathrm{Var}[y_{\boldsymbol{x}} \mid \mathcal{D}_n]}{\mathrm{Var}[y_{\boldsymbol{x}} \mid \boldsymbol{f}_{\mathcal{A}}, \mathcal{D}_n]} \right) \qquad \text{(backward)}.$$

Let $|\mathcal{S}| = m$ and $|\mathcal{A}| = k$. Then, the forward method has complexity $O(m \cdot k^3)$. For the backward method, observe that the variances are scalar and the covariance matrix $\mathrm{Var}[\boldsymbol{f}_{\mathcal{A}} \mid \mathcal{D}_n]$ only has to be inverted once for all points $\boldsymbol{x}$. Thus, the backward method has complexity $O(k^3 + m)$.

When the size $m$ of $\mathcal{S}$ is relatively small (and hence, all points in $\mathcal{S}$ can be considered during each iteration of the algorithm), GP inference corresponds simply to computing conditional distributions of a multivariate Gaussian. The performance can therefore be improved by keeping track of the full posterior distribution over $\boldsymbol{f}_{\mathcal{S}}$ of size $O(m^2)$ and conditioning on the latest observation during each iteration of the algorithm. In this case, after each observation the posterior can be updated at a cost of $O(m^2)$ which does not grow with the time $n$, unlike classical GP inference.

Overall, when $m$ is small, the computational complexity of ITL is $O(k^3 + m^2)$. When $m$ is large (or possibly infinite) and a subset of $\tilde{m}$ points is considered in a given iteration, the computational complexity of ITL is $O(k^3 + \tilde{m} \cdot n^3)$, neglecting the complexity of selecting the $\tilde{m}$ candidate points. In the latter case, the computational cost of ITL is dominated by the cost of GP inference.

Khanna et al. (2017) discuss distributed and stochastic approximations of greedy algorithms to (weakly) submodular problems that are also applicable to ITL.

## H   Additional GP Experiments & Details

We use homoscedastic Gaussian noise with standard deviation $\rho = 0.1$ and a discretization of $\mathcal{X} = [-3, 3]^2$ of size $2\,500$. Uncertainty bands correspond to one standard error over 10 random seeds.

**Additional experiments**   Figure 6 includes the following additional experiments:

1. *Extrapolation Setting* $(\mathcal{A} \cap \mathcal{S} = \emptyset)$: Right experiment from Figure 2 under the Gaussian kernel. ITL has a similar advantage as in the setting shown in Figure 3.

2. *Heteroscedastic Noise*: Left experiment from Figure 2 under the Gaussian kernel with heteroscedastic Gaussian noise

$$\rho(\boldsymbol{x}) = \begin{cases} 1 & \text{if } \boldsymbol{x} \in [-\frac{1}{2}, \frac{1}{2}]^2 \\ 0.1 & \text{otherwise} \end{cases}.$$

If observation noise is heteroscedastic, in considering *posterior* rather than *prior* uncertainty, ITL avoids points with high aleatoric uncertainty, which accelerates learning.

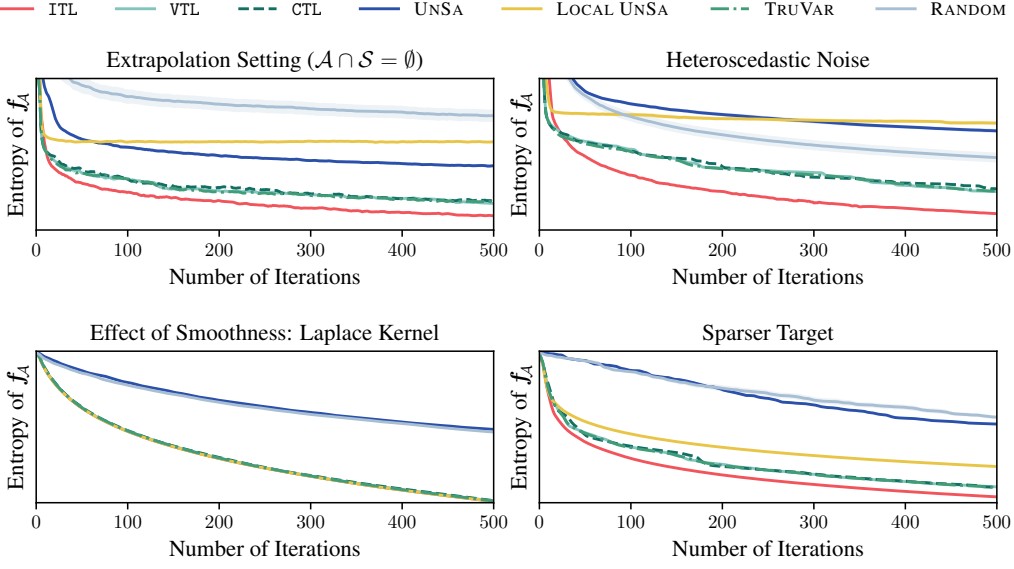

Figure 6: Additional GP experiments

3. *Effect of Smoothness*: Experiment from Figure 3 under the Laplace kernel. All algorithms except for US and RANDOM perform equally well. This validates our claims from Section 3.3: in the extreme non-smooth case of a Laplace kernel and $\mathcal{A} \subseteq \mathcal{S}$, points outside $\mathcal{A}$ do not provide any additional information, and ITL and "local" UNSA coincide.

4. *Sparser Target*: Experiment from Figure 3 under the Gaussian kernel, but with domain extended to $\mathcal{X} = [-10, 10]^2$.

**Hyperparameters of TRUVAR** As suggested by Bogunovic et al. (2016), we use $\tilde{\eta}^2_{(1)} = 1$, $r = 0.1$, and $\delta = 0$ (even though the theory only holds for $\delta > 0$). The TRUVAR baseline only applies when $\mathcal{A} \subseteq \mathcal{S}$ (cf. Section 6).

**Smoothing to reduce numerical noise** Applied running average with window 5 to entropy curves of Figures 2 and 6 to smoothen out numerical noise.

# I  Alternative Settings for Active Fine-Tuning

In our main experiments, we consider the setting $\mathcal{A} \cap \mathcal{S} = \emptyset$, i.e., the prediction targets cannot be used for fine-tuning since their labels are not known. This setting is particularly relevant for practical applications where the model is fine-tuned dynamically at test time to each prediction target (or a small set of prediction target). Put differently, in this "transductive" setting, extrapolation to new prediction targets happens at *test-time* with knowledge of the prediction target(s). This is in contrast to a more traditional "inductive" setting, where extrapolation happens at *train-time* without knowledge of the concrete prediction targets, but under the assumption of samples from (or knowledge of) the target distribution. In the following, we briefly survey two settings motivated from an "inductive" perspective.

## I.1  Prediction Targets are Contained in Sample Space: $\mathcal{A} \subseteq \mathcal{S}$

If labels can be obtained cheaply, one can also fine-tune on the prediction targets directly, i.e., $\mathcal{A} \subseteq \mathcal{S}$. Note, however, that the set $\mathcal{A}$ is still assumed to be small (e.g., $|\mathcal{A}| = 100$ in the CIFAR-100 experiment). We perform an experiment in this setting and report the results in Figure 7. The experiment shows that — similarly to the GP experiment from Figure 2 — there can be *additional value* in fine-tuning the model on relevant data selected from $\mathcal{S}$ beyond simply fine-tuning the model on $\mathcal{A}$.

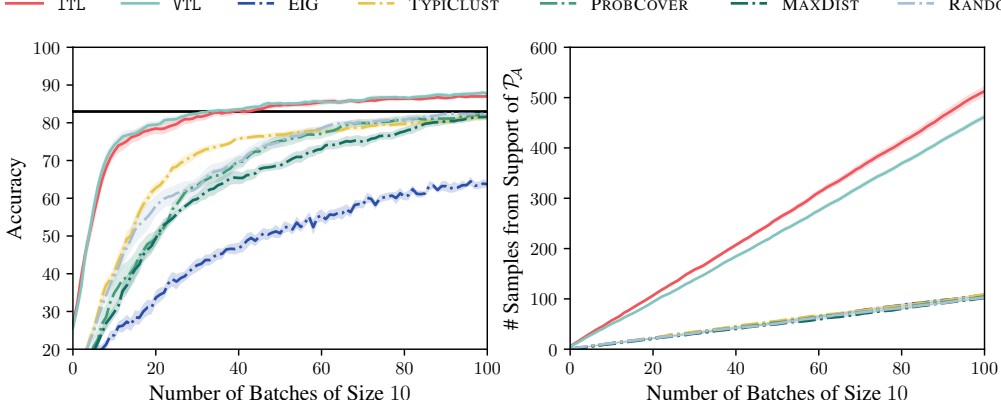

Figure 7: Evaluation of CIFAR-100 experiment in the setting $\mathcal{A} \subseteq \mathcal{S}$, i.e., one can also sample from the 100 prediction targets $\mathcal{A}$. The solid black line denotes the performance of the model fine-tuned on all of $\mathcal{A}$. This experiment shows that there is *additional value* in fine-tuning the model on relevant data from $\mathcal{S}$ beyond simply fine-tuning the model on $\mathcal{A}$. The baselines are summarized in Appendix J.5

## I.2 Active Domain Adaptation

Active DA (Rai et al., 2010; Saha et al., 2011; Berlind & Urner, 2015) studies the problem of selecting the most informative samples from a (large) target domain $\mathcal{A}$, given a model trained on a source domain $\mathcal{S}$. This problem can be cast as an instance of transductive active learning with target space $\mathcal{A}$ and sample space $\mathcal{S}' = \mathcal{S} \cup \mathcal{A}$ where the model is already conditioned on all of $\mathcal{S}$. This is slightly different from the setting considered in Section 4 where $\mathcal{A}$ is small and not necessarily part of the sample space. We hypothesize that ITL behaves similarly to recent work on active DA (Su et al., 2020; Prabhu et al., 2021; Fu et al., 2021): querying informative and diverse samples from $\mathcal{A}$ that are dissimilar to $\mathcal{S}$. Evaluating ITL and VTL empirically in this setting is a promising direction for future work.

## J  Additional NN Experiments & Details

We outline the active fine-tuning of NNs in Algorithm 1.

---

**Algorithm 1** Active Fine-Tuning of NNs

---

   **Given:** initialized or pre-trained model $f$, *small* sample $A \sim \mathcal{P}_\mathcal{A}$
   initialize dataset $\mathcal{D} = \emptyset$
   **repeat**
      sample $S \sim \mathcal{P}_\mathcal{S}$
      subsample target space $A' \overset{\text{u.a.r.}}{\sim} A$
      initialize batch $B = \emptyset$
      compute kernel matrix $\boldsymbol{K}$ over domain $[S, A']$
      **repeat** $b$ **times**
         compute acquisition function w.r.t. $A'$, based on $\boldsymbol{K}$
         add maximizer $\boldsymbol{x} \in S$ of acquisition function to $B$
         update conditional kernel matrix $\boldsymbol{K}$
      obtain labels for $B$ and add to dataset $\mathcal{D}$
      update $f$ using data $\mathcal{D}$

---

In Appendix J.1, we detail metrics and hyperparameters. We describe in Appendices J.2 and J.3 how to compute the (initial) conditional kernel matrix $\boldsymbol{K}$, and in Appendix J.4 how to update this matrix $\boldsymbol{K}$ to obtain conditional embeddings for batch selection.

In Appendix J.5, we show that ITL and CTL significantly outperform a wide selection of commonly used heuristics. In Appendices J.6 and J.7, we conduct additional experiments and ablations.

Table 1: Hyperparameter summary of NN experiments. (*) we train until convergence on oracle validation accuracy.

|  | MNIST | CIFAR-100 |
|---|---|---|
| $\rho$ | 0.01 | 1 |
| $M$ | 30 | 100 |
| $m$ | 3 | 10 |
| $k$ | 1 000 | 1 000 |
| batch size $b$ | 1 | 10 |
| # of epochs | (*) | 5 |
| learning rate | 0.001 | 0.001 |

Hübotter et al. (2024) discusses additional motivation and related work that has previously studied active fine-tuning, but which has largely focused on the training algorithm rather than data selection.

## J.1 Experiment Details

We evaluate the accuracy with respect to $\mathcal{P}_\mathcal{A}$ using a Monte Carlo approximation with out-of-sample data:

$$\text{accuracy}(\widehat{\boldsymbol{\theta}}) \approx \mathbb{E}_{(\boldsymbol{x},y)\sim\mathcal{P}_\mathcal{A}} \mathbb{1}\{y = \arg\max_i f_i(\boldsymbol{x};\widehat{\boldsymbol{\theta}})\}.$$

We provide an overview of the hyperparameters used in our NN experiments in Table 1. The effect of noise standard deviation $\rho$ is small for all tested $\rho \in [1, 100]$ (cf. ablation study in Table 2).[15] $M$ denotes the size of the sample $A \sim \mathcal{P}_\mathcal{A}$. In each iteration, we select the target space $\mathcal{A} \leftarrow A'$ as a random subset of $m$ points from $A$.[16] We provide an ablation over $m$ in Appendix J.6.

During each iteration, we select the batch $B$ according to the decision rule from a random sample from $\mathcal{P}_\mathcal{S}$ of size $k$.[17]

Since we train the MNIST model from scratch, we train from random initialization until convergence on oracle validation accuracy.[18] We do this to stabilize the learning curves, and provide the least biased (due to the training algorithm) results. For CIFAR-100, we train for 5 epochs (starting from the previous iterations' model) which we found to be sufficient to obtain good performance.

We use the ADAM optimizer (Kingma & Ba, 2014). In our CIFAR-100 experiments, we use a pre-trained EfficientNet-B0 (Tan & Le, 2019), and fine-tune the final and penultimate layers. We freeze earlier layers to prevent overfitting to the "few-shot" training data.

To prevent numerical inaccuracies when computing the ITL objective, we optimize

$$\text{I}(\boldsymbol{y}_\mathcal{A}; y_{\boldsymbol{x}} \mid \mathcal{D}_{n-1}) = \frac{1}{2} \log\left(\frac{\text{Var}[y_{\boldsymbol{x}} \mid \mathcal{D}_{n-1}]}{\text{Var}[y_{\boldsymbol{x}} \mid \boldsymbol{y}_\mathcal{A}, \mathcal{D}_{n-1}]}\right) \tag{44}$$

instead of Equation (43), which amounts to adding $\rho^2$ to the diagonal of the covariance matrix before inversion. This appears to improve numerical stability, especially when using gradient embeddings.[19]

---

[15]We use a larger noise standard deviation $\rho$ in CIFAR-100 to stabilize the numerics of batch selection via conditional embeddings (cf. Table 2).

[16]This appears to improve the training, likely because it prevents overfitting to peculiarities in the finite sample $A$ (cf. Figure 16).

[17]In large-scale problems, the work of Coleman et al. (2022) suggests to use an (approximate) nearest neighbor search to select the (large) candidate set rather than sampling u.a.r. from $\mathcal{P}_\mathcal{S}$. This can be a viable alternative to simply increasing $k$ and suggests future work.

[18]That is, to stop training as soon as accuracy on a validation set from $\mathcal{P}_\mathcal{A}$ decreases in an epoch.

[19]In our experiments, we observe that the effect of various choices of $\rho$ on this slight adaptation of the ITL decision rule has negligible impact on performance. The more prominent effect of $\rho$ appears to arise from the batch selection via conditional embeddings (cf. Table 2).

In our experiments, we use last-layer neural tangent embeddings[20] and $\boldsymbol{\Sigma} = \boldsymbol{I}$ to evaluate ITL and VTL, and select inputs for labeling and training $f$. Notably, we use this linear Gaussian approximation of $f$ only to guide the active data selection and not for inference.

### J.2  Embeddings and Kernels

Using a neural network to parameterize $f$, we evaluate the canonical approximations of $f$ by a stochastic process in the following.

An embedding $\phi(\boldsymbol{x})$ is a latent representation of an input $\boldsymbol{x}$. Collecting the embeddings as rows in the design matrix $\boldsymbol{\Phi}$ of a set of inputs $X$, one can approximate the network by the linear function $\boldsymbol{f}_X = \boldsymbol{\Phi}\boldsymbol{\beta}$ with weights $\boldsymbol{\beta}$. Approximating the weights by $\boldsymbol{\beta} \sim \mathcal{N}(\boldsymbol{\mu}, \boldsymbol{\Sigma})$ implies that $\boldsymbol{f}_X \sim \mathcal{N}(\boldsymbol{\Phi}\boldsymbol{\mu}, \boldsymbol{\Phi}\boldsymbol{\Sigma}\boldsymbol{\Phi}^\top)$. The covariance matrix $\boldsymbol{K}_{XX} = \boldsymbol{\Phi}\boldsymbol{\Sigma}\boldsymbol{\Phi}^\top$ can be succinctly represented in terms of its associated kernel $k(\boldsymbol{x}, \boldsymbol{x}') = \phi(\boldsymbol{x})^\top \boldsymbol{\Sigma} \phi(\boldsymbol{x}')$. Here,

- $\phi(\boldsymbol{x})$ is the latent representation of $\boldsymbol{x}$, and
- $\boldsymbol{\Sigma}$ captures the dependencies in the latent space.

While any choice of embedding $\phi$ is possible, the following are common choices:

1. *Last-Layer*: A common choice for $\phi(\boldsymbol{x})$ is the representation of $\boldsymbol{x}$ from the penultimate layer of the neural network (Holzmüller et al., 2023). Interpreting the early layers as a feature encoder, this uses the low-dimensional feature map akin to random feature methods (Rahimi & Recht, 2007).

2. *Output Gradients (eNTK)*: Another common choice is $\phi(\boldsymbol{x}) = \boldsymbol{\nabla}_{\boldsymbol{\theta}} \boldsymbol{f}(\boldsymbol{x}; \boldsymbol{\theta})$ where $\boldsymbol{\theta}$ are the network parameters (Holzmüller et al., 2023). Its associated kernel is known as the *empirical neural tangent kernel* (eNTK) and the posterior mean of this GP approximates ultra-wide NNs trained with gradient descent (Jacot et al., 2018; Arora et al., 2019; Lee et al., 2019; Khan et al., 2019; He et al., 2020; Malladi et al., 2023). Kassraie & Krause (2022) derive bounds of $\gamma_n$ under this kernel. If $\boldsymbol{\theta}$ is restricted to the weights of the final linear layer, then this embedding is simply the last-layer embedding.

3. *Loss Gradients*: Another possible choice is
$$\phi(\boldsymbol{x}) = \boldsymbol{\nabla}_{\boldsymbol{\theta}} \, \ell(\boldsymbol{f}(\boldsymbol{x}; \boldsymbol{\theta}), \hat{y}(\boldsymbol{x}))|_{\boldsymbol{\theta}=\widehat{\boldsymbol{\theta}}}$$
where $\ell$ is a loss function, $\hat{y}(\boldsymbol{x})$ is the predicted label, and $\widehat{\boldsymbol{\theta}}$ are the current parameter estimates (Ash et al., 2020).

4. *Outputs (eNNGP)*: Another possible choice is $\phi(\boldsymbol{x}) = \boldsymbol{f}(\boldsymbol{x})$, i.e., the output of the network. Its associated kernel is known as the *empirical neural network Gaussian process* (eNNGP) kernel (Lee et al., 2018).

5. *Predictive* (Kirsch, 2023): Given a Bayesian neural network (Blundell et al., 2015) or probabilistic (deep) ensemble (Lakshminarayanan et al., 2017), which induce samples $\boldsymbol{\theta}_1, \dots, \boldsymbol{\theta}_K \sim p(\boldsymbol{\theta})$ from the distribution over network parameters, one can approximate the predictive covariance $k(\boldsymbol{x}, \boldsymbol{x}') = \mathrm{Cov}_{\boldsymbol{\theta}}[f(\boldsymbol{x}; \boldsymbol{\theta}), f(\boldsymbol{x}'; \boldsymbol{\theta})]$. This kernel measures proximity in the prediction space rather than parameter space and as such does not require gradient information. The corresponding feature map is $\phi(\boldsymbol{x}) = \frac{1}{\sqrt{K}}[\bar{f}(\boldsymbol{x}; \boldsymbol{\theta}_1) \; \cdots \; \bar{f}(\boldsymbol{x}; \boldsymbol{\theta}_K)]^\top$ where $\bar{f}(\boldsymbol{x}; \boldsymbol{\theta}_k) \stackrel{\text{def}}{=} f(\boldsymbol{x}; \boldsymbol{\theta}_k) - \frac{1}{K} \sum_{l=1}^K f(\boldsymbol{x}; \boldsymbol{\theta}_l)$.

In the additional experiments from this appendix we use last-layer embeddings unless noted otherwise. We compare the performance of last-layer and the loss gradient embedding

$$\phi(\boldsymbol{x}) = \boldsymbol{\nabla}_{\boldsymbol{\theta}'} \, \ell_{\mathrm{CE}}(\boldsymbol{f}(\boldsymbol{x}; \boldsymbol{\theta}), \hat{y}(\boldsymbol{x}))|_{\boldsymbol{\theta}=\widehat{\boldsymbol{\theta}}} \tag{45}$$

where $\boldsymbol{\theta}'$ are the parameters of the final output layer, $\widehat{\boldsymbol{\theta}}$ are the current parameter estimates, $\hat{y}(\boldsymbol{x}) = \arg\max_i f_i(\boldsymbol{x}; \widehat{\boldsymbol{\theta}})$ are the associated predicted labels, and $\ell_{\mathrm{CE}}$ denotes the cross-entropy loss. This gradient embedding captures the potential update direction upon observing a new point (Ash et al., 2020). Moreover, Ash et al. (2020) show that for most neural networks, the norm of these gradient embeddings are a conservative lower bound to the norm assumed by taking any other proxy label $\hat{y}(\boldsymbol{x})$. In Figure 8, we observe only negligible differences in performance between this and the last-layer embedding.

---

[20]We observe essentially the same performance with loss gradient embeddings, cf. Appendix J.2.

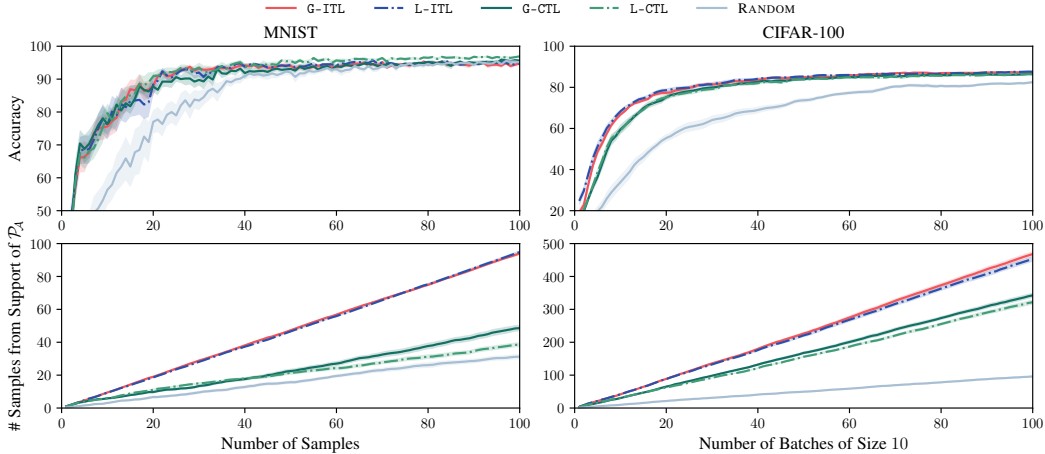

Figure 8: Comparison of loss gradient ("G-") and last-layer embeddings ("L-").

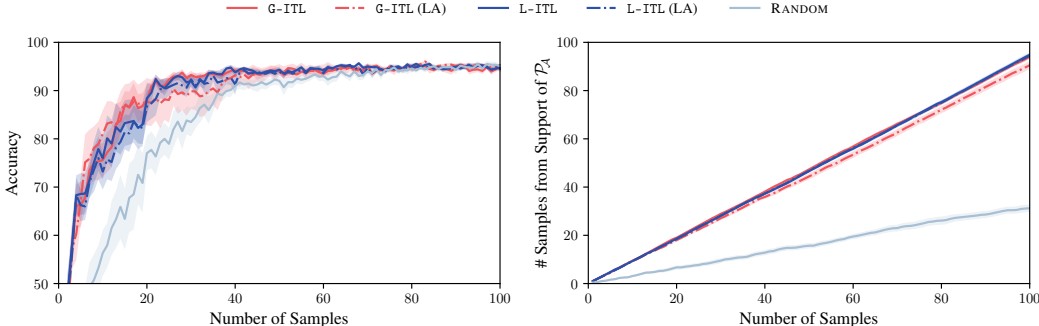

Figure 9: Uncertainty quantification (i.e., estimation of $\Sigma$) via a Laplace approximation (LA, Daxberger et al. (2021)) over last-layer weights using a Kronecker factored log-likelihood Hessian approximation (Martens & Grosse, 2015) and the loss gradient embeddings from Equation (45). The results are shown for the MNIST experiment. We do not observe a performance improvement beyond the trivial approximation $\Sigma = I$.

## J.3 Towards Uncertainty Quantification in Latent Space

A straightforward and common approximation of the uncertainty about NN weights is given by $\Sigma = I$, and we use this approximation throughout our experiments.

The poor performance of UNSA (cf. Appendix J.5) with this approximation suggests that with more sophisticated approximations, the performance of ITL, VTL, and CTL can be further improved. Further research is needed to study the effect of more sophisticated approximations of "uncertainty" in the latent space. For example, with parameter gradient embeddings, the latent space is the network parameter space where various approximations of $\Sigma$ based on Laplace approximation (Daxberger et al., 2021; Antorán et al., 2022), variational inference (Blundell et al., 2015), or Markov chain Monte Carlo (Maddox et al., 2019) have been studied. We also evaluate Laplace approximation (LA, Daxberger et al. (2021)) for estimating $\Sigma$ but see no improvement (cf. Figure 9). Nevertheless, we believe that uncertainty quantification is a promising direction for future work, with the potential to improve performance of ITL and its variations substantially.

## J.4 Batch Selection via Conditional Embeddings

We will refer to the greedy decision rule from Equation (3) as BACE, short for ***Ba**tch selection via **C**onditional **E**mbeddings*. BACE can be implemented efficiently using the Gaussian approximation of $f_X$ from Appendix J.2 by iteratively conditioning on the previously selected points $x_{n,1:i-1}$, and updating the kernel matrix $K_{XX}$ using the closed-form formula for the variance of conditional

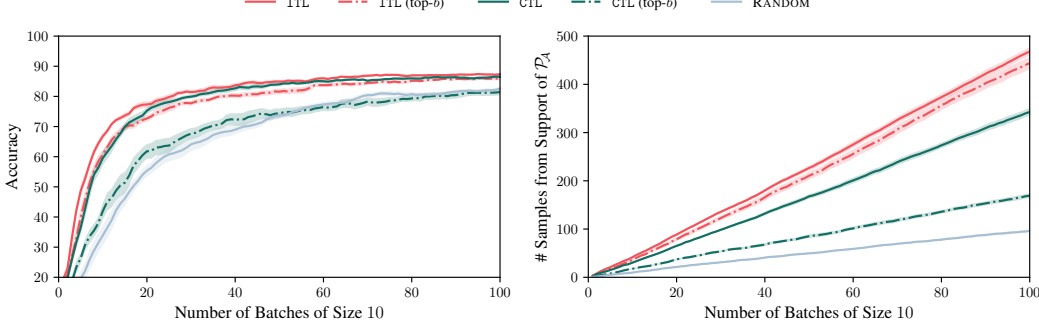

Figure 10: Advantage of batch selection via conditional embeddings over top-$b$ selection in the CIFAR-100 experiment.

Gaussians:

$$\boldsymbol{K}_{XX} \leftarrow \boldsymbol{K}_{XX} - \frac{1}{\boldsymbol{K}_{\boldsymbol{x}_j \boldsymbol{x}_j} + \rho^2} \boldsymbol{K}_{X\boldsymbol{x}_j} \boldsymbol{K}_{\boldsymbol{x}_j X} \tag{46}$$

where $j$ denotes the index of the selected $\boldsymbol{x}_{n,i}$ within $X$ and $\rho^2$ is the noise variance. Note that $\boldsymbol{K}_{\boldsymbol{x}_j \boldsymbol{x}_j}$ is a scalar and $\boldsymbol{K}_{X\boldsymbol{x}_j}$ is a row vector, and hence, this iterative update can be implemented efficiently.

We remark that Equation (3) is a natural extension of previous non-adaptive active learning methods, which typically maximize some notion of "distance" between points in the batch, to the "directed" setting (Ash et al., 2020; Zanette et al., 2021; Holzmüller et al., 2023; Pacchiano et al., 2024). BACE simultaneously maximizes "distance" between points in a batch and minimizes "distance" to points in $\mathcal{A}$.

**The sample efficiency of BACE**     $B_n$, and therefore also the greedily constructed $B_n'$ (which gives a constant-factor approximation with respect to the objective), yields diverse batches by design. In Figure 10, we compare BACE to selecting the top-$b$ points according to the decision rule (which does *not* yield diverse batches). We observe a significant improvement in accuracy and data retrieval when using BACE. We expect the gap between both approaches to widen further with larger batch sizes.

**Computational complexity of BACE**     As derived in Appendix G, a single batch selection step of BACE has complexity $O\big(b(k^3 + m^2)\big)$ where $b$ is the size of the batch, $k = |\mathcal{A}|$ is the size of the target space, and $m = |\mathcal{S}|$ is the size of the candidate set. In the case of large $m$, an alternative implementation whose runtime does not depend on $m$ is described in Appendix G.

### J.5 Baselines

In Figure 11, we compare against additional baselines:

- Both TYPICLUST (Hacohen et al., 2022) and PROBCOVER (Yehuda et al., 2022) are recent methods to select points that "cover" the data distribution well. To maintain comparability between algorithms, we use the same embeddings as for ITL which are re-computed before every new batch selection. ITL significantly outperforms TYPICLUST & PROBCOVER, which only attempt to cover $\mathcal{S}$ well without taking $\mathcal{A}$ into account (i.e., are "undirected").

- Mehta et al. (2022) introduced EIG for training neural classification models, which uses the same decision rule as ITL, but approximates the conditional entropy based on the networks' softmax output rather than using a GP approximation. We approximate the conditional entropy using a single gradient step of the hallucinated updates on the parameters of the final layer, as mentioned by Mehta et al. (2022). We observe that EIG is not competitive for batch-wise selection (CIFAR-100) since it does not encourage batch diversity. Moreover, we observe that EIG is orders of magnitude slower than ITL (since it has to compute $|\mathcal{S}| \cdot C$ individual gradient steps where $C$ is the number of classes). We note that since our datasets are balanced, the AEIG algorithm from Mehta et al. (2022) coincides with EIG.

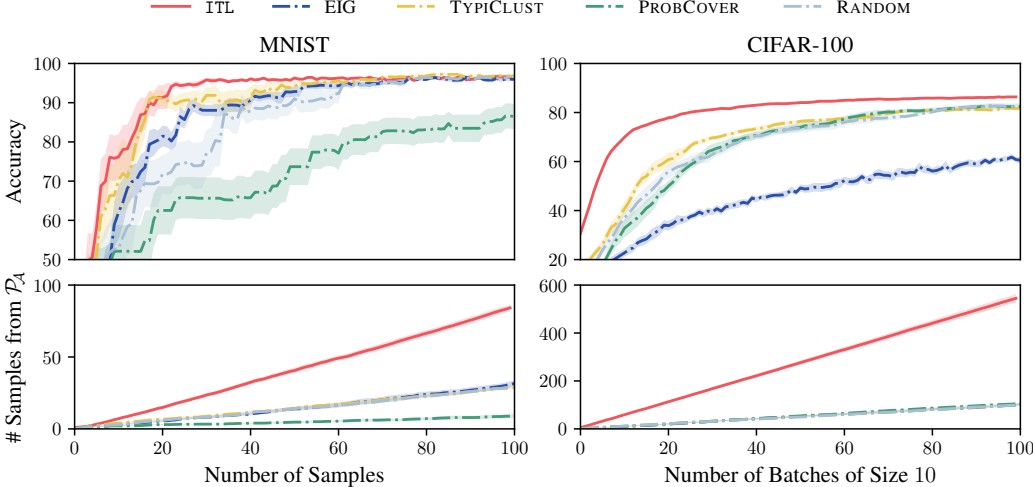

Figure 11: Comparison to baselines for the experiment of Figure 4.

Since, EIG does not have an open-source implementation, we implemented it ourselves following Mehta et al. (2022). For TYPICLUST & PROBCOVER, we use the author's implementation. In the figure, we show that ITL & VTL substantially outperform all baselines.

In the following, we briefly describe other commonly used "undirected" decision rules.

Denote the softmax distribution over labels $i$ at inputs $\boldsymbol{x}$ by

$$p_i(\boldsymbol{x}; \widehat{\boldsymbol{\theta}}) \propto \exp(f_i(\boldsymbol{x}; \widehat{\boldsymbol{\theta}})).$$

The following heuristics computed based on the softmax distribution aim to quantify the "uncertainty" about a particular input $\boldsymbol{x}$:

- MAXENTROPY (Settles & Craven, 2008):

$$\boldsymbol{x}_n = \underset{\boldsymbol{x} \in \mathcal{S}}{\arg\max} \, \mathrm{H}[p(\boldsymbol{x}; \widehat{\boldsymbol{\theta}}_{n-1})].$$

- MAXMARGIN (Scheffer et al., 2001; Settles & Craven, 2008):

$$\boldsymbol{x}_n = \underset{\boldsymbol{x} \in \mathcal{S}}{\arg\min} \, p_1(\boldsymbol{x}; \widehat{\boldsymbol{\theta}}_{n-1}) - p_2(\boldsymbol{x}; \widehat{\boldsymbol{\theta}}_{n-1})$$

  where $p_1$ and $p_2$ are the two largest class probabilities.

- LEASTCONFIDENCE (Lewis & Gale, 1994; Settles & Craven, 2008; Hendrycks & Gimpel, 2017; Tamkin et al., 2022):

$$\boldsymbol{x}_n = \underset{\boldsymbol{x} \in \mathcal{S}}{\arg\min} \, p_1(\boldsymbol{x}; \widehat{\boldsymbol{\theta}}_{n-1})$$

  where $p_1$ is the largest class probability.

An alternative class of decision rules aims to select diverse batches by maximizing the distances between points. Embeddings $\boldsymbol{\phi}(\boldsymbol{x})$ induce the (Euclidean) embedding distance

$$d_{\boldsymbol{\phi}}(\boldsymbol{x}, \boldsymbol{x}') \overset{\mathrm{def}}{=} \|\boldsymbol{\phi}(\boldsymbol{x}) - \boldsymbol{\phi}(\boldsymbol{x}')\|_2.$$

Similarly, a kernel $k$ induces the kernel distance

$$d_k(\boldsymbol{x}, \boldsymbol{x}') \overset{\mathrm{def}}{=} \sqrt{k(\boldsymbol{x}, \boldsymbol{x}) + k(\boldsymbol{x}', \boldsymbol{x}') - 2k(\boldsymbol{x}, \boldsymbol{x}')}.$$

It is straightforward to see that if $k(\boldsymbol{x}, \boldsymbol{x}') = \boldsymbol{\phi}(\boldsymbol{x})^\top \boldsymbol{\phi}(\boldsymbol{x}')$, then embedding and kernel distances coincide, i.e., $d_{\boldsymbol{\phi}}(\boldsymbol{x}, \boldsymbol{x}') = d_k(\boldsymbol{x}, \boldsymbol{x}')$.

- MAXDIST (Holzmüller et al., 2023; Yu & Kim, 2010; Sener & Savarese, 2017; Geifman & El-Yaniv, 2017) constructs the batch by choosing the point with the maximum distance to the nearest previously selected point:

$$\boldsymbol{x}_n = \arg\max_{\boldsymbol{x}\in\mathcal{S}} \min_{i<n} d(\boldsymbol{x}, \boldsymbol{x}_i)$$

- Similarly, K-MEANS++ (Holzmüller et al., 2023) selects the batch via K-MEANS++ seeding (Arthur et al., 2007; Ostrovsky et al., 2013). That is, the first centroid $\boldsymbol{x}_1$ is chosen uniformly at random and the subsequent centroids are chosen with a probability proportional to the square of the distance to the nearest previously selected centroid:

$$\mathbb{P}(\boldsymbol{x}_n = \boldsymbol{x}) \propto \min_{i<n} d(\boldsymbol{x}, \boldsymbol{x}_i)^2.$$

When using the loss gradient embeddings from Equation (45), this decision rule is known as BADGE (Ash et al., 2020).

Finally, we summarize common kernel-based decision rules.

- UNDIRECTED ITL chooses

$$\begin{aligned} \boldsymbol{x}_n &= \arg\max_{\boldsymbol{x}\in\mathcal{S}} \mathrm{I}(\boldsymbol{f}_{\mathcal{S}}; y_{\boldsymbol{x}} \mid \mathcal{D}_{n-1}) \\ &= \arg\max_{\boldsymbol{x}\in\mathcal{S}} \mathrm{I}(f_{\boldsymbol{x}}; y_{\boldsymbol{x}} \mid \mathcal{D}_{n-1}) \,. \end{aligned}$$

This can be shown to be equivalent to MAXDET (Holzmüller et al., 2023) which selects

$$\boldsymbol{x}_n = \arg\max_{\boldsymbol{x}\in\mathcal{S}} \left| \boldsymbol{K}_{\boldsymbol{x}} + \sigma^2 \boldsymbol{I} \right|$$

where $\boldsymbol{K}_{\boldsymbol{x}}$ denotes the kernel matrix over $\boldsymbol{x}_{1:n-1}\cup\{\boldsymbol{x}\}$, conditioned on the prior observations $\mathcal{D}_{n-1}$.

- UNSA (Lewis & Catlett, 1994) which with embeddings $\boldsymbol{\phi}_{n-1}$ after round $n-1$ corresponds to:

$$\boldsymbol{x}_n = \arg\max_{\boldsymbol{x}\in\mathcal{S}} \sigma_{n-1}^2(\boldsymbol{x}) = \arg\max_{\boldsymbol{x}\in\mathcal{S}} \left\| \boldsymbol{\phi}_{n-1}(\boldsymbol{x}) \right\|_2^2 \,.$$

With batch size $b = 1$, UNSA coincides with UNDIRECTED ITL. When evaluated with gradient embeddings, this acquisition function is similar to previously used "embedding length" or "gradient length" heuristics (Settles & Craven, 2008).

- UNDIRECTED VTL (Cohn, 1993) is the special case of VTL without specified prediction targets (i.e., $\mathcal{A} = \mathcal{S}$). In the literature, this decision rule is also known as BAIT (Holzmüller et al., 2023; Ash et al., 2021).

We compare to the abovementioned decision rules and summarize the results in Figure 12. We observe that most "undirected" decision rules perform worse (and often significantly so) than RANDOM. This is likely due to frequently selecting points from the support of $\mathcal{P}_{\mathcal{S}}$ which are not in the support of $\mathcal{P}_{\mathcal{A}}$ since the points are "adversarial examples" that the model $\widehat{\boldsymbol{\theta}}$ is not trained to perform well on. In the case of MNIST, the poor performance can also partially be attributed to the well-known "cold-start problem" (Gao et al., 2020).

In Figure 4, we also compare to the following "directed" decision rules:

- COSINESIMILARITY (Settles & Craven, 2008) selects $\boldsymbol{x}_n = \arg\max_{\boldsymbol{x}\in\mathcal{S}} \angle_{\boldsymbol{\phi}_{n-1}}(\boldsymbol{x}, \mathcal{A})$ where

$$\angle_{\boldsymbol{\phi}}(\boldsymbol{x}, \mathcal{A}) \stackrel{\text{def}}{=} \frac{1}{|\mathcal{A}|} \sum_{\boldsymbol{x}'\in\mathcal{A}} \frac{\boldsymbol{\phi}(\boldsymbol{x})^\top \boldsymbol{\phi}(\boldsymbol{x}')}{\|\boldsymbol{\phi}(\boldsymbol{x})\|_2 \|\boldsymbol{\phi}(\boldsymbol{x}')\|_2}.$$

- INFORMATIONDENSITY (Settles & Craven, 2008) is defined as the multiplicative combination of MAXENTROPY and COSINESIMILARITY:

$$\boldsymbol{x}_n = \arg\max_{\boldsymbol{x}\in\mathcal{S}} \mathrm{H}[p(\boldsymbol{x}; \widehat{\boldsymbol{\theta}}_{n-1})] \cdot \left( \angle_{\boldsymbol{\phi}_{n-1}}(\boldsymbol{x}, \mathcal{A}) \right)^\beta$$

where $\beta > 0$ controls the relative importance of both terms. We set $\beta = 1$ in our experiments.

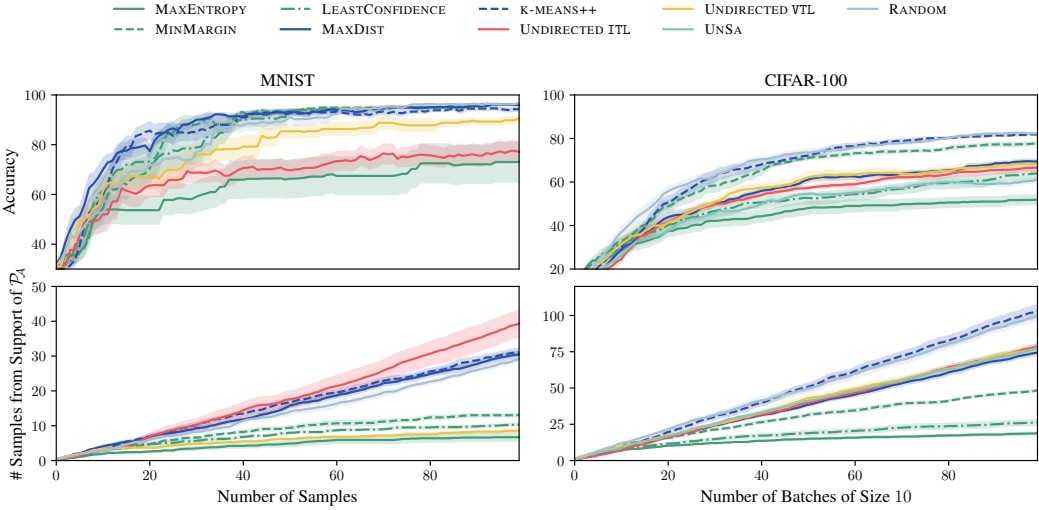

Figure 12: Comparison of "undirected" baselines for the experiment of Figure 4. In the MNIST experiment, UNSA and UNDIRECTED ITL coincide, and we therefore only plot the latter.

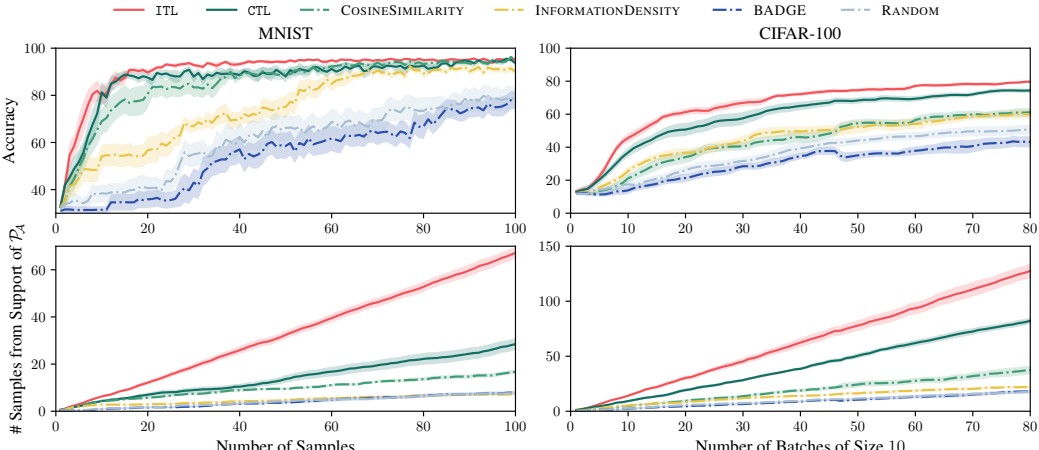

Figure 13: Imbalanced $\mathcal{P}_\mathcal{S}$ experiment.

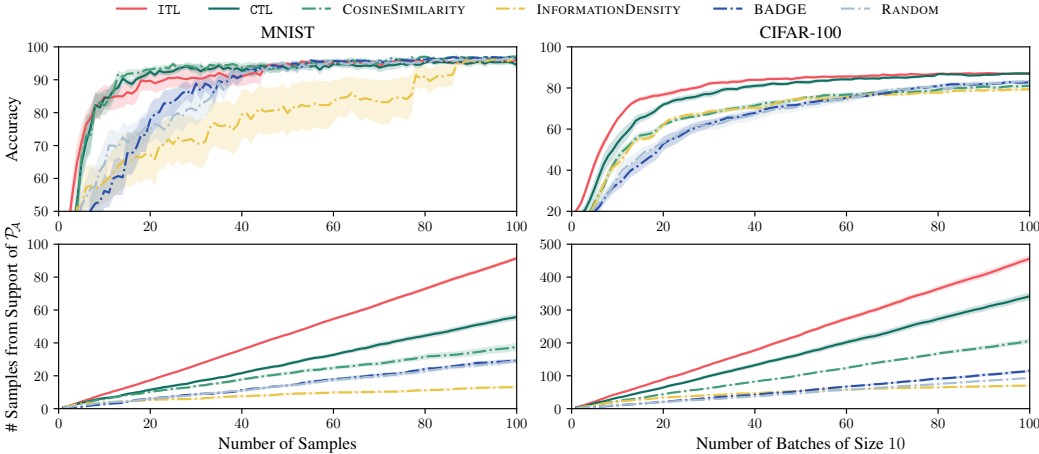

Figure 14: Imbalanced $A \sim \mathcal{P}_\mathcal{A}$ experiment.

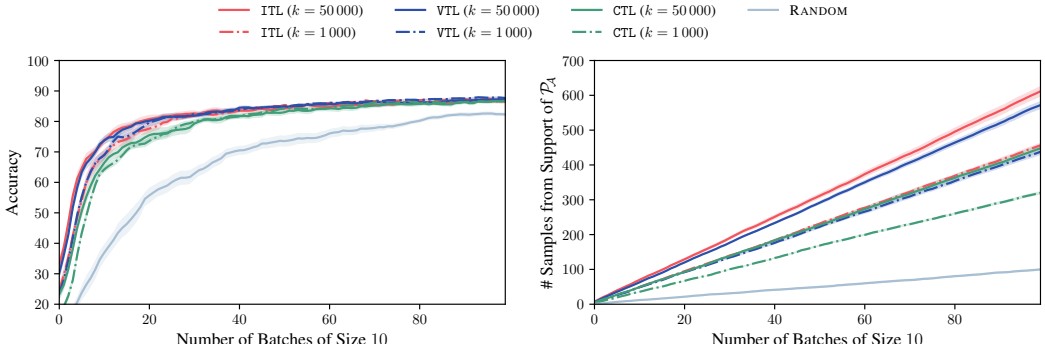

Figure 15: Performance of VTL & choice of $k$ in the CIFAR-100 experiment.

## J.6 Additional experiments

We conduct the following additional experiments:

1. *Imbalanced $\mathcal{P}_\mathcal{S}$* (Figure 13): We artificially remove $80\%$ of the support of $\mathcal{P}_\mathcal{A}$ from $\mathcal{P}_\mathcal{S}$. For example, in case of MNIST, we remove $80\%$ of the images with labels 3, 6, and 9 from $\mathcal{P}_\mathcal{S}$. This makes the learning task more difficult, as $\mathcal{P}_\mathcal{A}$ is less represented in $\mathcal{P}_\mathcal{S}$, meaning that the "targets" are more sparse. The trend of ITL outperforming CTL which outperforms RANDOM is even more pronounced in this setting.

2. *Imbalanced $A \sim \mathcal{P}_\mathcal{A}$* (Figure 14): We artificially remove $50\%$ of part of the support of $\mathcal{P}_\mathcal{A}$ while generating $A \sim \mathcal{P}_\mathcal{A}$ to evaluate the robustness of ITL and CTL in presence of an imbalanced target space $\mathcal{A}$. Concretely, in case of MNIST, we remove $50\%$ of the images with labels 3 and 6 from $A$. In case of CIFAR-100, we remove $50\%$ of the images with labels $\{0, \ldots, 4\}$ from $A$. We still observe the same trends as in the other experiments.

3. *VTL & choice of $k$* (Figure 15): We observe that VTL performs almost as well as ITL. Additionally, we evaluate the effect of the number of points $k$ at which the decision rule is evaluated. Not surprisingly, we observe that the performance of ITL, VTL, and CTL improves with larger $k$.

4. *Choice of $m$* (Figure 16): Next, we evaluate the choice of $m$, i.e., the size of the target space $\mathcal{A}$ relative to the number $M$ of candidate points $A \sim \mathcal{P}_\mathcal{A}$. We write $p = m/M$. We generally observe that a larger $p$ leads to better performance (with $p = 1$ being the best choice). However, it appears that a smaller $p$ can be beneficial with respect to accuracy when a large number of batches are selected. We believe that this may be because a smaller $p$ improves the diversity between selected batches.

5. *Choice of $M$* (Figure 17): Finally, we evaluate the choice of $M$, i.e., the size of $A \sim \mathcal{P}_\mathcal{A}$. Not surprisingly, we observe that the performance of ITL improves with larger $M$.

## J.7 Ablation study of noise standard deviation $\rho$

In Table 2, we evaluate the CIFAR-100 experiment with different noise standard deviations $\rho$. We observe that the performance of batch selection via conditional embeddings drops (mostly for the less numerically stable gradient embeddings) if $\rho$ is too small, since this leads to numerical inaccuracies when computing the conditional embeddings. Apart from this, the effect of $\rho$ is negligible.

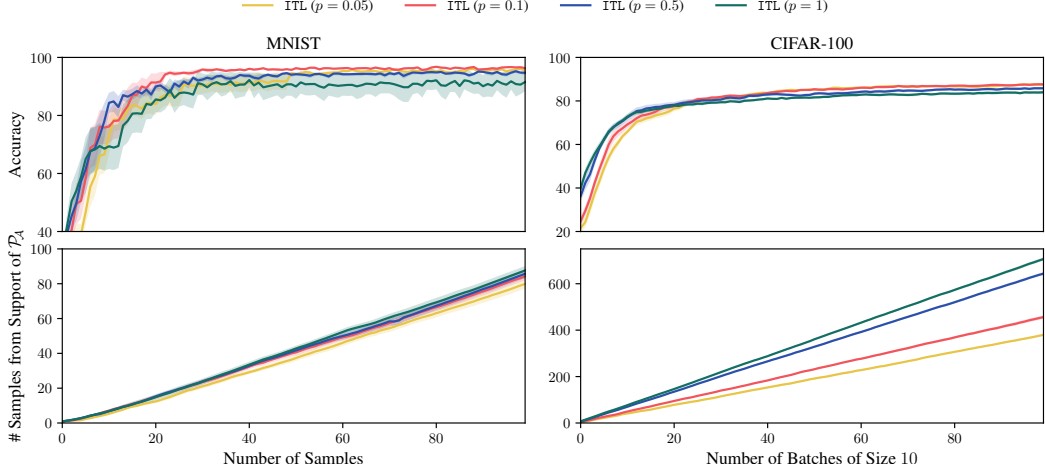

Figure 16: Evaluation of the choice of $m$ relative to the size $M$ of $A \sim \mathcal{P}_A$. Here, $p = m/M$.

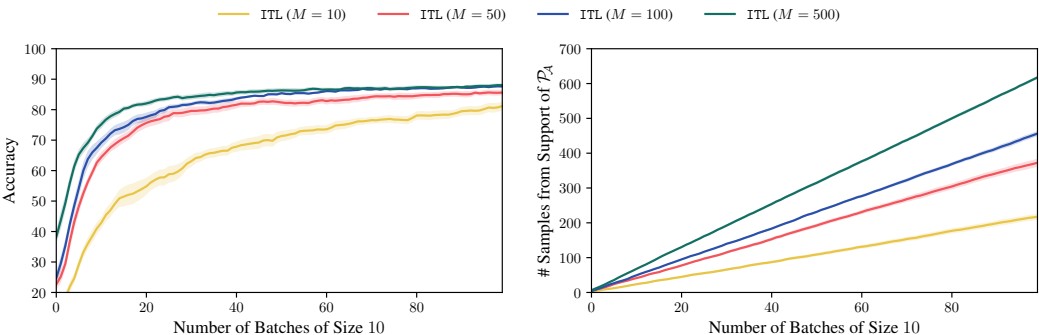

Figure 17: Evaluation of the choice of $M$, i.e., the size of $A \sim \mathcal{P}_A$, in the CIFAR-100 experiment.

Table 2: Ablation study of noise standard deviation $\rho$ in the CIFAR-100 experiment. We list the accuracy after 100 rounds per decision rule, with its standard error over 10 random seeds. "(top-$b$)" denotes variants where batches are selected by taking the top-$b$ points according to the decision rule rather than using batch selection via conditional embeddings. Shown in **bold** are the best performing decision rules, and shown in *italics* are results due to numerical instability.

| $\rho$ | 0.0001 | 0.01 | 1 | 100 |
|---|---|---|---|---|
| G-ITL | $78.26 \pm 1.40$ | $79.12 \pm 1.19$ | $\mathbf{87.16 \pm 0.29}$ | $\mathbf{87.18 \pm 0.28}$ |
| L-ITL | $\mathbf{87.52 \pm 0.48}$ | $\mathbf{87.52 \pm 0.41}$ | $\mathbf{87.53 \pm 0.35}$ | $86.47 \pm 0.22$ |
| G-CTL | $58.68 \pm 2.11$ | $81.44 \pm 1.04$ | $86.52 \pm 0.44$ | $\mathbf{86.92 \pm 0.56}$ |
| L-CTL | $\mathbf{86.40 \pm 0.71}$ | $\mathbf{86.38 \pm 0.75}$ | $86.00 \pm 0.69$ | $84.78 \pm 0.39$ |
| G-ITL (top-$b$) | $85.84 \pm 0.54$ | $85.92 \pm 0.52$ | $85.84 \pm 0.54$ | $85.55 \pm 0.46$ |
| L-ITL (top-$b$) | $85.44 \pm 0.58$ | $85.46 \pm 0.54$ | $85.44 \pm 0.59$ | $85.29 \pm 0.36$ |
| G-CTL (top-$b$) | $82.27 \pm 0.67$ | $82.27 \pm 0.67$ | $82.27 \pm 0.67$ | $82.27 \pm 0.67$ |
| L-CTL (top-$b$) | $80.73 \pm 0.68$ | $80.73 \pm 0.68$ | $80.73 \pm 0.68$ | $80.73 \pm 0.68$ |
| BADGE | $83.24 \pm 0.60$ | $83.24 \pm 0.60$ | $83.24 \pm 0.60$ | $83.24 \pm 0.60$ |
| INFORMATIONDENSITY | $79.24 \pm 0.51$ | $79.24 \pm 0.51$ | $79.24 \pm 0.51$ | $79.24 \pm 0.51$ |
| RANDOM | $82.49 \pm 0.66$ | $82.49 \pm 0.66$ | $82.49 \pm 0.66$ | $82.49 \pm 0.66$ |

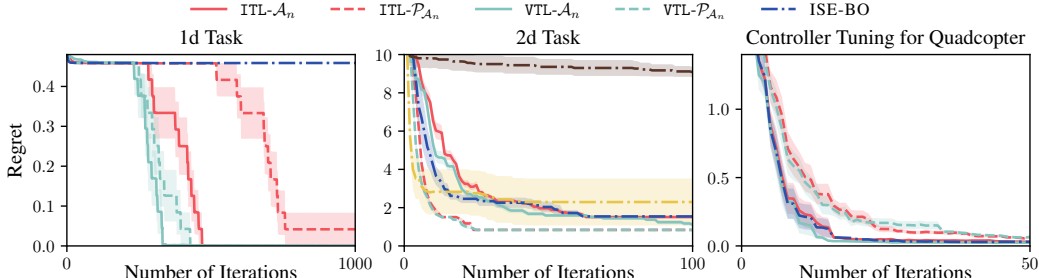

Figure 18: We perform the tasks of Figure 5 using Thompson sampling to evaluate the stochastic target space $\mathcal{P}_{\mathcal{A}n}$. We additionally compare to GOOSE (cf. Appendix K.2.3) and ISE-BO (cf. Appendix K.2.4).

## K    Additional Safe BO Experiments & Details

In Appendix K.1, we discuss the use of stochastic target spaces in the safe BO setting. We provide a comprehensive overview of prior works in Appendix K.2 and an additional experiment highlighting that ITL, unlike SAFEOPT, is able to "jump past local barriers" in Appendix K.3. In Appendix K.4, we provide details on the experiments from Figure 5.

### K.1    A More Exploitative Stochastic Target Space

Alternatively to the target space $\mathcal{A}_n$ which comprises all potentially optimal points, we evaluate the stochastic target space

$$\mathcal{P}_{\mathcal{A}n}(\cdot) = \mathbb{P}(\operatorname*{arg\,max}_{\boldsymbol{x} \in \mathcal{X}: g(\boldsymbol{x}) \geq 0} f(\boldsymbol{x}) = \cdot \mid \mathcal{D}_n) \tag{47}$$

which effectively weights points in $\mathcal{A}_n$ according to how likely they are to be the safe optimum, and is therefore more exploitative than the uniformly-weighted target space discussed so far. Samples from $\mathcal{P}_{\mathcal{A}n}$ can be obtained efficiently via Thompson sampling (Thompson, 1933; Russo et al., 2018). Observe that $\mathcal{P}_{\mathcal{A}n}$ is supported precisely on the set of potential maximizers $\mathcal{A}_n$. We provide a formal analysis of stochastic target spaces in Appendix E. Whether transductive active learning with $\mathcal{A}_n$ or $\mathcal{P}_{\mathcal{A}n}$ performs better is task-dependent, as we will see in the following.

Note that performing ITL with this target space is analogous to output-space entropy search (Wang & Jegelka, 2017). Samples from $\mathcal{P}_{\mathcal{A}n}$ can be obtained via Thompson sampling (Thompson, 1933; Russo et al., 2018). That is, in iteration $n + 1$, we sample $K \in \mathbb{N}$ independent functions $f^{(j)} \sim f \mid \mathcal{D}_n$ from the posterior distribution and select $K$ points $\boldsymbol{x}^{(1)}, \ldots, \boldsymbol{x}^{(K)}$ which are a safe maximum of $f^{(1)}, \ldots, f^{(K)}$, respectively.

**Experiments**    In Figure 18, we contrast the performance of ITL with $\mathcal{P}_{\mathcal{A}n}$ to the performance of ITL with the exact target space $\mathcal{A}_n$. We observe that their relative performance is instance dependent: in tasks that require more difficult expansion, ITL with $\mathcal{A}_n$ converges faster, whereas in simpler tasks (such as the 2d experiment), ITL with $\mathcal{P}_{\mathcal{A}n}$ converges faster. We compare against the GOOSE algorithm (Turchetta et al., 2019) which is a heuristic extension of SAFEOPT that explores more greedily in directions of (assumed) high reward (cf. Appendix K.2.3). GOOSE suffers from the same limitations as SAFEOPT, which were highlighted in Section 5, and additionally is limited by its heuristic approach to expansion which fails in the 1d task and safe controller tuning task. Analogously to our experiments with SAFEOPT, we also compare against ORACLE GOOSE which has oracle knowledge of the true Lipschitz constants.

The different behaviors of ITL with $\mathcal{A}_n$ and $\mathcal{P}_{\mathcal{A}n}$, respectively, as well as SAFEOPT and GOOSE are illustrated in Figure 19. We observe that ITL with $\mathcal{A}_n$ and SAFEOPT expand the safe set more "uniformly" since the set of potential maximizers encircles the true safe set.[21] Intuitively, this is because the set of potential maximizers *conservatively* captures migh points might be safe and

---

[21]This is because typically, there will always remain points in $\widehat{\mathcal{S}}_n \setminus \mathcal{S}_n$ of which the safety cannot be fully determined, and since, they cannot be observed, it can also not be ruled out that they have high objective value.

optimal. In contrast, ITL with $\mathcal{P}_{\mathcal{A}_n}$ and GoOSE focus exploration and expansion in those regions where the objective is likely to be high.

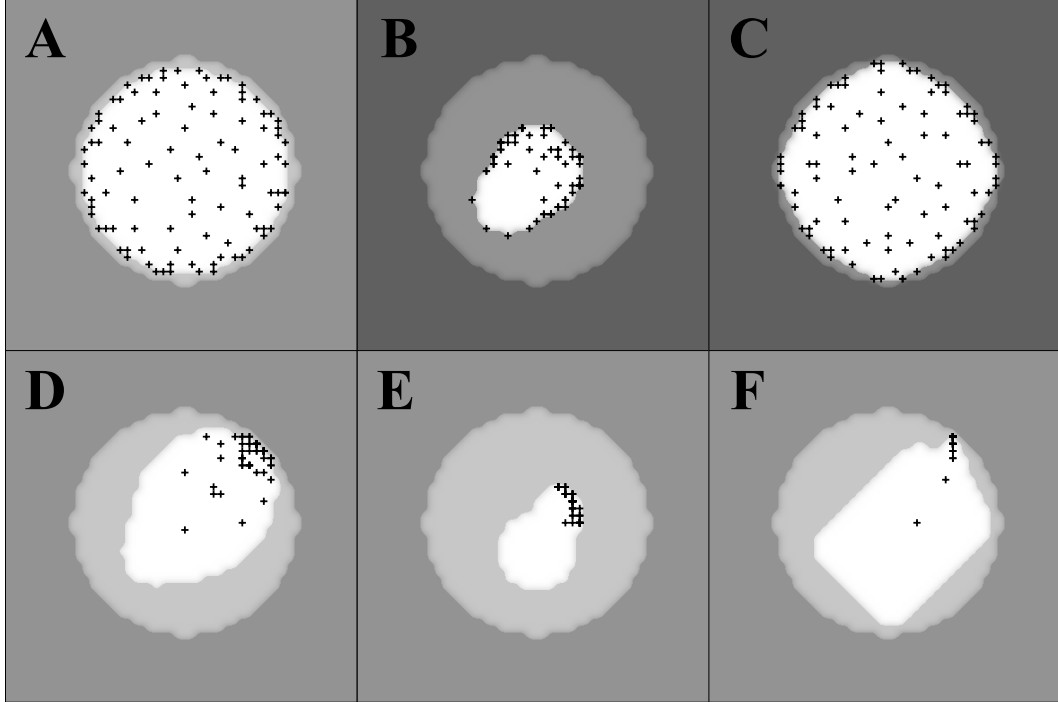

Figure 19: The first 100 samples of **(A)** ITL with $\mathcal{A}_n$, **(B)** SAFEOPT, **(C)** ORACLE SAFEOPT, **(D)** ITL with $\mathcal{P}_{\mathcal{A}_n}$, **(E)** GoOSE, **(F)** ORACLE GoOSE. The white region denotes the pessimistic safe set $\mathcal{S}_{100}$, the light gray region denotes the true safe set $\mathcal{S}^\star$ (i.e., the "island"), and the darker gray regions denotes unsafe points (i.e., the "ocean").

### K.2 Detailed Comparison with Prior Works

The most widely used method for Safe BO is SAFEOPT (Sui et al., 2015; Berkenkamp et al., 2021) which keeps track of separate candidate sets for expansion and exploration and uses UNSA to pick one of the candidates in each round. Treating expansion and exploration separately, sampling is directed towards expansion in *all* directions — even those that are known to be suboptimal. The safe set is expanded based on a Lipschitz constant of $g^\star$, which is assumed to be known. In most real-world settings, this constant is unknown and has to be estimated using the GP. This estimate is generally conservative and results in suboptimal performance. To this end, Berkenkamp et al. (2016) proposed HEURISTIC SAFEOPT which relies solely on the confidence intervals of $g$ to expand the safe set, but lacks convergence guarantees. More recently, Bottero et al. (2022) proposed ISE which queries parameters from $\mathcal{S}_n$ that yield the most "information" about the safety of another parameter in $\mathcal{X}$. Hence, ISE focuses solely on the expansion of the safe set $\mathcal{S}_n$ and does not take into account the objective $f$. In practice, this can lead to significantly worse performance on the simplest of problems (cf. Figure 5). In contrast, ITL balances expansion of and exploration within the safe set. Furthermore, ISE does not have known convergence guarantees of the kind of Theorem 5.1. In parallel independent work, Bottero et al. (2024) proposed a combination of ISE and max-value entropy search (Wang & Jegelka, 2017) for which they derive a similar guarantee to Theorem 5.1.[22] Similar to SAFEOPT, their method aims to expand the safe set in all directions including those that are known to be suboptimal. In contrast, ITL directs expansion only towards potentially optimal regions.

In the 1d task and quadcopter experiment (cf. Figure 5), we observe that SAFEOPT and even ORACLE SAFEOPT converge significantly slower than ITL to the safe optima. We believe this is due to their conservative Lipschitz-continuity/global smoothness-based expansion, as opposed to ITL's expansion,

---

[22]We provide an empirical evaluation in Appendix K.2.4.

which adapts to the local smoothness of the constraints. HEURISTIC SAFEOPT, which does not rely on the Lipschitz constant for expansion, does not efficiently expand the safe set due to its heuristic that only considers single-step expansion. This is especially the case for the 1d task. Furthermore, in the 2d task, we notice the suboptimality of ISE since it does not take into account the objective, and purely aims to expand the safe set. ITL, on the other hand, balances expansion and exploration.

### K.2.1 SAFEOPT

SAFEOPT (Sui et al., 2015; Berkenkamp et al., 2021) is a well-known algorithm for Safe BO.

**Lipschitz-based expansion** SAFEOPT expands the set of known-to-be safe points by assuming knowledge of an upper bound $L_i$ to the Lipschitz constant of the unknown constraints $g_i^\star$.[23] In each iteration, the (pessimistic) safe set $\mathcal{S}_n$ is updated to include all points which can be reached safely (with respect to the Lipschitz continuity) from a known-to-be-safe point $\boldsymbol{x} \in \mathcal{S}_n$. Formally,

$$\mathcal{S}_n^{\text{SAFEOPT}} \stackrel{\text{def}}{=} \bigcup_{\boldsymbol{x} \in \mathcal{S}_{n-1}^{\text{SAFEOPT}}} \{\boldsymbol{x}' \in \mathcal{X} \mid \tag{48}$$
$$l_{n,i}(\boldsymbol{x}) - L_i \|\boldsymbol{x} - \boldsymbol{x}'\|_2 \geq 0 \text{ for all } i \in \mathcal{I}_s\}.$$

The expansion of the safe set is illustrated in Figure 20.

We remark two main limitations of this approach. First, the Lipschitz constant is an additional safety critical hyperparameter of the algorithm, which is typically not known. The RKHS assumption (cf. Assumption C.25) induces an assumption on the Lipschitz continuity, however, the worst-case a-priori Lipschitz constant is typically very large, and prohibitive for expansion. Second, the Lipschitz constant is global property of the unknown function, meaning that it does not adapt to the local smoothness. For example, a constraint may be "flat" in one direction (permitting straightforward expansion) and "steep" in another direction (requiring slow expansion). Furthermore, the Lipschitz constant is constant over time, whereas ITL is able to adapt to the local smoothness and reduce the (induced) Lipschitz constant over time.

**Undirected expansion** SAFEOPT addresses the trade-off between expansion and exploration by focusing learning on two different sets. First, the set of *maximizers*

$$\mathcal{M}_n^{\text{SAFEOPT}} \stackrel{\text{def}}{=} \{\boldsymbol{x} \in \mathcal{S}_n^{\text{SAFEOPT}} \mid u_{n,f}(\boldsymbol{x}) \geq \max_{\boldsymbol{x}' \in \mathcal{S}_n^{\text{SAFEOPT}}} l_{n,f}(\boldsymbol{x})\}$$

which contains all *known-to-be-safe* points which are potentially optimal. Note that if $\mathcal{S}_n^{\text{SAFEOPT}} = \mathcal{S}_n$ then $\mathcal{M}_n^{\text{SAFEOPT}} \subseteq \mathcal{A}_n$ since $\mathcal{A}_n$ contains points which are potentially optimal and potentially safe *but possibly unsafe*.

To facilitate expansion, for each point $\boldsymbol{x} \in \mathcal{S}_n$, the algorithm considers a set of *expanding points*

$$\mathcal{F}_n^{\text{SAFEOPT}}(\boldsymbol{x}) \stackrel{\text{def}}{=} \{\boldsymbol{x}' \in \mathcal{X} \setminus \mathcal{S}_n^{\text{SAFEOPT}} \mid u_{n,i}(\boldsymbol{x}) - L_i \|\boldsymbol{x} - \boldsymbol{x}'\|_2 \geq 0 \text{ for all } i \in \mathcal{I}_s\}$$

A point is expanding if it is unsafe initially and can be (optimistically) deduced as safe by observing $\boldsymbol{x}$. The set of *expanders* corresponds to all known-to-be-safe points which optimistically lead to expansion of the safe set:

$$\mathcal{G}_n^{\text{SAFEOPT}} \stackrel{\text{def}}{=} \{\boldsymbol{x} \in \mathcal{S}_n \mid |\mathcal{F}_n(\boldsymbol{x})| > 0\}.$$

That is, an expander is a safe point $\boldsymbol{x}$ which is "close" to at least one expanding point $\boldsymbol{x}'$. Observe that here, we start with a safe $\boldsymbol{x}$ and then find a close and potentially safe $\boldsymbol{x}'$ using the Lipschitz-property of the constraint function. Thus, the set of expanding points is inherently limited by the assumed Lipschitzness (cf. Figure 20), and generally a subset of the potential expanders $\mathcal{E}_n$ (cf. Equation (27)):

**Lemma K.1.** *For any $n \geq 0$, if $\mathcal{S}_n^{\text{SAFEOPT}} = \mathcal{S}_n$ then*

$$\bigcup_{\boldsymbol{x} \in \mathcal{S}_n} \mathcal{F}_n^{\text{SAFEOPT}}(\boldsymbol{x}) \subseteq \mathcal{E}_n.$$

---

[23]Recall that due to the assumption that $\|g_i^\star\|_k < \infty$, $g_i^\star$ is indeed Lipschitz continuous.

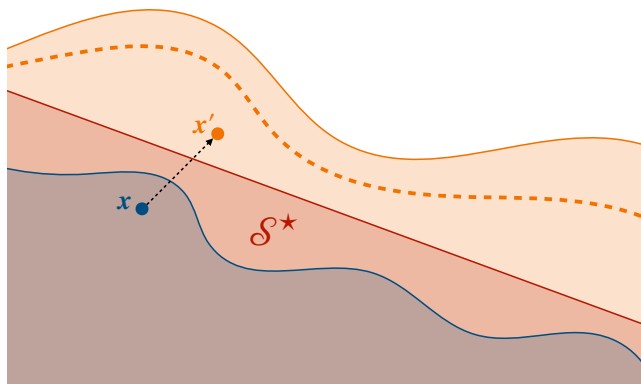

Figure 20: Illustration of the expansion of the safe set à la SAFEOPT. Here, the blue region denotes the pessimistic safe set $\mathcal{S}$, the red region denotes the true safe set $\mathcal{S}^\star$, and the orange region denotes the optimistic safe set $\widehat{\mathcal{S}}$. Whereas ITL learns about the point $\boldsymbol{x}'$ *directly*, SAFEOPT expands the safe set using the reduction of uncertainty at $\boldsymbol{x}$, and then extrapolating using the Lipschitz constant (cf. Equation (48)). The dashed orange line denotes the expanding points of SAFEOPT which under-approximate the optimistic safe set of ITL (cf. Lemma K.1). Thus, ITL may even learn about points in $\widehat{\mathcal{S}}$ which are "out of reach" for SAFEOPT.

*Proof.* Without loss of generality, we consider the case where $\mathcal{I}_s = \{i\}$. We have

$$\mathcal{E}_n = \widehat{\mathcal{S}}_n \setminus \mathcal{S}_n = \{\boldsymbol{x} \in \mathcal{X} \setminus \mathcal{S}_n \mid u_{n,i}(\boldsymbol{x}) \geq 0\}.$$

The result follows directly by observing that $L_i \|\boldsymbol{x} - \boldsymbol{x}'\|_2 \geq 0$. $\qquad\square$

SAFEOPT then selects $\boldsymbol{x}_{n+1}$ according to uncertainty sampling *within* the maximizers and expanders: $\mathcal{M}_n^{\text{SAFEOPT}} \cup \mathcal{G}_n^{\text{SAFEOPT}}$. We remark that due to the separate handling of expansion and exploration, SAFEOPT expands the safe set in *all* directions — even those that are known to be suboptimal. In contrast, ITL only expands the safe set in directions that are potentially optimal by balancing expansion and exploration through the single set of potential maximizers $\mathcal{A}_n$.

**Based on uncertainty sampling**  As mentioned in the previous paragraph, SAFEOPT selects as next point the maximizer/expander with the largest prior uncertainty.[24] In contrast, ITL selects the point within $\mathcal{S}_n$ which minimizes the posterior uncertainty within $\mathcal{A}_n$. Note that the two approaches are not identical as typically $\mathcal{M}_n^{\text{SAFEOPT}} \cup \mathcal{G}_n^{\text{SAFEOPT}} \subset \mathcal{S}_n^{\text{SAFEOPT}}$ and $\mathcal{A}_n \not\supseteq \mathcal{S}_n$.

We show empirically in Section 3.3 that depending on the kernel choice (i.e., the smoothness assumptions), uncertainty sampling within a given target space neglects higher-order information that can be attained by sampling outside the set. This can be seen even more clearly when considering linear functions, in which case points outside the maximizers and expanders can be equally informative as points inside.

Finally, note that the set of expanders is constructed "greedily", i.e., only considering *single-step* expansion. This is necessitated as the inference of safety is based on single reference points. Instead, ITL directly quantifies the information gained towards the points of interest without considering intermediate reference points.

**Requires homoscedastic noise**  SAFEOPT imposes a homoscedasticity assumption on the noise which is an artifact of the analysis of uncertainty sampling. It is well known that in the presence of heteroscedastic noise, one has to distinguish epistemic and aleatoric uncertainty. Uncertainty sampling fails because it may continuously sample a high variance point where the variance is dominated by aleatoric uncertainty, potentially missing out on reducing epistemic uncertainty at points with small aleatoric uncertainty. In contrast, maximizing mutual information naturally takes

---

[24]The use of uncertainty sampling for safe sequential decision-making goes back to Schreiter et al. (2015) and Sui et al. (2015).

into account the two sources of uncertainty, preferring those points where epistemic uncertainty is large and aleatoric uncertainty is small (cf. Appendix C.1).

**Suboptimal reachable safe set**  Sui et al. (2015) and Berkenkamp et al. (2021) show that SAFEOPT converges to the optimum within the closure $\bar{\mathcal{R}}_\epsilon^{\text{SAFEOPT}}(\mathcal{S}_0)$ of

$$\mathcal{R}_\epsilon^{\text{SAFEOPT}}(\mathcal{S}) \stackrel{\text{def}}{=} \mathcal{S} \cup \{\boldsymbol{x} \in \mathcal{X} \mid \exists \boldsymbol{x}' \in \mathcal{S} \text{ such that}$$
$$f_i^\star(\boldsymbol{x}') - (L_i\|\boldsymbol{x} - \boldsymbol{x}'\|_2 + \epsilon) \geq 0 \text{ for all } i \in \mathcal{I}_s\}.$$

Note that analogously to the expansion of the safe set, the "expansion" of the reachable safe set is based on "inferring safety" through a reference point in $\mathcal{S}$ and using Lipschitz continuity. This is opposed to the reachable safe set of ITL (cf. Definition C.29).

We remark that under the additional assumption that a Lipschitz constant is known, ITL can easily be extended to expand its safe set based on the kernel *and* the Lipschitz constant, resulting in a strictly larger reachable safe set than SAFEOPT. We leave the concrete formalization of this extension to future work. Moreover, we do not evaluate this extension in our experiments, as we observe that even without the additional assumption of a Lipschitz constant, ITL outperforms SAFEOPT in practice.

### K.2.2  HEURISTIC SAFEOPT

Berkenkamp et al. (2016) also implement a heuristic variant of SAFEOPT which does not assume a known Lipschitz constant. This heuristic variant uses the same (pessimistic) safe sets $\mathcal{S}_n$ as ITL. The set of maximizers is identical to SAFEOPT. As expanders, the heuristic variant considers all safe points $\boldsymbol{x} \in \mathcal{S}_n$ that if $\boldsymbol{x}$ were to be observed next with value $\boldsymbol{u}_n(\boldsymbol{x})$ lead to $|\mathcal{S}_{n+1}| > |\mathcal{S}_n|$. We refer to this set as $\mathcal{G}_n^{\text{H-SAFEOPT}}$. The next point is then selected by uncertainty sampling within $\mathcal{M}_n^{\text{SAFEOPT}} \cup \mathcal{G}_n^{\text{H-SAFEOPT}}$.

The heuristic variant shares some properties with SAFEOPT, such that it is based on uncertainty sampling, not adapting to heteroscedastic noise, and separate notions of maximizers and expanders (leading to an "undirected" expansion of the safe set). Note that there are no known convergence guarantees for heuristic SAFEOPT. Importantly, note that similar to SAFEOPT the set of expanders is constructed "greedily", and in particular, does only take into account *single-step* expansion. In contrast, an objective such as ITL which quantifies the "information gained towards expansion" also actively seeks out *multi-step* expansion.

### K.2.3  GOOSE

To address the "undirected" expansion of SAFEOPT discussed in the previous section, Turchetta et al. (2019) proposed *goal-oriented safe exploration* (GOOSE). GOOSE extends any unsafe BO algorithm (which we subsequently call an oracle) to the safe setting. In our experiments, we evaluate GOOSE-UCB which uses UCB as oracle and which is also the variant studied by Turchetta et al. (2019). In the following, we assume for ease of notation that $\mathcal{I}_s = \{c\}$.

Given the oracle proposal $\boldsymbol{x}^\star$, GOOSE first determines whether $\boldsymbol{x}^\star$ is safe. If $\boldsymbol{x}^\star$ is safe, $\boldsymbol{x}^\star$ is queried next. Otherwise, GOOSE first learns about the safety of $\boldsymbol{x}^\star$ by querying "expansionist" points until the oracle's proposal is determined to be either safe or unsafe.

GOOSE expands the safe set identically to SAFEOPT according to Equation (48). In the context of GOOSE, $\mathcal{S}_n^{\text{SAFEOPT}}$ is called the *pessimistic safe set*. To determine that a point cannot be deduced as safe, GOOSE also keeps track of a Lipschitz-based *optimistic safe set*:

$$\widehat{\mathcal{S}}_{n,\epsilon}^{\text{GOOSE}} \stackrel{\text{def}}{=} \bigcup_{\boldsymbol{x} \in \mathcal{S}_{n-1}^{\text{SAFEOPT}}} \{\boldsymbol{x}' \in \mathcal{X} \mid$$
$$u_{n,c}(\boldsymbol{x}) - L_c\|\boldsymbol{x} - \boldsymbol{x}'\|_2 - \epsilon \geq 0\}.$$

We summarize the algorithm in Algorithm 2 where we denote by $\mathcal{O}(\mathcal{X})$ the oracle proposal over the domain $\mathcal{X}$.

It remains to discuss the heuristic used to select the "expansionist" points. GOOSE considers all points $\boldsymbol{x} \in \mathcal{S}_n^{\text{SAFEOPT}}$ with confidence bands of size larger than the accuracy $\epsilon$, i.e.,

$$\mathcal{W}_{n,\epsilon}^{\text{GOOSE}} \stackrel{\text{def}}{=} \{\boldsymbol{x} \in \mathcal{S}_n^{\text{SAFEOPT}} \mid u_{n,c}(\boldsymbol{x}) - l_{n,c}(\boldsymbol{x}) > \epsilon\}.$$

**Algorithm 2** GOOSE

---

**Given:** Lipschitz constant $L_c$, prior model $\{f, g_c\}$, oracle $\mathcal{O}$, and precision $\epsilon$
Set initial safe set $\mathcal{S}_0^{\text{SAFEOPT}}$ based on prior
$\widehat{\mathcal{S}}_{n,\epsilon}^{\text{GOOSE}} \leftarrow \mathcal{X}$
$n \leftarrow 0$
**for** $k$ from 1 to $\infty$ **do**
$\quad \boldsymbol{x}_k^\star \leftarrow \mathcal{O}(\widehat{\mathcal{S}}_{n,\epsilon}^{\text{GOOSE}})$
$\quad$ **while** $\boldsymbol{x}_k^\star \notin \mathcal{S}_n^{\text{SAFEOPT}}$ **do**
$\quad\quad$ Observe "expansionist" point $\boldsymbol{x}_{n+1}$, set $n \leftarrow n + 1$, and update model and safe sets
$\quad$ **end while**
$\quad$ Observe $\boldsymbol{x}_k^\star$, set $n \leftarrow n + 1$, and update model and safe sets
**end for**

---

Which of the points in this set is evaluated depends on a set of learning targets $\mathcal{A}_{n,\epsilon}^{\text{GOOSE}} \stackrel{\text{def}}{=} \widehat{\mathcal{S}}_{n,\epsilon}^{\text{GOOSE}} \setminus \mathcal{S}_n^{\text{SAFEOPT}}$ akin to the "potential expanders" $\mathcal{E}_n$ (cf. Equation (27)), to each of which we assign a priority $h(\boldsymbol{x})$. When $h(\boldsymbol{x})$ is large, this indicates that the algorithm is prioritizing to determine whether $\boldsymbol{x}$ is safe. We use as heuristic the negative $\ell_1$-distance between $\boldsymbol{x}$ and $\boldsymbol{x}^\star$. GOOSE then considers the set of *potential immediate expanders*

$$\mathcal{G}_{n,\epsilon}^{\text{GOOSE}}(\alpha) \stackrel{\text{def}}{=} \{\boldsymbol{x} \in \mathcal{W}_{n,\epsilon}^{\text{GOOSE}} \mid \exists \boldsymbol{x}' \in \mathcal{A}_{n,\epsilon}^{\text{GOOSE}} \text{ with}$$
$$\text{priority } \alpha \text{ such that } u_{n,c}(\boldsymbol{x}) - L_c\|\boldsymbol{x} - \boldsymbol{x}'\|_2 \geq 0\}.$$

The "expansionist" point selected by GOOSE is then any point in $\mathcal{G}_{n,\epsilon}^{\text{GOOSE}}(\alpha^\star)$ where $\alpha^\star$ denotes the largest priority such that $|\mathcal{G}_{n,\epsilon}^{\text{GOOSE}}(\alpha^\star)| > 0$.

We observe empirically that the sample complexity of GOOSE is not always better than that of SAFEOPT. Notably, the expansion of the safe set is based on a "greedy" heuristic. Moreover, determining whether a single oracle proposal $\boldsymbol{x}^\star$ is safe may take significant time. Consider the (realistic) example where the prior is uniform, and UCB proposes a point which is far away from the safe set and suboptimal. GOOSE will typically attempt to derive the safety of the proposed point until the uncertainty at *all* points within $\mathcal{S}_0^{\text{SAFEOPT}}$ is reduced to $\epsilon$.[25] Thus, GOOSE can "waste" a significant number of samples, aiming to expand the safe set towards a known-to-be suboptimal point. In larger state spaces, due to the greedy nature of the expansion strategy, this can lead to GOOSE being effectively stuck at a suboptimal point for a significant number of rounds.

### K.2.4 ISE and ISE-BO

Recently, Bottero et al. (2022) proposed an information-theoretic approach to efficiently expand the safe set which they call *information-theoretic safe exploration* (ISE). Specifically, they choose the next action $\boldsymbol{x}_n$ by approximating

$$\arg\max_{\boldsymbol{x} \in \mathcal{S}_{n-1}} \underbrace{\max_{\boldsymbol{x}' \in \mathcal{X}} \mathrm{I}(\mathbb{1}\{g_{\boldsymbol{x}'} \geq 0\}; y_{\boldsymbol{x}} \mid \mathcal{D}_{n-1})}_{\alpha^{\text{ISE}}(\boldsymbol{x})}. \tag{ISE}$$

In a parallel independent work, Bottero et al. (2024) extended ISE to the Safe BO problem where they propose to choose $\boldsymbol{x}_n$ according to

$$\arg\max_{\boldsymbol{x} \in \mathcal{S}_{n-1}} \max\{\alpha^{\text{ISE}}(\boldsymbol{x}), \alpha^{\text{MES}}(\boldsymbol{x})\} \tag{ISE-BO}$$

where $\alpha^{\text{MES}}$ denotes the acquisition function of max-value entropy search (Wang & Jegelka, 2017). Similarly to SAFEOPT, ISE-BO treats expansion and exploration separately, which leads to "undirected" expansion of the safe set. That is, the safe set is expanded in all directions, even those that are known to be suboptimal. In contrast, ITL balances expansion and exploration through the single set of potential maximizers $\mathcal{A}_n$. With a stochastic target space, ITL generalizes max-value entropy search (cf. Appendix K.1).

We evaluate ISE-BO in Figure 18 and observe that it does not outperform ITL and VTL in any of the tasks, while performing poorly in the 1d task and suboptimally in the 2d task.

---

[25]This is because the proposed point typically remains in the optimistic safe set when it is sufficiently far away from the pessimistic safe set.

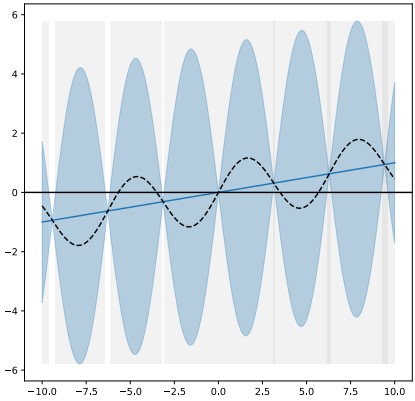

Figure 21: The ground truth $f^\star$ is shown as the dashed black line. The solid black line denotes the constraint boundary. The GP prior is given by a linear kernel with $\text{sin}$-transform and mean $0.1x$. The light gray region denotes the initial optimistic safe set $\widehat{\mathcal{S}}_0$ and the dark gray region denotes the initial pessimistic safe set $\mathcal{S}_0$.

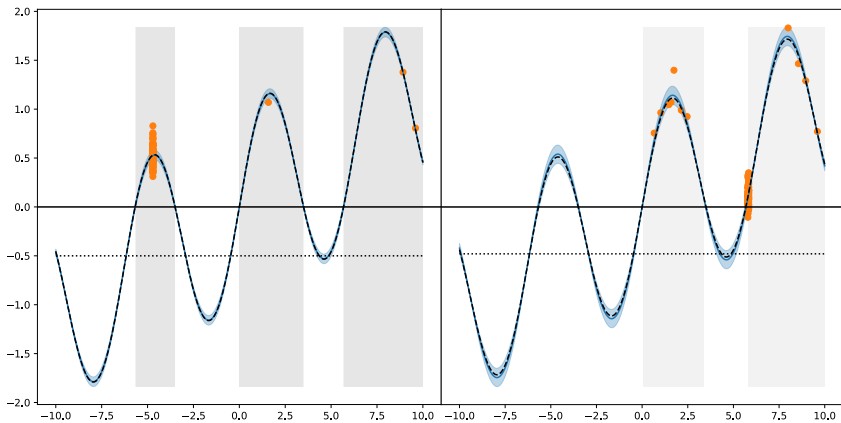

Figure 22: First 100 samples of ITL using the potential expanders $\mathcal{E}_n$ (cf. Equation (27)) as target space (left) and SAFEOPT sampling only from the set of expanders $\mathcal{G}_n^{\text{SAFEOPT}}$ (right).

## K.3 Jumping Past Local Barriers

In this additional experiment we demonstrate that ITL is able to extrapolate safety beyond local unsafe "barriers", which is a fundamental limitation of Lipschitz-based methods such as SAFEOPT. We consider the ground truth function and prior statistical model shown in Figure 21. Note that initially, there are three disjoint safe "regions" known to the algorithm corresponding to two of the three safe "bumps" of the ground truth function. In this experiment, the main challenge is to "jump past" the local barrier separating the leftmost and initially unknown safe "bump".

Figure 22 shows the sampled points during the first 100 iterations of SAFEOPT and ITL. Clearly, SAFEOPT does not discover the third safe "bump" while ITL does. Indeed, it is a fundamental limitation of Lipschitz-based methods that they can never "jump past local barriers", even if the oracle Lipschitz constant were to be known and tight (i.e., locally accurate) around the barrier. This is because Lipschitz-based methods expand to the point $\boldsymbol{x}$ based on a reference point $\boldsymbol{x}'$, and by definition, if $\boldsymbol{x}$ is added to the safe set so are all points on the line segment between $\boldsymbol{x}$ and $\boldsymbol{x}'$. Hence, if there is a single point on this line segment which is unsafe (i.e., a "barrier"), the algorithm will *never* expand past it. This limitation does not exist for kernel-based algorithms as expansion occurs in function space.

Moreover, note that for a non-stationary kernel such as in this example, ITL samples the "closest points" in function space rather than Euclidean space. We observe that SAFEOPT still samples "locally at the boundary" whereas ITL samples the most informative point which in this case is

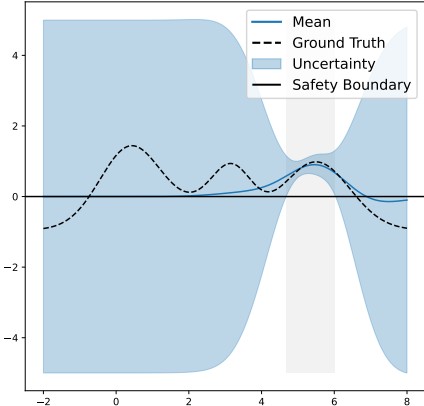

Figure 23: Ground truth and prior well-calibrated model in 1d synthetic experiment. The function serves simultaneously as objective and as constraint. The light gray region denotes the initial safe set $\mathcal{S}_0$.

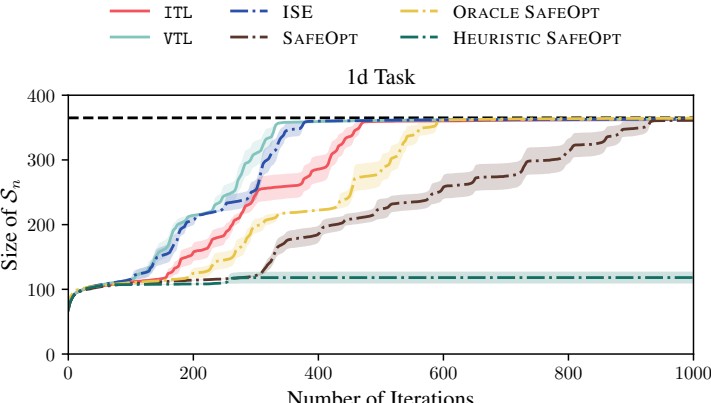

Figure 24: Size of $\mathcal{S}_n$ in 1d synthetic experiment. The dashed black line denotes the size of $\mathcal{S}^\star$. In this task, "discovering" the optimum is closely linked to expansion of the safe set, and HEURISTIC SAFEOPT fails since it does not expand the safe set sufficiently.

the local maximum of the sinusoidal function. In other words, ITL adapts to the geometry of the function. This generally leads us to believe that ITL is more capable to exploit (non-stationary) prior knowledge than distance-based methods such as SAFEOPT.

### K.4 Experiment Details

#### K.4.1 Synthetic Experiments

**1d task** Figure 23 shows the objective and constraint function, as well as the prior. We discretize using 500 points. The main difficulty in this experiment lies in sufficiently expanding the safe set to discover the global maximum. Figure 24 plots the size of the safe set $\mathcal{S}_n$ for the compared algorithms, which in this experiment matches the achieved regret closely.

**2d task** We model our constraint in the form of a spherical "island" where the goal is to get a good view of the coral reef located to the north-east of the island while staying in the interior of the island during exploration (cf. Figure 25). The precise objective and constraint functions are unknown to the agent. Hence, the agent has to gradually and safely update its belief about boundaries of the "island" and the location of the coral reef. The prior is obtained by a single observation within the center of the island $[-0.5, 0.5]^2$. We discretize using $2\,500$ points.

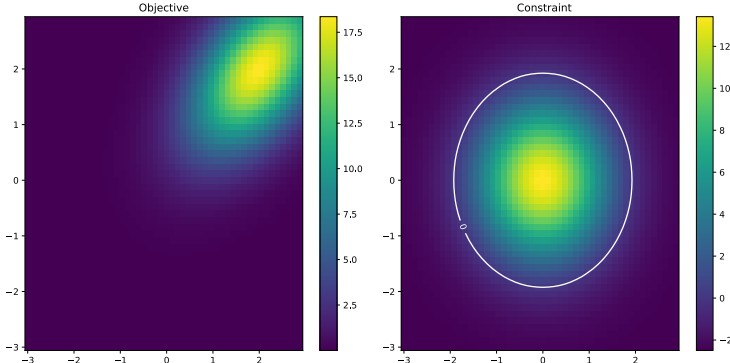

Figure 25: Ground truth in 2d synthetic experiment.

### K.4.2 Safe Controller Tuning for Quadcopter

**Modeling the real-world dynamics**  We learn a feedback policy (i.e., "control gains") to compensate for inaccuracies in the initial controller. In our experiment, we model the real world dynamics and the adjusted model using the PD control feedback (Widmer et al., 2023),

$$\boldsymbol{\delta}_t(\boldsymbol{x}) \stackrel{\text{def}}{=} (\boldsymbol{x}^\star - \boldsymbol{x})[(\boldsymbol{s}^\star - \boldsymbol{s}_t)\ (\dot{\boldsymbol{s}}^\star - \dot{\boldsymbol{s}}_t)], \tag{49}$$

where $\boldsymbol{x}^\star$ are the *unknown* ground truth disturbance parameters, and $\boldsymbol{s}^\star$ and $\dot{\boldsymbol{s}}^\star$ are the desired state and state derivative, respectively. This yields the following ground truth dynamics:

$$\boldsymbol{s}_{t+1}(\boldsymbol{x}) = \boldsymbol{T}(\boldsymbol{s}_t, \boldsymbol{u}_t + \boldsymbol{\delta}_t(\boldsymbol{x})). \tag{50}$$

The feedback parameters $\boldsymbol{x} = [\boldsymbol{x}_p\ \boldsymbol{x}_d]^\top$ can be split into $\boldsymbol{x}_p$ tuning the state difference which are called *proportional parameters* and $\boldsymbol{x}_d$ tuning the state derivative difference which are called *derivative parameters*. We use the "critical damping" heuristic to relate the proportional and derivative parameters: $\boldsymbol{x}_d = 2\sqrt{\boldsymbol{x}_p}$. We thus consider the restricted domain $\mathcal{X} = [0, 20]^4$ where each dimension corresponds to the proportional feedback to one of the four rotors.

Ground truth disturbance parameters are sampled from a chi-squared distribution with one degree of freedom (i.e., the square of a standard normal distribution), $\boldsymbol{x}_p^\star \sim \chi_1^2$, and $\boldsymbol{x}_d^\star$ is determined according to the critical damping heuristic.

**The learning problem**  The goal of our learning problem is to move the quadcopter from its initial position $\boldsymbol{s}(0) = [1\ 1\ 1]^\top$ (in Euclidean space with meter as unit) to position $\boldsymbol{s}^\star = [0\ 0\ 2]^\top$. Moreover, we aim to stabilize the quadcopter at the goal position, and therefore regularize the control signal towards an action $\boldsymbol{u}^\star$ which results in hovering (approximately) without any disturbances. We formalize these goals with the following objective function:

$$f^\star(\boldsymbol{x}) \stackrel{\text{def}}{=} -\sigma\left(\sum_{t=0}^T \|\boldsymbol{s}^\star - \boldsymbol{s}_t(\boldsymbol{x})\|_{\boldsymbol{Q}}^2 + \|\boldsymbol{u}^\star - \boldsymbol{u}_t(\boldsymbol{x})\|_{\boldsymbol{R}}^2\right) \tag{51}$$

where $\sigma(v) \stackrel{\text{def}}{=} \tanh((v - 100)/100)$ is used to smoothen the objective function and ensure that its range is $[-1, 1]$. The non-smoothed control objective in Equation (51) is known as a *linear-quadratic regulator* (LQR) which we solve exactly for the undisturbed system using ILQR (Tu et al., 2023). Finally, we want to ensure at all times that the quadcopter is at least $0.5$ meter above the ground, that is,

$$g^\star(\boldsymbol{x}) \stackrel{\text{def}}{=} \min_{t \in [T]} \boldsymbol{s}_t^z(\boldsymbol{x}) - 0.5 \tag{52}$$

where we denote by $\boldsymbol{s}_t^z$ the z-coordinate of state $\boldsymbol{s}_t$.

We use a time horizon of $T = 3$ seconds which we discretize using 100 steps. The objective is modeled by a zero-mean GP with a Matérn($\nu = 5/2$) kernel with lengthscale 0.1, and the constraint is modeled by a GP with mean $-0.5$ and a Matérn($\nu = 5/2$) kernel with lengthscale 0.1. The prior is obtained by a single observation of the "safe seed" $[0\ 0\ 0\ 10]^\top$.

**Adaptive discretization**   We discretize the domain $\mathcal{X}$ adaptively using coordinate LINEBO (Kirschner et al., 2019). That is, in each iteration, one of the four control dimensions is selected uniformly at random, and the active learning oracle is executed on the corresponding one-dimensional subspace.

**Safety**   Using the (unsafe) constrained BO algorithm EIC (Gardner et al., 2014) leads constraint violation,[26] while ITL and VTL do not violate the constraints during learning for any of the random seeds.

**Hyperparameters**   The observation noise is Gaussian with standard deviation $\rho = 0.1$. We let $\beta = 10$. The control target is $\boldsymbol{u}^\star = [1.766\ 0\ 0\ 0]^\top$.

The state space is 12-dimensional where the first three states correspond to the velocity of the quadcopter, the next three states correspond to its acceleration, the following three states correspond to its angular velocity, and the last three states correspond to its angular velocity in local frame. The LQR parameters are given by

$$\boldsymbol{Q} = \text{diag}\{1, 1, 1, 1, 1, 1, 0.1, 0.1, 0.1, 0.1, 0.1, 0.1\} \quad \text{and}$$
$$\boldsymbol{R} = 0.01 \cdot \text{diag}\{5, 0.8, 0.8, 0.3\}.$$

The quadcopter simulation was adapted from Chandra (2023).

Each one-dimensional subspace is discretized using $2\,000$ points.

**Random seeds**   We repeat the experiment for 25 different seeds where the randomness is over the ground truth disturbance, observation noise, and the randomness in the algorithm.

---

[26]On average, 1.6 iterations of the first 50 violate the constraints.

Table 3: Magnitudes of $\gamma_n$ for common kernels. The magnitudes hold under the assumption that $\mathcal{X}$ is compact. Here, $B_\nu$ is the modified Bessel function. We take the magnitudes from Theorem 5 of Srinivas et al. (2009) and Remark 2 of Vakili et al. (2021). The notation $\widetilde{O}(\cdot)$ subsumes log-factors. For $\nu = 1/2$, the Matérn kernel is equivalent to the Laplace kernel. For $\nu \to \infty$, the Matérn kernel is equivalent to the Gaussian kernel. The functions sampled from a Matérn kernel are $\lceil \nu \rceil - 1$ mean square differentiable. The kernel-agnostic bound follows by simple reduction to a linear kernel in $|\mathcal{X}|$ dimensions.

| Kernel | $k(\boldsymbol{x}, \boldsymbol{x'})$ | $\gamma_n$ |
|---|---|---|
| Linear | $\boldsymbol{x}^\top \boldsymbol{x'}$ | $O(d \log(n))$ |
| Gaussian | $\exp\left(-\frac{\|\boldsymbol{x}-\boldsymbol{x'}\|_2^2}{2h^2}\right)$ | $\widetilde{O}\left(\log^{d+1}(n)\right)$ |
| Laplace | $\exp\left(-\frac{\|\boldsymbol{x}-\boldsymbol{x'}\|_1}{h}\right)$ | $\widetilde{O}\left(n^{\frac{d}{1+d}} \log^{\frac{1}{1+d}}(n)\right)$ |
| Matérn | $\frac{2^{1-\nu}}{\Gamma(\nu)}\left(\frac{\sqrt{2\nu}\|\boldsymbol{x}-\boldsymbol{x'}\|_2}{h}\right)^\nu B_\nu\left(\frac{\sqrt{2\nu}\|\boldsymbol{x}-\boldsymbol{x'}\|_2}{h}\right)$ | $\widetilde{O}\left(n^{\frac{d}{2\nu+d}} \log^{\frac{2\nu}{2\nu+d}}(n)\right)$ |
| any | | $O(|\mathcal{X}| \log(n))$ |

