# OpenReview forum: "Transductive Active Learning: Theory and Applications"
_NeurIPS.cc/2024/Conference — NeurIPS 2024 poster_

### Official Review · Reviewer_yyUG · 2024-07-03

**Soundness:** 3
**Presentation:** 2
**Contribution:** 2
**Rating:** 5
**Confidence:** 2

**Summary:**

This paper considers active learning with Gaussian Process in a transductive setting where the learner wants to optimize the model performance on a target space A while it can only sample from a sample space S. It proposes a few algorithms that sequentially choose the example that minimizes a few variations of "posterior uncertainty" of the model on A. Assuming the measure for such uncertainty is submodular, it provides bounds on the convergence rate of the proposed algorithm. Empirically, it evaluates the proposed methods on two scenarios: fine-tuning and safe Bayesian optimization.

**Strengths:**

- This paper considers a practically relevant setup of active learning.

- The paper is mostly written clearly.

- The proposed methods are sound. It gives both theoretical guarantees and empirical evaluation.

**Weaknesses:**

- My major concern is novelty and significance:
  - The main idea of the proposed query strategy is to minimize posterior uncertainty. This is a well-known method in active learning, and it is an especially straightforward choice in a Bayesian setting.
  - For the theoretical results, I'm not very familiar with Gaussian Process literature, but I'm not convinced that the theoretical guarantees in Section 3.1 and 3.2 are nontrivial. It would help readers appreciate the results if the authors could clarify the challenges of obtaining such results, tightness of the bounds, how strong it is compared to other baseline sample strategies, or any new insights from such bounds.

--------
I've read author feedback. Now I agree that the theoretical part offers interesting techniques and results (which roughly matches known results in similar problems), but I personally think the results are still a bit unsatisfactory (specifically, the dependency on |S|, the RHS of the bound in Thm 3.3 can grow with n increases). In addition, since this paper claims the theory part is one of the major contributions, it would be better if there are more explanations, discussions, and comparisons for the results.

Apart from that, I still tend to think the proposed AL approach is a well-known idea in active learning and applying it to transductive learning is a bit natural, so the contribution of this part is not significant enough for a top conference. And I agree with reviewer aHJz that "few-shot fine-tuning of large neural networks" is a bit oversold.

**Questions:**

In addition to issues in the Weaknesses section, a few questions:

- For Section 4,
  - is there any existing Bayesian active learning algorithms that can be used as baselines in this setup?
  - it is mentioned in line 213-214 that "we will focus here on .. $A \cap S = \emptyset$", but in Figure 3 and line 243 it is comparing samples from $P_A$. How can the algorithm sample from $P_A$ if the sample space S does not intersect with A? What is the exact setup of the experiments?
  - How do you choose hyperparameters for the baselines?

- For section 5, I'm not familiar with this task, but in my opinion, it does not provide enough details and explanations. For example
  - What exactly is your sample strategy? In particular, how do you ensure the chosen samples are safe? In line 276 it says "a natural choice for the target space of safe BO is A_n", but it looks like A_n can be unsafe.
  - In line 317-318, "VTL minimizes marginal variances, .. which are decisive for expanding the safe set". Why?

---

> ### Author Rebuttal · Authors · 2024-08-06
>
> Thank you for reviewing our paper! Please find our detailed responses to the concerns raised.
> Please let us know if you have any further concerns or suggestions.
>
> ## Concerns
>
> > The main idea of the proposed query strategy is to minimize posterior uncertainty. This is a well-known method in active learning, and it is an especially straightforward choice in a Bayesian setting.
>
> Thank you for pointing this out.
> We do not claim to be the first to propose ITL or VTL (line 82 and lines 333-342).
> However, we are the first to show convergence guarantees.
> As we show in Section 3.3, on transductive instances (where $\mathcal{A} \not\subseteq \mathcal{S}$), our proposed methods differ substantially from the widely recognized inductive methods (e.g., uncertainty sampling) that have been derived based on the same idea of "minimizing posterior uncertainty".
> In our view, the fact that the proposed sample strategy is a natural choice can be seen as an advantage over baselines.
>
> > For the theoretical results, I'm not very familiar with Gaussian Process literature, but I'm not convinced that the theoretical guarantees in Section 3.1 and 3.2 are nontrivial. It would help readers appreciate the results if the authors could clarify the challenges of obtaining such results, tightness of the bounds, how strong it is compared to other baseline sample strategies, or any new insights from such bounds.
>
> None of the prior works studying ITL and VTL derive convergence guarantees related to Theorems 3.2 and 3.3.
> Similar convergence guarantees were only known for the special case where $\mathcal{A} = \mathcal{S}$.
>
> Core to our analysis are Theorems C.12 and C.13, where we use the approximate monotonicity of the marginal gains and the chain rule of information gain to derive a convergence rate of the marginal gains.
> The main technical challenge and novelty of our generalization bounds lies in the definition and proof of existence of "approximate Markov boundaries" (see Definition C.15 and Section C.6.1).
>
> An interesting insight from our bounds is that they shrink with $\gamma_{\mathcal{A}, \mathcal{S}}(n)$ (as opposed to $\gamma_n$ of inductive approaches).
> The information capacity $\gamma_{\mathcal{A}, \mathcal{S}}(n)$ of the target space can be *significantly* smaller than the total information capacity across a large domain $\mathcal{X}$.
> We observe this empirically in our experiments, where we show that inductive active learning methods are vastly less efficient than ITL/VTL when prediction targets are known.
>
> We will expand the discussion of the contributions and implications of our new theoretical results, given the additional available page in the camera-ready version.
>
> ## Questions
>
> 1. The most common (inductive) active learning baseline is "uncertainty sampling".
> We compare against this baseline in Section J.5 (Figure 11) alongside many additional baselines.
>
> 2. $\mathcal{P}\_{\mathcal{A}}$ is the data-distribution limited to the target labels (see footnote 5 on page 6).
> We say that a sample is in the support of $\mathcal{P}\_{\mathcal{A}}$ if its label is one of the target labels (e.g., in $\\\{0, \dots, 9\\\}$ in case of CIFAR-100).
> The sample space $\mathcal{S}$ includes images with such labels.
> However, the set of images in $\mathcal{A}$ is not in $\mathcal{S}$ (i.e., their labels are not known).
> Methods that sample many points from the support of $\mathcal{P}\_{\mathcal{A}}$ are able to retrieve the relevant data from $\mathcal{S}$.
> We discuss the relevance of this particular setting and also evaluate the alternative setting $\mathcal{A} \subseteq \mathcal{S}$ in Appendix I.
>
> 3. We discuss all baselines in detail and specify their hyperparameters where applicable in Appendix J.5.
> We use the author's implementations where applicable (e.g., for TypiClust and ProbCover).
> We have included all of our code required to reproduce the results in the supplementary material and intend to make it public upon publication.
>
> 4. We use the proposed methods for transductive active learning (ITL and VTL) with the specified target space $\mathcal{A}_n$ and sample space (i.e., pessimistic safe set) $\mathcal{S}_n$ (lines 277-278).
> You are absolutely right in pointing out that $\mathcal{A}_n$ can be unsafe.
> However, by definition, all of the sample space is safe with high probability.
> Extrapolating *outside* of $\mathcal{S}_n$ to those targets in $\mathcal{A}_n$ that are unsafe is precisely what we employ in our proof of Theorem 5.1.
> In this way, and unlike prior work, ITL \& VTL can address the expansion and exploration simultaneously.
>
> 5. VTL quantifies "uncertainty" by the marginal variances of predictions. As such, VTL greedily minimizes the marginal variances.
> The pessimistic safe set is defined as $\mathcal{S}_n = \\\{x \mid l_n(x) \geq 0\\\}$ where the lower confidence bound is roughly the mean minus the standard deviation (see eq. (25) in Section C.8).
> That is, minimizing the variance at $x$ is the most effective way of determining whether $x$ can be added to $\mathcal{S}_n$ or not.
>
> >I'm not familiar with this task, but in my opinion, it does not provide enough details and explanations.
>
> Given the limited space in the main body of the paper, we had to extract some of the background and details to the appendix.
> We include an extensive discussion of the relevant background in Appendix K.2.
>
> ---
>
> We hope to have addressed the reviewer's concern regarding the novelty of our paper, which
> 1. introduces a new problem setting generalizing classical inductive active learning,
> 2. derives novel convergence guarantees for a natural family of algorithms, and
> 3. shows the superior performance of these algorithms on two new applications of active learning (unlocked by the generalized problem setting) as compared to the state-of-the-art.
>
> We would appreciate it if the reviewer would increase their score for "contribution" and their overall score for our paper. We would be happy to answer any remaining questions or concerns.

---

> ### Author Response · Authors · 2024-08-09
> **Discussion Period**
>
> Dear Reviewer,
>
> We hope this message finds you well. We have noticed that our detailed rebuttal, addressing each of the concerns raised in your review, has not yet received any feedback. We understand the demanding nature of the review process and appreciate the time and effort invested in evaluating our work.
>
> We kindly urge you to consider our responses to your questions, as we believe they adeptly address your concerns. With some days left, we hope we can still have a fruitful rebuttal period.
>
> Thank you, Authors

---

> > ### Comment · Reviewer_yyUG · 2024-08-10
> >
> > Since one of the main claimed contributions is the theory part, I would like to take a deeper look. I'm not very familiar with theory in gaussian process or bayesian learning, so it would take some time.
> >
> > Questions and comments so far:
> >
> > - Given there are a lot of notations involved, it would be clearer if you could provide a list of notations for reference.
> >
> > - It is still unclear to me how strong/tight your results are. It would be easier if you could refer to some similar upper or lower bounds.
> >
> >   - For Thm 3.2, is C a universal constant, or it depends on other things? Specifically, how does it scale with the size of A, S, or X, and what happens if they're infinite (I saw your comment to Reviewer WV1c13, but could you give some quantitative characterization)?
> >
> >    - For thm 3.3
> >       - how realistic are the assumptions (bounded norm for f*, sublinear \gamma_n)?
> >       - It looks \beta_n is increasing as n increases, won't this make the bound very weak?
> >       - Why isn't there a term that characterizes how close f* can be approximated by the model class (Gaussian process)?
> >
> > - For Lemma C.16 and the comments thereafter, how can you conclude b_\epsilon is universal constant? doesn't it depends on all other terms in inequality (15)?

---

> > > ### Author Response · Authors · 2024-08-10
> > >
> > > We thank the reviewer for taking a close look at our novel convergence guarantees.
> > >
> > > > It is still unclear to me how strong/tight your results are. It would be easier if you could refer to some similar upper or lower bounds.
> > >
> > > One can show that $\sigma_n^2(\boldsymbol{x}) \geq \frac{1}{n+1} \sigma_0^2(\boldsymbol{x}) \in \Omega(1/n)$ which is achieved if the point $\boldsymbol{x}$ is sampled repeatedly (and no other points have to be generalized to). Please see eq. (7.26) of [1]. The derivation is along the lines of [2, Section 2].
> > > In particular, our eq. (2) is tight up to $\gamma_n$.
> > >
> > > Our results in the agnostic kernelized setting (Theorem 3.3) match regret bounds of kernelized bandit algorithms in the same setting.
> > > For example, [3, Theorem 4] prove for the celebrated and widely used GP-UCB algorithm that $R_n / n \leq \widetilde{O}(\frac{\gamma_n + \|\|{f^*}\|\|_k \sqrt{\gamma_n}}{\sqrt{n}})$ where $R_n$ is the cumulative regret up to round $n$ and $\widetilde{O}(\cdot)$ subsumes log-factors.
> > > A similar result can be found in [4, Theorem 3] (ICML 2020 Test of Time Award winner).
> > > Our results match the rates of results in this widely cited and often applied line of work on kernelized bandits.
> > >
> > > > For Thm 3.2, is $C$ a universal constant, or it depends on other things? Specifically, how does it scale with the size of $\mathcal{A}$, $\mathcal{S}$, or $\mathcal{X}$, and what happens if they're infinite (I saw your comment to Reviewer WV1c13, but could you give some quantitative characterization)?
> > >
> > > You can find the constant $C$ defined precisely in the proof of Theorem 3.2 (cf. lines 1227 and 1229). $C$ is independent of $n$.
> > > For a finite discretization of a continuous domain, our results hold as stated in the main body of the paper.
> > > We generalize our analysis to continuous domains when the RKHS is finite-dimensional in Section C.6.4.
> > > With this generalization, intuitively, the dimensionality of the RKHS assumes the role of the size of $\mathcal{S}$ from the finite setting.
> > >
> > > > For thm 3.3, how realistic are the assumptions (bounded norm for f*, sublinear \gamma_n)?
> > >
> > > $f^*$ has bounded norm iff it is element in the corresponding RKHS. We remark that for standard kernels (such as RBF/squared exponential or Matérn kernels), the space of functions is extremely large; containing, e.g., the smooth or the continuous functions defined over $\mathcal{X}$.
> > > Our dependence on $\|\|f^*\|\|$ and $\gamma_n$ are standard compared to the large literature on kernelized bandits [e.g., 3] as discussed above.
> > >
> > > We provide the magnitudes of $\gamma_n$ for common kernels in Table 3 (last page of Appendix). $\gamma_n$ is logarithmic in $n$ for many common kernels. We further remark that $\gamma_n$ is a very standard measure of "information capacity" / "function class complexity" in the literature on kernelized bandits. [3] is merely one of the many works that fundamentally rely on this notion.
> > >
> > > > It looks \beta_n is increasing as n increases, won't this make the bound very weak?
> > >
> > > Yes, $\beta_n(\delta)$ is increasing with $n$, however, only very slowly at the rate $\sqrt{\gamma_n}$.
> > > This dependence on $\beta_n$ is standard in the literature on kernelized bandits [e.g., 3, Theorem 2].
> > >
> > > > Why isn't there a term that characterizes how close f* can be approximated by the model class (Gaussian process)?
> > >
> > > The term $\|\| f^* \|\|$ measures "how easy" it is to find $f^*$ within the model class (the RKHS) with $\|\| f^* \|\| = \infty$ if $f^*$ is not at all in the model class.
> > > The term $\gamma_n$ measures the "size" / "capacity" of the model class.
> > >
> > > > For Lemma C.16 and the comments thereafter, how can you conclude b_\epsilon is universal constant? doesn't it depends on all other terms in inequality (15)?
> > >
> > > We mean "universal" with respect to $n$ and $\boldsymbol{x}$. We are very sorry for the confusion and have updated this line to point this out explicitly. Thank you for highlighting this!
> > >
> > > ---
> > >
> > > In summary, we would like to mention:
> > >
> > > 1. Our results match rates of prior work from the widely recognized and large literature on kernelized bandits. We use the same measures of size and complexity of model class as most works from kernelized bandits.
> > >
> > > 2. Our dependence on $\gamma_{\mathcal{A}, \mathcal{S}}(n)$ can be dramatically better than $\gamma_n$, since this only measures the information capacity of prediction targets $\mathcal{A}$ as opposed to of the entire domain $\mathcal{X}$.
> > >
> > > We hope to have addressed the reviewer's questions adeptly.
> > > We are happy to answer any other questions that may help with the interpretation of our results!
> > >
> > > ---
> > >
> > > 1. Williams and Rasmussen. Gaussian processes for machine learning, volume 2. MIT press Cambridge, MA, 2006.
> > > 2. Opper and Vivarelli. General bounds on bayes errors for regression with gaussian processes. NeurIPS, 11, 1998.
> > > 3. Chowdhury and Gopalan. On kernelized multi-armed bandits. ICML, 2017.
> > > 4. Srinivas et al. Gaussian process optimization in the bandit setting: No regret and experimental design. ICML, 2010.

---

> > > > ### Author Response · Authors · 2024-08-10
> > > > **Brief List of Notation**
> > > >
> > > > As requested, the following includes a brief list of notations that are defined in the appendix. Kindly let us know if you have a question about any other symbol.
> > > >
> > > > | Symbol | Reference |
> > > > |-|-|
> > > > | $\sigma^2$ | maximum initial variance (line 1015) |
> > > > | $\tilde{\sigma}^2$ | maximum [initial variance + noise variance] (line 1016) |
> > > > | $\Delta_{\mathcal{A}}(\boldsymbol{x} \mid X)$ | marginal gain (Lemma C.2) |
> > > > | $\Gamma_n$ | maximal marginal gain in iteration $n+1$ (line 1084) |
> > > > | $\alpha_{\mathcal{A}, \mathcal{S}}(n)$ and $\alpha_n$ | task complexity (definition C.6) |
> > > > | $\bar{\kappa}(X \mid D)$ | information ratio (eq. (8)) |
> > > > | $\kappa_{\mathcal{A}}(n)$ | submodularity ratio (definition C.10) |
> > > > | $b_{\epsilon}$ | bound on size of approx. Markov boundary (eq. (16)) |
> > > >
> > > > The definitions of $\alpha, \bar{\kappa}, \kappa$ are all to generalize to instances where Assumption 3.1 is satisfied approximately. Under Assumption 3.1, all can be bounded trivially: $\alpha \leq 1$ and $\bar{\kappa}, \kappa \geq 1$.

---

> > > > > ### Comment · Reviewer_yyUG · 2024-08-11
> > > > >
> > > > > Thanks for the explanation and the added list of notation! I think it would make the paper clearer if some of them are added to the future version (e.g. list of notation, how the results or parameters compared to existing literature, and what they would be in some examples like Table 3).
> > > > >
> > > > > I still have a few questions:
> > > > >
> > > > > - Tightness of Thm 3.2:
> > > > >   - Is this notion of "constant" C also standard in literature? I'm a bit concerned about it since it involves many factors ($\sigma^2$, $|S|$, $1/\lambda_\min(K)$) which do not look negligible to me. In particular, it looks not very strong to me since it scales linearly in  $|S|$, and it involves maximum initial variance for every example in A (if $\sigma_0^2(x)$ is mostly small but extremely big for some points, the bound of $\sigma_n^2(x)$ would be quite big, even for x such that $\sigma_0^2(x)$ is small). And perhaps it can be clearer if you can explain exactly what it is (up to universal constants) in some specific examples.
> > > > >   - I would expect there to be a smoother transition between (1) and (2) as x moves into $A\cap S$. Is there any explanation why it goes from $1/\sqrt{n}$ to $1/n$ directly?
> > > > >
> > > > > - Tightness of Thm 3.3: For results that you cited (e.g. [3, Theorem 4]), the regret diminishes as $n \to \infty$, but for your bound, the irreducible part gets even larger. Can you comment on this?
> > > > >
> > > > > - Proof: For the inequality following line 1225, how is the $\bar{\kappa}_n$ term bounded?

---

> ### Author Response · Authors · 2024-08-11
>
> We thank the reviewer for their continued look into our new theoretical results.
> We will add a version of this table of notation to the camera-ready version as well as a brief discussion on related results from other problem settings. Thank you for this suggestion and improving our work in this regard.
>
> > Is this notion of "constant" $C$ also standard in literature? I'm a bit concerned about it since it involves many factors which do not look negligible to me. In particular, it looks not very strong to me since it scales linearly in $|\mathcal{S}|$, and it involves maximum initial variance for every example in A. And perhaps it can be clearer if you can explain exactly what it is (up to universal constants) in some specific examples.
>
> The dependence on the initial variance is standard in kernelized analysis of bandits. Usually it is assumed that the kernel is such that $k(\boldsymbol{x},\boldsymbol{x}) \leq 1$ which alleviates this.
> In our analysis, the dependence on $|\mathcal{S}|$ is required to achieve convergence to the irreducible variance (kindly see the discussion below).
> We would be happy to make this dependence evident in the main theorem statement if the reviewer thinks this is important.
> A linear dependence on the size of $|\mathcal{S}|$ is standard in works on safe exploration (i.e., where "extrapolation" is required due to restrictions on the domain) [see, e.g., 1].
>
> We further thank the reviewer for the suggestion, and will update the appendix to clarify whenever we mention "universal constant" what this term is a constant with respect to.
>
> > I would expect there to be a smoother transition between (1) and (2) as x moves into . Is there any explanation why it goes from  to  directly?
>
> The difference in rates is indeed interesting.
> In our analysis this stems from using a "nested" learning problem that learns the function over all of $\mathcal{S}$.
> This appears to be required in our view to achieve the *very strong* convergence to the *irreducible* variance $\eta_{\mathcal{S}}^2(\boldsymbol{x})$, since there is a slight mismatch between the original learning problem wrt. $\mathcal{A}$ and a different learning problem wrt. $\mathcal{S}$.
> It might be possible to obtain tighter results in the special case $\mathcal{S} \subseteq \mathcal{A}$ (i.e., when the two learning problems are not misaligned).
> We are inclined to leave this study as an interesting direction for future work.
>
> > Tightness of Thm 3.3: For results that you cited (e.g. [3, Theorem 4]), the regret diminishes as $n\to\infty$, but for your bound, the irreducible part gets even larger. Can you comment on this?
>
> We would like to point out that in the referenced works on bandits no "extrapolation" is required since *all* points can be observed directly. These works therefore do not have a notion of "irreducible error".
> However, we agree while the slight increase of this error in the agnostic setting is likely an artifact of us performing the analysis in a Bayesian setting, in this Bayesian setting this error is tight.
>
> > Proof: For the inequality following line 1225, how is the $\bar{\kappa}_n$ term bounded?
>
> It follows from Assumption 3.1 (submodularity) that $\bar{\kappa}_n \geq 1$. We have updated the proof to make this more explicit.
>
> ---
>
> We would further like to highlight that due to the large set of instances of *transductive* active learning, our results constitute new results in multiple settings:
> 1. The *extrapolation* to points outside $\mathcal{S}$ which unlocks the application to Safe BO, leading to a new sota algorithm.
> 2. *Directed* learning where $\mathcal{A} \subseteq \mathcal{S}$. Here the analysis simplifies and we prove the rate $\gamma_{\mathcal{A},\mathcal{S}}(n)/n$.
>
> Finally, we would like to mention that beyond our theoretical contributions, the application of transductive active learning to two new applications, Safe BO and the discussed fine-tuning setting, are also key contributions of our work.
> In both of these settings our findings show that our proposed approaches to transductive active learning substantially improve upon the current state of the art.
>
> We hope that upon reevaluation these substantial contributions are acknowledged by the reviewer by increasing their score.
>
> ---
>
> 1. Sui, Yanan, et al. "Safe exploration for optimization with Gaussian processes." International conference on machine learning. PMLR, 2015.

---

> > ### Comment · Reviewer_yyUG · 2024-08-13
> >
> > Thanks again for the detailed explanation. I will adjust my review.

---

> > > ### Author Response · Authors · 2024-08-13
> > >
> > > We sincerely appreciate you for reviewing and giving insightful comments on our manuscript. Thank you.

---

### Official Review · Reviewer_aHJz · 2024-07-06

**Soundness:** 3
**Presentation:** 4
**Contribution:** 3
**Rating:** 4
**Confidence:** 4

**Summary:**

The paper investigates the generalization of active learning to scenarios with specific prediction targets and limited information due to constrained sample spaces, offering a flexible framework adaptable to various domains such as recommender systems, molecular design, and robotics. It introduces novel generalization bounds of independent interest for active learning. The practical performance of the model relies on its ability to capture correlations between points (learning the latent structure) and accurately estimate uncertainty. The study demonstrates that sampling relevant and diverse points significantly improves performance across various applications, surpassing the state-of-the-art methods.

**Strengths:**

The paper is well written and clearly structured with novel contributions in generalization bounds. It provides sufficient theoretical analysis for specific acquisition functions in active learning with experimental results on some datasets.

**Weaknesses:**

The authors use MNIST and CIFAR100 as experiments on active few-shot tuning. First, image classification is a well-studied topic and lack of novelty. Image classification uses convolutional networks for classifier while current LLMs uses auto-regressive models therefore most of theoretical assumptions and results might not be able to extend. Even though the paper is well-written, the claims that the results shall be able to extend to prompt optimization is groundless. It would be more interesting to study the prompt tuning/efficient data selection for fine-tuning LLMs.
Second, the baselines for active learning is not state-of-the-art. For instance, there are more state-of-the-art baselines than badge, typicluster and probcover, for AL domains in image classification. Moreover, there has been work in using influence function to select data to fine tune LLMs or image classifications that worth comparing.

**Questions:**

N/A

---

> ### Author Response · Authors · 2024-07-31
> **Ask for Clarification**
>
> We thank the reviewer for their review. To address the points raised in our rebuttal effectively, we kindly ask for some clarification and additional references.
>
> > For instance, there are more state-of-the-art baselines than badge, typicluster and probcover, for AL domains in image classification.
>
> We would greatly appreciate it if the reviewer could point us to those state-of-the-art baselines. We would be more than happy to compare against additional approaches. To the best of our knowledge, BADGE, TypiClust, and ProbCover are some of the most widely recognized and best-performing AL methods for neural network training to date.
>
> > Moreover, there has been work in using influence function to select data to fine tune LLMs or image classifications that worth comparing.
>
> As we are not fully familiar with this specific line of work, we would be grateful if the reviewer could provide references that they believe are relevant for comparison.
>
> Thank you in advance for your assistance in providing these clarifications. We believe that this will facilitate a more productive rebuttal process.

---

> > ### Comment · Reviewer_aHJz · 2024-08-03
> >
> > 1. There are some active learning papers published in vision venues.
> > [1] Li, Jingyao, et al. "Bal: Balancing diversity and novelty for active learning." IEEE Transactions on Pattern Analysis and Machine Intelligence (2023).
> > [2] Parvaneh, Amin, et al. "Active learning by feature mixing." Proceedings of the IEEE/CVF conference on computer vision and pattern recognition. 2022.
> >
> > 2. Below are references for influence function to finetune language models or image classification tasks.
> > [1] Kwon, Yongchan, et al. "Datainf: Efficiently estimating data influence in lora-tuned llms and diffusion models." arXiv preprint arXiv:2310.00902 (2023).
> > [2] Liu, Zhuoming, et al. "Influence selection for active learning." Proceedings of the IEEE/CVF international conference on computer vision. 2021.
> > [3] Zhao, Zhuokai, Yibo Jiang, and Yuxin Chen. "Direct Acquisition Optimization for Low-Budget Active Learning." arXiv preprint arXiv:2402.06045 (2024).
> > [4] Xia, Mengzhou, et al. "Less: Selecting influential data for targeted instruction tuning." arXiv preprint arXiv:2402.04333 (2024).

---

> ### Author Rebuttal · Authors · 2024-08-06
>
> Thank you for reviewing our paper! We also very much appreciate the quick response to our follow-up question. You can find below our response to some of your questions. Please let us know if you have any further concerns or suggestions.
>
> > The paper is well written and clearly structured with novel contributions in generalization bounds. It provides sufficient theoretical analysis for specific acquisition functions in active learning with experimental results on some datasets.
>
> Thank you!
>
> ## Concerns
>
> > The authors use MNIST and CIFAR100 as experiments on active few-shot tuning. First, image classification is a well-studied topic and lack of novelty.
>
> We chose image classification as a testbed *precisely because* it is widely studied.
> To the best of our knowledge, MNIST and CIFAR are common tasks for evaluation of active learning [1-3] as well as other works on fine-tuning [e.g., 4].
> Given that baselines and benchmarks on active learning have been using image classification as testbeds, we intend with this choice to compete against the strongest possible baselines.
>
> This being said, we strongly agree that applying VTL to other domains (for example, fine-tuning of LLMs) is a very exciting research direction, which we intend to pursue in future work.
>
> > Image classification uses convolutional networks for classifier while current LLMs uses auto-regressive models therefore most of theoretical assumptions and results might not be able to extend. Even though the paper is well-written, the claims that the results shall be able to extend to prompt optimization is groundless. It would be more interesting to study the prompt tuning/efficient data selection for fine-tuning LLMs.
>
> We agree that the results might not extend directly to LLMs, and that this is an exciting direction for further research.
> Can you point us to the claim that "our results improve prompt optimization"?
> We did not intend to make this claim.
> In lines 251-256, we address *model optimization* (i.e., "fine-tuning" model weights, like we do here with vision experiments) and suggest that our work opens up this direction of future work.
>
> ### Related work on Inductive Active Learning
>
> We would like to emphasize that Section 4 specifically addresses the (transductive) transfer setting, where we fine-tune to a specific set of (out-of-distribution) examples. This is different from the canonical application of (inductive) active learning to pre-training or fine-tuning without distribution mismatch.
> We compare against around 10 of the most widely used inductive active learning strategies.
> You can find the comparison to all baselines in Section J.5.
> Our experiments show that we outperform *all inductive baselines significantly*.
> We believe that this constitutes a comprehensive evaluation of approaches to inductive active learning, and shows the advantage of transductive active learning.
> We do not expect any alternative inductive methods to perform dramatically better since they do not take the distribution shift into account.
>
> ### Related Work on Influence Functions
>
> Thank you for pointing out this line of work!
> In particular, [5] which was published at ICML 2024, is relevant to the transfer setting we also consider in Section 4.
>
> The following is a brief comparison between our work and the influence functions line of work:
> - We remark that the selection strategy of [5] is the same as the "CosineSimilarity" baseline considered in our experiments, when loss gradient embeddings (cf. Section J.2) are used (and SGD is the used optimizer). We show in our experiments that VTL substantially outperforms CosineSimilarity when the batch size is larger than one.
> - A major limitation of the line of work on influence functions, including [5], is that interactions between samples are not taken into account. That is, the selected samples are not diverse and may contain redundant information. We suspect that this is the main reason why VTL substantially outperforms these methods. [5] discuss this in their limitations.
>     VTL offers an efficient and effective approach to take interactions between samples and account: synthesizing approaches to retrieve *relevant* examples (e.g., CosineSimilarity) and approaches to retrieve *diverse* examples (e.g., inductive active learning).
> - A drawback that [5] points out in Appendix I is that a smaller loss does not always lead to a higher accuracy. Perhaps the loss is therefore not the right proxy goal. In contrast, VTL directly minimizes the bound on the approximation error from Theorem 3.3.
> - VTL also takes into account points with a high negative cosine similarity; something that [5] suggests in Appendix K.2.
>
> We will include this comparison to the influence functions line of work in our section "Related Work".
>
> ---
>
> We hope to have addressed the concerns raised by the reviewer.
> Given the recognized contributions that unlock new applications of active learning, and that the application in Section 4 offers a natural and effective way to select both *relevant and diverse* points, we would appreciate it if the reviewer would increase their score for our paper. We would be happy to answer any remaining questions or concerns.
>
> ---
>
> References:
> 1. Gal, Y., Islam, R., and Ghahramani, Z. Deep bayesian active learning
> with image data. In ICML, 2017.
> 2. Zhang, Jifan, et al. Labelbench: A comprehensive framework for benchmarking adaptive
> label-efficient learning. 2023.
> 3. Lüth, Carsten, et al. Navigating the pitfalls of active learning evaluation: A systematic framework
> for meaningful performance assessment. In NeurIPS, 2023.
> 4. Wei, A., Hu, W., and Steinhardt, J. More than a toy: Random matrix models predict how real-world neural representations generalize. In ICML, 2022.
> 5. Xia, Mengzhou, et al. ”Less: Selecting influential data for targeted instruction tuning.” In ICML, 2024.

---

> > ### Comment · Reviewer_aHJz · 2024-08-11
> >
> > Hi I have read all the rebuttals. I do appreciate the authors efforts in literature review and agree that the references the authors list are more comprehensive. However, I think fine-tuning model weights for image classification task would not serve as a novel contribution and the learning paradigm for fine tuning deep neural networks is different from fine tuning LLMs as LLMs are generative models (the authors cite [1] and argue this paper could be an extension for the future work for fine-tuning the model for each prompt.) I think the contribution below is not novel and a bit vague.
> >
> > "We apply the transductive active learning framework to batch-wise active few-shot fine tuning of large neural networks and to safe Bayesian optimization. We empirically show, in both cases, that ITL and VTL outperform the state-of-the-art.
> >
> > [1] Hardt, M. and Sun, Y. Test-time training on nearest neighbors for large language models. ICLR, 2024.
> >
> > I would lean towards keeping my score.

---

> ### Author Response · Authors · 2024-08-09
> **Discussion Period**
>
> Dear Reviewer,
>
> We hope this message finds you well. We have noticed that our detailed rebuttal, addressing each of the concerns raised in your review, has not yet received any feedback. We understand the demanding nature of the review process and appreciate the time and effort invested in evaluating our work.
>
> We kindly urge you to consider our responses to your questions, as we believe they adeptly address your concerns. With some days left, we hope we can still have a fruitful rebuttal period.
>
> Thank you, Authors

---

> ### Author Response · Authors · 2024-08-11
> **Response to reviewer’s comment**
>
> Dear reviewer,
>
> Thanks for your active engagement in the rebuttal period. We have a follow up question on your comment:
>
> “However, I think fine-tuning model weights for image classification task would not serve as a novel contribution.”
>
> First, while we might be of a different opinion, we respect your opinion regarding the contribution of our fine tuning experiments. However, we would like to mention that there are some very important other contributions of our paper that should also be considered in the evaluation (below is a non exhaustive list):
>
> 1. We formulate the problem of TAL which results in natural decision rules/acquisition functions of ITL, VTL.
>
> 2. We are the first to give uniform convergence rates for the posterior uncertainty in regions beyond the sample space. This implies a new generalisation bound for RKHS functions. Note that our assumptions are very general and common in active learning literature.
>
> 3. We evaluate our method on two distinct domains, (i) fine-tuning of large neural networks and (ii) safe BO. In both cases, our method outperforms the baselines.
>
> We hope this clarifies our contributions further. Moreover, even though the reviewer thinks our contribution for the fine tuning case are not novel, our work has other significant contributions. Given these contributions, we kindly ask the reviewer to reconsider their evaluation of our work. Looking forward to the reviewer’s response.

---

> ### Author Response · Authors · 2024-08-13
>
> Dear reviewer, we would like to thank you for your efforts in reviewing our submission and pointing us to the related work on influence functions. We will include the above comparison to this line of work in our updated version.
>
> Regarding our citation of [1], our intention was to mention that in future work, the methods of this submission could potentially be extended to the setting described in [1].
> We did *not* intend to suggest that this submission is an extension to [1].
> We agree with the reviewer that our methods cannot be applied trivially (due to the different learning paradigms), however, selecting relevant and diverse data seems also to be important for the fine-tuning of LLMs (as argued in the limitations of [2]).
> We will clarify this in the updated version, and thank the reviewer for highlighting this.
>
> With only one day left, we would appreciate it if the reviewer can express their thoughts about our latest comment and potentially raise their score.
>
> [1] Hardt, M. and Sun, Y. Test-time training on nearest neighbors for large language models. ICLR, 2024.
>
> [2] Xia, Mengzhou, et al. ”Less: Selecting influential data for targeted instruction tuning.” In ICML, 2024.

---

### Official Review · Reviewer_4uNa · 2024-07-12

**Soundness:** 3
**Presentation:** 3
**Contribution:** 3
**Rating:** 7
**Confidence:** 2

**Summary:**

This paper introduces and considers approaches for the generalized (transductive) active learning problem, where the space of prediction targets and samples are not necessarily the same. The authors propose methods ITL and VTL to select samples in order to minimize the posterior uncertainty about the target function with the target space $A$. They prove favorable properties of these methods and compare their performance to other active learning techniques on empirical settings, including batch active learning and safe Bayesian optimization.

**Strengths:**

- The paper is well-written and organized, with ample thorough discussions.
- The problem of transductive active learning is novel to the best of my knowledge and may be of interest to the broader ML community.
- Compelling empirical results are provided that support the theoretical results of the paper.

**Weaknesses:**

- The paper is quite dense and long, with a total of 68 pages when appendices are considered.

**Questions:**

1. How does the computational cost of the methods fare in practice to other methods?

**Limitations:**

Yes

---

> ### Author Rebuttal · Authors · 2024-08-01
>
> Thank you for reviewing our paper! Please find our response to your question.
> Please let us know if you have any further
> concerns or suggestions.
>
> > - The paper is well-written and organized, with ample thorough discussions.
> > - The problem of transductive active learning is novel to the best of my knowledge and may be of interest to the broader ML community.
> > - Compelling empirical results are provided that support the theoretical results of the paper.
>
> Thank you!
>
> > How does the computational cost of the methods fare in practice to other methods?
>
> ITL scales cubically in the size of the target space $\mathcal{A}$ since it takes into account mutual interaction between prediction targets.
> In contrast, VTL scales in linearly in size of $\mathcal{A}$.
> We included a more detailed analysis of the computational complexity in Appendix G.
> In all our experiments, the computational cost of ITL \& VTL is around the cost of the fastest baselines, since for these computationally light methods cost tends to be dominated by computing the kernel / the embeddings.
>
> ---
>
> We are glad to see the recognition that transductive active learning can be of interest to the broader ML community, given that it unlocks new applications of active learning.
> Having addressed the remaining question provided by the reviewer, and given the contributions of this paper, we hope that the reviewer considers to increase their score for our paper. We would be happy to answer any remaining questions or concerns.

---

> > ### Comment · Reviewer_4uNa · 2024-08-10
> >
> > Thank you for your response. I read the reviews and the responses. I lean towards keeping my score.

---

> > > ### Author Response · Authors · 2024-08-13
> > >
> > > We would like to thank you for your efforts in reviewing our work and for recognizing its relevance to the broader ML community.

---

### Official Review · Reviewer_WV1c · 2024-07-13

**Soundness:** 3
**Presentation:** 3
**Contribution:** 3
**Rating:** 7
**Confidence:** 4

**Summary:**

This paper introduces transductive active learning based on the uncertainty of GP when the labeling space and target space can be different. They have assumptions such as submodularity of information gain and information capacity’s sub-linearity in GP, resulting in a bound for variance of GP’s posterior with the proposed labeling algorithm. Under a specific setup, this can go to 0, and it cannot avoid the irreducible error when the deviation between labeling space and target space is larger. The various properties of the proposed algorithm(ITL and VTL) are examined by simple examples showing labeling in the target space as much as possible.
The proposed algorithm's applications are interesting, including few-shot learning and safe BO (rarely considered in conventional active learning). These parts show the algorithm’s flexibility and applicability well. The results are promising.

**Strengths:**

Among many advantages, the theoretical aspect is better because it can provide the criteria for the use of kernels that are strongly connected to the tuning of the algorithms. Also, the applications are impressive and enlarging the area of active learning. The use of GP is solid because it can provide a solid mathematical background and reliable uncertainty in many cases, compared to variational Bayes approaches.

**Weaknesses:**

This paper gives too little attention to conventional active learning. Although the novelty of the new area is essential in this paper, traditional active learning tasks are still popular. In this view, this paper does not provide strong evidence for the superiority compared to conventional active learning. At first glance, the proposed active learning algorithm seems too specific. In theoretical aspects, it is unsatisfactory to consider only the spaces. More clarification of theoretical aspects in the paper can be helpful to strengthen the paper. $gamma_n$’s sub-linearity can be clarified as the mathematical formula. In GP, the kernel is essential, and the experiments can reveal what kernel is best or robust. The discussion about kernels is too short.

**Questions:**

Q1: What’s the difference between $\gamma_n$ and  $\gamma_{\mathcal{A},\mathcal{S}}(n).$ These notations can be improved.
Q2: If only the target space is (uncountable) separable space, can the theorem have any change?

**Limitations:**

Not well-discussed

---

> ### Author Rebuttal · Authors · 2024-08-06
>
> Thank you for your review and detailed comments! You can find below our response to some of your questions. Please let us know if you have any further concerns or suggestions.
>
> > Among many advantages, the theoretical aspect is better because it can provide the criteria for the use of kernels that are strongly connected to the tuning of the algorithms. Also, the applications are impressive and enlarging the area of active learning.
>
> Thank you!
>
> ## Concerns
>
> > This paper gives too little attention to conventional active learning. Although the novelty of the new area is essential in this paper, traditional active learning tasks are still popular. In this view, this paper does not provide strong evidence for the superiority compared to conventional active learning. At first glance, the proposed active learning algorithm seems too specific.
>
> Thank you for pointing this out!
> Our paper introduces *transductive* active learning (TAL), where instances are specified by a target space $\mathcal{A}$ and a sample space $\mathcal{S}$.
> The "classical" (*inductive*) active learning (we will call it IAL in the following) that is studied in most prior works on active learning is the special case of TAL where $\mathcal{A} = \mathcal{S}$.
> That is, intuitively, in IAL one tries to learn "everything" which amounts to extracting as much information as possible.
> In contrast, in TAL, learning *can* be directed to a particular region of the domain.
> This is also why the special case of IAL seems most useful for pre-training (and it has been studied extensively in this setting), while TAL unlocks various applications of active learning for fine-tuning.
>
> We allude to this in lines 84-86 where we show that in the classical IAL setting, ITL is equivalent to CoreSet which is a standard baseline.
> So if one were to apply the methods proposed here in the IAL setting, one would recover the existing results.
> This goes to show that our proposed algorithm is in fact *more general* than prior approaches since it does not only address IAL but also the more general TAL.
> To argue for the advantages of TAL over the limited view of IAL, we therefore focus our experiments on novel applications of TAL.
> As you rightly point out, IAL has already been extensively studied, and TAL unlocks many new applications of active learning.
>
> > $\gamma_n$’s sub-linearity can be clarified as the mathematical formula.
>
> Thank you for this suggestion. We updated the paper to more prominently pointing out that in most cases (including all instances where the domain is finite), $\gamma_n$ is always sub-linear.
>
> > In GP, the kernel is essential, and the experiments can reveal what kernel is best or robust. The discussion about kernels is too short.
>
> Thank you also for this suggestion.
> Kernels are discussed extensively in other resources (e.g., the kernel cookbook [1]).
> Given the limited space in the submission, we opted to not include an extensive discussion on kernels, beyond the extensive discussion in section 4, lines 182-194.
> With the additional page of the camera-ready version, we are able to extend this discussion slightly and reference background resources.
>
> > Limitations: Not well-discussed
>
> The main limitation of this work is that we focus solely on sequential decision-making *given* some model, rather than asking how one should construct such a model so that it is representative of the ground truth.
> We implicitly address this scope in lines 12-13.
> Kindly let us know of any other potential limitations, and we would be more than happy to address them.
>
> ## Questions
>
> > Q1: What’s the difference between $\gamma_n$ and $\gamma_{\mathcal{A}, \mathcal{S}}(n)$?
>
> $\gamma_n = \max_{\substack{X \subseteq \mathcal{X}}, |X| \leq n} \mathrm{I}(f_{\mathcal{X}} ; y_{X})$, whereas $\gamma_{\mathcal{A},\mathcal{S}}(n) = \max_{\substack{X \subseteq \textcolor{red}{\mathcal{S}}}, |X| \leq n} \mathrm{I}(f_{\textcolor{red}{\mathcal{A}}} ; y_{X})$.
> $\gamma_n$ is used extensively in prior literature that studies inductive active learning. The definition of $\gamma_{\mathcal{A},\mathcal{S}}(n)$ is motivated by the fact that transductive active learning is *generalizing* inductive active learning where $\mathcal{X} = \mathcal{A} = \mathcal{S}$.
> In the inductive case, $\gamma_n = \gamma_{\mathcal{A},\mathcal{S}}(n)$.
> In other transductive cases, it can be that $\gamma_{\mathcal{A},\mathcal{S}}(n) \ll \gamma_n$.
>
> > Q2: If only the target space is (uncountable) separable space, can the theorem have any change?
>
> Infinite (but compact) target spaces can be addressed, e.g., via discretization arguments which are common in the Bayesian optimization literature (see, e.g., appendix C.1 of [2]).
> That is, if the target space can be covered approximately using a finite set of points, Theorems 3.2 and 3.3 extend directly.
>
> ---
>
> We believe that our work is highly relevant since (1) it generalizes widely studied approaches to IAL while (2) unlocking entirely new promising applications of active learning where the use of prior approaches was limited.
>
> Having addressed all of the questions provided by the reviewer, and given the contributions of this paper, we hope that our answers prompt the reviewer to reconsider their evaluation and potentially increase their score. We would be happy to answer any remaining questions or concerns.
>
> ---
>
> References:
> 1. Kernel cookbook. https://www.cs.toronto.edu/~duvenaud/cookbook/.
> 2. Srinivas, Niranjan, et al. ”Gaussian process optimization in the bandit setting: No regret and experimental design.” ICML (2010).

---

> > ### Comment · Reviewer_WV1c · 2024-08-11
> > **Reply to the Rebuttal**
> >
> > Thanks for your reply. Almost all my concerns are resolved.  I only want the results applied in the conventional AL learning setups, which can be expected shown in other chances.
> > I'll keep my score.

---

> > > ### Author Response · Authors · 2024-08-13
> > >
> > > We sincerely thank you for reviewing and giving insightful comments on our manuscript.

---

### Author Rebuttal · Authors · 2024-08-06

We thank all reviewers for their feedback, and would like to emphasize the novel contributions of our work.

1. We introduce a new problem setting generalizing classical inductive active learning. All reviewers appear to agree that this new problem setting is relevant and unlocks several new applications of active learning.
2. We derive convergence guarantees for a natural family of algorithms (which recover widely recognized approaches such as "uncertainty sampling" for the inductive special case).
3. We show that these algorithms achieve a significant improvement upon the state-of-the-art in two new applications of active learning (unlocked by the generalized problem setting).
In both applications, the use of active learning leads to a unique advantage over prior work: Synthesizing *relevance & diversity* when sampling in Section 4; Synthesizing *expansion & exploration* when sampling in Section 5.

We believe that transductive active learning is a powerful paradigm for learning under resource constraints.
As noted by the reviewers, this work opens up the possibility to study many other applications of transductive active learning, such as efficient data selection for fine-tuning LLMs (reviewer aHJz).

---

### Decision · Program_Chairs · 2024-09-25

**Decision:**

Accept (poster)

**Comment:**

The reviewers agree that the proposed framework is a novel and interesting generalization of active learning that can potentially be used in a variety of applications. They also appreciate the resulting theoretical guarantees. The main weaknesses mentioned were regarding situating the proposed method with regard to previous active learning methods (addressed in the rebuttals) and the strength of the generalization bounds. Due to the generality of the framework, I feel that this paper would be impactful as a foundation on which future work can be done. Therefore, I  would recommend to accept the paper.